# ARECHO: Autoregressive Evaluation via Chain-Based Hypothesis Optimization for Speech Multi-Metric Estimation

**Jiatong Shi**[1]   **Yifan Cheng**[2]   **Bo-Hao Su**[1]   **Hye-jin Shim**[1]   **Jinchuan Tian**[1]
**Samuele Cornell**[1]   **Yiwen Zhao**[1]   **Siddhant Arora**[1]   **Shinji Watanabe**[1]
[1] Carnegie Mellon University
[2] Huazhong University of Science and Technology
jiatongs@cs.cmu.edu

## Abstract

Speech signal analysis poses significant challenges, particularly in tasks such as speech quality evaluation and profiling, where the goal is to predict multiple perceptual and objective metrics. For instance, metrics like PESQ (Perceptual Evaluation of Speech Quality), STOI (Short-Time Objective Intelligibility), and MOS (Mean Opinion Score) each capture different aspects of speech quality. However, these metrics often have different scales, assumptions, and dependencies, making joint estimation non-trivial. To address these issues, we introduce **ARECHO** (*Autoregressive Evaluation via Chain-based Hypothesis Optimization*), a chain-based, versatile evaluation system for speech assessment grounded in autoregressive dependency modeling. ARECHO is distinguished by three key innovations: (1) a comprehensive speech information tokenization pipeline; (2) a dynamic classifier chain that explicitly captures inter-metric dependencies; and (3) a two-step confidence-oriented decoding algorithm that enhances inference reliability. Experiments demonstrate that ARECHO significantly outperforms the baseline framework across diverse evaluation scenarios, including enhanced speech analysis, speech generation evaluation, and, noisy speech evaluation. Furthermore, its dynamic dependency modeling improves interpretability by capturing inter-metric relationships. Across tasks, ARECHO offers *reference-free* evaluation using its dynamic classifier chain to support subset queries (single or multiple metrics) and reduces error propagation via confidence-oriented decoding.

## 1 Introduction

Speech assessment and profiling are essential components in the speech processing community, owing to the inherently complex and multidimensional nature of speech signals (Huang et al., 2022; Yi et al., 2022; Shi et al., 2024; Torcoli et al., 2021). These signals typically encompass various attributes, such as clarity, naturalness, emotional expressiveness, and acoustic quality, which are challenging to characterize comprehensively. While subjective evaluation remains the gold standard for assessing speech quality due to its capacity for nuanced and context-aware judgments, it suffers from several limitations, including inter-rater variability, limited scalability, and insufficient coverage of diverse evaluation dimensions (Huang et al., 2022; Cooper et al., 2024; Zielinski et al., 2008; Loizou, 2011; Jiménez et al., 2021; Naderi et al., 2020). Consequently, objective methods have emerged as scalable and consistent alternatives that aim to approximate subjective judgments (Cooper et al., 2024; Torcoli et al., 2021; Shi et al., 2025a).

Prior work has introduced numerous specialized metrics targeting specific speech characteristics, including speech perceived quality, naturalness in synthesized speech, and paralinguistic features

such as emotion and speaker identity (Reddy et al., 2021; Saeki et al., 2022; Yi et al., 2022; Huang et al., 2024c; Wu et al., 2024; Goncalves et al., 2024; Jung et al., 2024). Although these metrics provide valuable insights individually, they are often analyzed in isolation, overlooking potential inter-dependencies among them. Comprehensive profiling through multi-metric evaluation offers notable advantages, including greater efficiency, more holistic analysis, and the potential to leverage shared information across metrics (Zhang et al., 2024). In light of these benefits, recent research has explored unified frameworks that predict multiple speech assessment metrics concurrently (Kumar et al., 2023; Tjandra et al., 2025; Shi et al., 2025b).

Despite their promise, unified multi-metric prediction systems present several key challenges stemming from the heterogeneity of speech metrics, limited supervision, and the lack of inter-metric reasoning:

*Challenge I: Diverse Scale Issues*. Speech metrics vary significantly in scale and type, complicating joint modeling and optimization. For example, MOS (Mean Opinion Score) ranges from 1 to 5 and SI-SNR is unbounded over $(-\infty, \infty)$. Optimizing such diverse metrics with uniform loss functions (e.g., L1) can lead to biased learning, overemphasizing metrics with larger numerical ranges while under-representing perceptually salient ones like MOS.

*Challenge II: Limited Data Availability*. Many metrics rely on auxiliary references that are often unavailable in practical scenarios. PESQ requires clean reference audio, WER depends on transcripts, and speaker similarity requires paired utterances. As a result, real-world datasets are often partially labeled, demanding systems that can flexibly adapt to semi-supervised or weakly supervised conditions and support arbitrary subsets of supervision targets during both training and inference.

*Challenge III: Dependency Modeling with Flexible Control*. Existing frameworks such as Uni-VERSA (Shi et al., 2025b) and TorchSquim (Kumar et al., 2023) typically predict all metrics independently and in parallel. This limits their ability to leverage the inherent dependencies among metrics, for instance, the natural correlation between intelligibility and naturalness, which could otherwise inform both reasoning and prediction. Moreover, in the presence of incomplete labeling, parallel prediction becomes inefficient and less effective in generalizing from available cues.

To address these challenges, we propose **ARECHO** (Autoregressive Evaluation via Chain-based Hypothesis Optimization), a flexible and dependency-aware evaluation framework for speech assessment. ARECHO formulates speech evaluation as a chain-based prediction task, utilizing a dynamic classifier chain to model inter-metric relationships in an autoregressive fashion. To handle the diverse scales and types of speech metrics (e.g., categorical vs. continuous, bounded vs. unbounded), we design a speech tokenization pipeline that robustly encodes these heterogeneous metric values into a unified token space. This formulation not only improves predictive performance but also improves interpretability through structured dependency reasoning.[1]

The key contributions of this work are as follows:

(1) We design a comprehensive speech tokenization pipeline that explicitly handles the diversity of metric types and scales by encoding them into a consistent and learnable token representation.

(2) We introduce a dynamic classifier chain algorithm that explicitly captures and exploits dependencies among evaluation metrics.

(3) We propose a two-step confidence-oriented decoding strategy that improves robustness and prediction reliability by dynamically adjusting the inference trajectory.

(4) We conduct extensive experiments demonstrating that ARECHO consistently outperforms existing frameworks across multiple speech domains, including synthesized, enhanced, and corrupted speech analysis, while providing improved interpretability and flexibility.

## 2    Related Works

**Speech Assessment**. Speech evaluation has usually been studied in the context of speech generation tasks, particularly speech synthesis and speech enhancement (SE). In these domains, assessment efforts are typically oriented toward task-specific goals, for instance, naturalness in speech synthesis and noise reduction in SE (Cooper et al., 2024; Hu & Loizou, 2007b). In speech synthesis, subjective

---

[1]The code is open-sourced at `https://github.com/ftshijt/espnet/tree/universa_plus`

human evaluation remains the gold standard, with evaluations ranging from general assessments such as overall naturalness to more fine-grained dimensions such as speaking style and expressiveness (Li et al., 2024; Yang et al., 2024; Shimizu et al., 2024; Feng & Yoshimoto, 2024). However, such human evaluations face well-known limitations, including challenges in scaling, score inconsistency, and limited coverage of diverse perceptual factors (Huang et al., 2022; Zielinski et al., 2008; Loizou, 2011; Jiménez et al., 2021; Naderi et al., 2020). To overcome these limitations, recent works have introduced objective evaluation systems that are trained to predict human perceptual scores, such as mean opinion scores (MOS), using supervised learning frameworks (Falk et al., 2008; Yoshimura et al., 2016; Lo et al., 2019; Saeki et al., 2022; Huang et al., 2024c). These models have gained popularity in speech synthesis due to their scalability and ability to generalize to unseen systems. A similar trend is observed in SE, where classical signal-based metrics (e.g., signal-to-noise ratio) are widely used with simulated data experiments (Perlmutter et al., 1977; Hansen & Pellom, 1998; Xu et al., 2013). Yet, they often fail to reflect human perception accurately, especially in real-world, noisy scenarios (Hu & Loizou, 2007b). As a result, perceptually aligned objective metrics have been proposed to bridge the gap between computational evaluation and subjective judgment (Hu & Loizou, 2007a; Reddy et al., 2021; Rao et al., 2021; Yi et al., 2022).

While the above metrics are directly tied to task-specific outcomes, there is growing interest in broader frameworks for general speech assessment or profiling (Zezario et al., 2022a, 2024; Chen & Tsao, 2022; Close et al., 2024; Zezario et al., 2022b; Kumar et al., 2023; Tjandra et al., 2025; Shi et al., 2025a,b). Such frameworks aim to extract a rich set of meta-information that spans across multiple dimensions of the speech signal. This can include not only quality and intelligibility, but also speaker traits, emotional content, and environmental context. In this work, we align with this broader vision and propose a system that moves beyond task-specific metrics to support general-purpose, multi-faceted speech evaluation.[2]

**Multi-Metric Evaluation and Dependency Modeling**. Speech evaluation inherently involves multiple dimensions, such as naturalness, intelligibility, and emotional expression, that are often correlated through shared acoustic and prosodic cues. Effectively capturing the dependencies among these metrics is therefore essential for producing faithful and insightful assessments of speech quality.

Recent work has increasingly focused on general-purpose evaluation frameworks that unify diverse metrics and applications (Zezario et al., 2022a,b, 2024; Chen & Tsao, 2022; Close et al., 2024; Kumar et al., 2023; Tjandra et al., 2025; Shi et al., 2025a,b). These systems aim to provide scalable, task-agnostic profiling across various speech properties, including quality, speaker traits, emotional content, and environmental conditions. Frameworks such as UniVERSA (Shi et al., 2025b) and TorchSquim (Kumar et al., 2023) support multi-metric prediction, but typically treat each metric independently, overlooking the latent correlations that can enhance performance and interpretability. For instance, improvements in naturalness often align with gains in intelligibility or perceived emotion due to overlapping signal characteristics.

In parallel, captioning-based approaches have explored generating natural language rationales to explain metric predictions (Ghosh et al., 2024; Xie et al., 2025; Ghosh et al., 2025; Wang et al., 2025b; Deshmukh et al., 2025; Ma et al., 2025; Kuan & Lee, 2025; Wen et al., 2025; Wang et al., 2025a; Huang et al., 2024a, 2025; Chen et al., 2025). While these methods offer interpretability, they often rely on pretrained LLMs and may struggle with precision or non-textual metrics. ARECHO instead focuses on structured, metric-token-based modeling, enabling exact scoring and more fine-grained control.

## 3 ARECHO

### 3.1 Base Task Formulation

We adopt the general task formulation from (Shi et al., 2025b; Kumar et al., 2023) for speech multi-metric estimation, with an extension to explicitly handle categorical metrics in addition to numerical ones. Let $i^{\text{th}}$ paired sample from dataset $\mathcal{D}$ be represented as $(\mathbf{S}^i, Y^i)$, where $\mathbf{S}^i$ denotes a single-channel speech signal and $Y^i$ denotes the set of associated evaluation metrics. Each $Y^i = \{y_b^i\}_{b \in \mathcal{B}}$ consists of multiple metric values, where $b$ is the index of a metric. Here, $\mathcal{B}$ is a set of indices, which

---

[2]We adopt the term "metric" broadly in this work; see Appendix A for a detailed discussion.

can be partitioned into indices for numerical metrics $\mathcal{B}_{\text{num}}$ and indices for categorical metrics $\mathcal{B}_{\text{cat}}$, i.e., $\mathcal{B} = \mathcal{B}_{\text{num}} \cup \mathcal{B}_{\text{cat}}$.[3]

The core model predicts all metrics directly from the input signal:

$$\hat{Y}^i = f(\mathbf{S}^i), \tag{1}$$

where $f(\cdot)$ denotes the base prediction model and $\hat{Y}^i = \{\hat{y}_b^i\}_{b \in \mathcal{B}}$ represents the predicted metrics.[4]

The training objective for the multi-metric estimation minimizes the prediction error across all metrics using a regression ($n$-norm) and cross-entropy losses for $\mathcal{B}_{\text{num}}$ and $\mathcal{B}_{\text{cat}}$, respectively:

$$L_{\mathcal{B}}^i = L_{\mathcal{B}_{\text{num}}}^i + L_{\mathcal{B}_{\text{cat}}}^i = \sum_{b \in \mathcal{B}_{\text{num}}} ||y_b^i - \hat{y}_b^i||_n + \sum_{b' \in \mathcal{B}_{\text{cat}}} \text{CE}(y_{b'}^i, \hat{y}_{b'}^i), \tag{2}$$

where $n = 1$ in our experiments and $\text{CE}(\cdot)$ is the cross-entropy loss function.

Building on the task formulation introduced above and the challenges outlined in Sec. 1, including heterogeneous metric scales, partial supervision, and the lack of inter-metric reasoning, we propose **ARECHO**, a flexible and robust framework for multi-metric speech evaluation. ARECHO addresses these limitations through three key algorithmic components: (1) a comprehensive tokenization framework that standardizes diverse metric types into a unified representation space; (2) a dynamic classifier chain that captures inter-metric dependencies via flexible, data-driven sequencing; and (3) a two-step confidence-oriented decoding strategy that enhances prediction reliability under uncertainty. Each of these components is described in detail in the following subsections.

## 3.2 Tokenizing Everything

To address the above *Challenge I* of heterogeneity across evaluation metrics, ranging from unbounded continuous scores to discrete categorical labels, we introduce a unified tokenization framework that transforms all metric values into a shared discrete representation space. This formulation enables ARECHO to model metric prediction as a sequence generation task over tokens, allowing consistent treatment of diverse metric types and facilitating autoregressive dependency modeling.

Given a sample $(\mathbf{S}^i, Y^i)$, where $Y^i = \{y_b^i\}_{b \in \mathcal{B}}$ includes both numerical and categorical metrics, we define a set of tokenization functions $\mathcal{T} = \mathcal{T}_b$, where each $\mathcal{T}_b$ maps a ground-truth value $y_b^i$ to a discrete token $z_b^i$ from a finite vocabulary $\mathcal{V}_b$:

$$z_b^i = \mathcal{T}_b(y_b^i), \quad z_b^i \in \mathcal{V}_b, \tag{3}$$

where the total vocabulary is $\mathcal{V} = \bigcup_{b \in \mathcal{B}}(\mathcal{V}_b)$.

The full tokenized label sequence for sample $i$ becomes $Z^i = \{z_b^i\}_{b \in \mathcal{B}}$, which serves as the target sequence for autoregressive prediction. For numerical metrics (i.e., $b \in \mathcal{B}_{\text{num}}$), we apply quantization-based tokenization by partitioning the value range into uniformly or adaptively spaced bins.[5] For categorical metrics (i.e., $b \in \mathcal{B}_{\text{cat}}$), the tokenization is direct by mapping each class label to a unique token.

The inverse function $\mathcal{T}_b^{-1} : \mathcal{V}_b \rightarrow \mathbb{R}$ or $\mathcal{T}_b^{-1} : \mathcal{V}_b \rightarrow \mathcal{C}_b$ (where $\mathcal{C}_b$ is the label set for categorical metric $b \in \mathcal{B}_{\text{cat}}$) is used to reconstruct predictions:

$$\hat{y}_b^i = \mathcal{T}_b^{-1}(\hat{z}_b^i), \quad \hat{z}_b^i \in \mathcal{V}_b. \tag{4}$$

This formulation allows unified modeling across metric types, improves dependency learning, and enables flexible inference (see Sec. 3.3).

## 3.3 Dynamic Classifier Chain

To address *Challenge III: Dependency Modeling with Flexible Control*, we introduce a dynamic classifier chain architecture that models inter-metric dependencies while allowing flexible prediction

---

[3]Please refer to Appendix F for examples of metrics and their types.

[4]Unlike (Shi et al., 2025b), we omit these modalities and leave them for future work.

[5]Please refer to Appendix B for more discussion.). While Linear works well overall, Normal and Log-based schemes can be beneficial for metrics with skewed distributions (e.g., SNR, MOS).

orders. Our design is inspired by the multi-label classifier chain model (Read et al., 2011, 2021), but generalizes it to a token-level formulation suitable for autoregressive sequence modeling.

**Motivating Example.** Consider a case where we want to predict three metrics: `Gender`, `Emotion`, and `MOS`. Instead of predicting these values independently, our model constructs a target sequence:

$$\mathbf{T}_{\text{full}} = [\texttt{<Gender>}, \texttt{Male}, \texttt{<Emotion>}, \texttt{Happy}, \texttt{<MOS>}, \texttt{3.78}]$$

Here, tokens like `<Gender>` and `<MOS>` are *metadata tokens* that serve as prompts, indicating which metric the model should predict next. The following tokens, `Male`, `Happy`, `3.78`, are the corresponding *value tokens*, representing the model's predictions for each metric.[6]

**Formal Definition.** For each metric $b \in \mathcal{B}$, we define a metadata token $m_b \in \mathcal{M}$ drawn from a finite vocabulary $\mathcal{M}$. These tokens act as queries for predicting the associated values. Note that $\mathcal{M}$ is a separate set of metadata tokens and is disjoint from $\mathcal{V}$, the value token vocabulary.

The full target sequence for training is a flat, interleaved sequence of metadata and value tokens:

$$\mathbf{T}_{\text{full}}^i = [m_{b_1}, z_{b_1}^i, m_{b_2}, z_{b_2}^i, \ldots, m_{b_K}, z_{b_K}^i], \tag{5}$$

where $[b_1, \ldots, b_K]$ is a permutation of selected metrics for a given input, and $z_{b_k}^i$ is the value token for metric $b_k$ in sample $i$.

**Training Objective.** This sequence is modeled autoregressively:

$$P(\mathbf{T}_{\text{full}}^i \mid \mathbf{S}^i) = \prod_{t=0}^{2K-1} P(x_t^i \mid \mathbf{T}_{<t}^i, \mathbf{S}^i), \tag{6}$$

where $x_t^i \in \mathcal{M} \cup \mathcal{V}$ is either a metadata or value token, and $\mathbf{S}^i$ is the input speech representation.

Randomizing the metric order during training exposes the model to diverse conditioning patterns, helping it generalize across different evaluation needs.

The advantages of the use of metadata tokens include: *Metric Identification*: Metadata tokens specify which metric to predict, even when value formats overlap (e.g., many metrics return scores in similar ranges). *Dependency Control*: Prior metric predictions are part of the sequence history, enabling context-aware inference of subsequent metrics.

**Flexible Inference.** During inference, the model begins with a metadata token (e.g., `<Emotion>`) and autoregressively generates its corresponding value token (e.g., `Sad`). This procedure iterates for any subset of requested metrics, allowing dynamic chain-style reasoning in arbitrary query orders. Users can either (i) specify a fixed prediction order or (ii) rely on our two-step confidence-oriented decoding in Sec. 3.4 to automatically determine an effective sequence. Such flexibility is particularly advantageous when metric availability or user interests differ across samples.

**Support for Partial Supervision.** The autoregressive formulation also helps address *Challenge II: Limited Label Availability*. For samples with only a few known metrics, the training target simply omits the unavailable ones. For instance:

$$\mathbf{T}_{\text{partial}}^i = [m_{b_1}, z_{b_1}^i, m_{b_3}, z_{b_3}^i].$$

This allows the model to learn from partially labeled data without masking or imputation.

Compared to conventional classifier chains that assume fixed label spaces and static prediction orders (Chen et al., 2018; Gerych et al., 2021), our token-level formulation generalizes the idea for more expressive and modular metric modeling. We elaborate on how this supports efficient decoding in Section 3.4.

## 3.4 Two-step Confidence-oriented Decoding

While the dynamic classifier chain enables flexible and dependency-aware prediction of multiple metrics, it introduces challenges at inference time, particularly due to the presence of *metadata* tokens. Since the order of metric prompts is randomly optimized during training, the model's confidence in predicting metadata tokens can be unreliable, making it difficult to guide decoding based on their probabilities.

---

[6]We provide a more detailed running example in Appendix C.

To mitigate the decoding instability introduced by *metadata* tokens in the dynamic classifier chain, we adopt a *two-step confidence-oriented decoding* strategy that guides inference using the confidence of *value* tokens instead of the less reliable metadata scores.

Let $\mathcal{B}_K$ be the set of all metrics and $\hat{\mathbf{T}}_{\text{prev}}$ the current decoded prefix after $k$ metrics, corresponding to the already-predicted subset $\mathcal{B}_k \subseteq \mathcal{B}_K$. With $K - k$ metrics still to predict, the decoding for each remaining metric $b \in \mathcal{B}_K \backslash \mathcal{B}_k$ proceeds in two phases:

- **Step 1: Preliminary Prediction.** Append the metadata token $\hat{m}_b$ to the prefix,

$$\hat{\mathbf{T}} = \hat{\mathbf{T}}_{\text{prev}} + \hat{m}_b,$$

  and obtain a provisional value token $\hat{z}_b$ together with its softmax-based confidence $\text{Conf}(\hat{z}_b)$.

- **Step 2: Confidence-driven Candidate Search.** Using the intermediate prefix $\tilde{\mathbf{T}}_{\text{pre}} = \hat{\mathbf{T}}_{\text{prev}} + \hat{m}_b$, extract the $B$ most probable candidate values

$$\mathcal{Z}_b^{(B)} = \text{Top-}B\big\{P(v \mid \mathbf{S}, \tilde{\mathbf{T}}_{\text{pre}})\big\}.$$

  For every $\tilde{z}_b \in \mathcal{Z}_b^{(B)}$, form the candidate sequence

$$\tilde{\mathbf{T}} = \tilde{\mathbf{T}}_{\text{pre}} + \tilde{z}_b,$$

  compute its log-likelihood $\log P(\tilde{\mathbf{T}} \mid \mathbf{S})$, and retain the highest-scoring sequence as the final prediction for metric $b$.

After all candidates for every $b \in \mathcal{B}_K \backslash \mathcal{B}_k$ are evaluated, we keep the top-$B$ partial hypotheses (by log-likelihood) to serve as prefixes for the next metric. By revisiting low-confidence predictions through this confidence-aware beam search, the proposed strategy substantially improves the stability and accuracy of autoregressive multi-metric inference.

# 4 Experimental Setup

## 4.1 Datasets

To comprehensively evaluate the generalization ability, robustness, and versatility of ARECHO, we conduct experiments across a diverse set of datasets spanning multiple speech domains. These datasets are selected to represent key practical scenarios that require distinct types of evaluation metrics and pose varying challenges in terms of signal complexity, perceptual quality, and annotation availability. Specifically, we include (1) basic speech data to provide foundational coverage of speech variability across domains, (2) simulated corrupted speech datasets to test objective quality prediction under controlled noise conditions, (3) enhanced speech recordings to assess robustness to natural distortions and variability, and (4) synthesized speech datasets to evaluate naturalness and expressiveness of generated speech. We briefly describe each dataset below; additional details, including licensing and preprocessing procedures, are provided in Appendix E.

**Basic Speech Data.** We include data sampled from the OWSM-V3 corpus (Peng et al., 2023), a large-scale aggregation of speech recognition and translation datasets. This collection provides general-purpose coverage over a wide range of speakers, styles, and domains.

**Corrupted Speech Data (Simulation)**. We used simulated SE data generated via the URGENT2024 challenge (Zhang et al., 2024) training data generation script [7]. Furthermore, we also include, for training purposes only, the wVoice Bank+DEMAND Veaux et al. (2013); Thiemann et al. (2013) benchmark dataset for speech denoising.

**Enhanced Speech Data**. Together with the simulated data described previously, we used the blind test set from the URGENT2024 challenge. It includes both simulated and real-world recordings with speech corrupted by one or more of the distortions in the same way as mentioned above. These recordings were enhanced by various participants' submitted systems in the URGENT2024 challenge.

**Synthesized Speech Data**. We utilize two benchmark datasets: VoiceMOS 2022 Challenge (Huang et al., 2022) and NISQA (Mittag et al., 2021). VoiceMOS 2022 challenge was developed for the

---

[7]Available github.com/urgent-challenge/urgent2024_challenge

VoiceMOS Challenge 2022, comprising a diverse collection of synthetic speech generated by various TTS systems. NISQA (Mittag et al., 2021) dataset also contains synthesized speech samples with corresponding quality ratings across multiple perceptual dimensions.

Given these sources, we construct unified training, development, and test sets by carefully sampling across all domains. This curation ensures that the model is exposed to a balanced mixture of evaluation metrics and acoustic conditions, promoting generalization across tasks.

To further examine the effect of training data scale, we prepare two training configurations: a smaller `Base` set (308.77 hours) and a larger `Scale` set (2137.74 hours). Each is paired with a shared development set (18.65 hours). Evaluation is performed using four domain-specific test sets corresponding to the main application areas: simulated enhancement (4.51 hours), enhanced speech data (30.12 hours), and speech synthesis-related data (3.46 hours). Full statistical details are available in the Appendix E.5.

## 4.2 Model Setups

**Metrics in ARECHO.** To ensure broad coverage and comprehensive speech profiling, `ARECHO` incorporates a diverse set of evaluation metrics from two main sources: (1) automatically computed metrics derived from existing models or algorithms, and (2) pre-annotated information extracted from dataset metadata or human subjective evaluations.

For the first category, we employ the VERSA toolkit (Shi et al., 2025a) to estimate 47 independent metrics, 25 dependent metrics, and 7 non-matching metrics.[8] For the second category, we include 8 ground-truth metrics derived from dataset annotations, such as language labels, emotional categories, and human-annotated MOS scores. In total, ARECHO models 87 metrics, comprising 65 numerical and 22 categorical metrics.[9]

**Baseline Setup.** We adopt the `UniVERSA` model (Shi et al., 2025b) as our baseline. It uses a Transformer-based audio encoder built on WavLM representations (Chen et al., 2022) to extract shared speech embeddings. Each metric is then independently predicted using a metric-specific pooling layer followed by an X-vector-based prediction head (Snyder et al., 2018). Regression targets are predicted as scalars, whereas classification targets use a softmax output corresponding to the number of classes.

**Tokenization.** To study the effect of discrete metric modeling, we implement a variant called `UniVERSA-T`, where all numerical metrics are converted into classification tasks via uniform quantization tokenization, using the same architecture as the original `UniVERSA`.

**Proposed Model Setup.** For `ARECHO`, we retain the same audio encoder as UniVERSA but replace the prediction heads with a Transformer-based decoder that autoregressively generates the full token sequence $\mathbf{T}_{\text{full}}^i$ as defined in Sec. 3.3.

All models are trained under both the `Base` and `Scale` training configurations. Detailed architecture specifications, training procedures, and decoding hyperparameters are provided in Appendix G. Additional ablation studies exploring the effects of model components and training configurations can be found in Appendix I.

## 4.3 Evaluation

We adopt standard evaluation metrics for regression and classification tasks. For each numerical metric, we compute the mean squared error (MSE), linear correlation coefficient (LCC), and Kendall's tau (KTAU). For each categorical metric, we report accuracy (ACC) and F1 score. Evaluation scores are averaged across all numerical and categorical metrics, respectively, to provide overall regression and classification performance.[10]

---

[8]The complete configuration of VERSA to calculate related metrics is detailed in Appendix F.

[9]A complete breakdown of all supported metrics, including their types, sources, and usage, is provided in the Appendix F.

[10]In appendices, we additionally report root mean squared error (RMSE), mean absolute error (MAE), and Spearman's rank correlation coefficient (SRCC) for regression metrics; precision, and recall for classification metrics. The complete experimental result with all metrics are presented in Appendix H.

Table 1: Main experimental results for comparison between baseline and ARECHO. The "Domain" indicates the evaluation set used for the model assessment.

| Data | Domain | Model | Token | Chain | Regression Metrics | | | Classification Metrics | |
|------|--------|-------|-------|-------|-------|-------|--------|--------|-------|
| | | | | | MSE (↓) | LCC (↑) | KTAU (↑) | Acc (↑) | F1 (↑) |
| Base | Dev. | UniVERSA | ✗ | ✗ | 160.06 | 0.69 | 0.53 | 0.68 | 0.42 |
| | | UniVERSA-T | ✓ | ✗ | 40.95 | 0.78 | 0.68 | 0.70 | 0.46 |
| | | ARECHO | ✓ | ✓ | **25.73** | **0.86** | **0.72** | **0.71** | **0.51** |
| | Enhanced | UniVERSA | ✗ | ✗ | 61.54 | 0.71 | 0.54 | 0.69 | 0.43 |
| | | UniVERSA-T | ✓ | ✗ | 27.34 | 0.81 | 0.68 | 0.70 | 0.47 |
| | | ARECHO | ✓ | ✓ | **20.58** | **0.84** | **0.69** | **0.72** | **0.51** |
| | Corrupted | UniVERSA | ✗ | ✗ | 170.65 | 0.61 | 0.48 | 0.70 | 0.46 |
| | | UniVERSA-T | ✓ | ✗ | 77.72 | 0.74 | 0.67 | 0.71 | 0.50 |
| | | ARECHO | ✓ | ✓ | **44.22** | **0.82** | **0.70** | **0.72** | **0.55** |
| | Synthesized | UniVERSA | ✗ | ✗ | 58.79 | 0.76 | 0.54 | 0.69 | 0.45 |
| | | UniVERSA-T | ✓ | ✗ | 8.10 | 0.84 | 0.68 | 0.72 | 0.50 |
| | | ARECHO | ✓ | ✓ | **4.99** | **0.91** | **0.78** | **0.79** | **0.65** |
| | Avg. Test | UniVERSA | ✗ | ✗ | 96.99 | 0.69 | 0.52 | 0.69 | 0.45 |
| | | UniVERSA-T | ✓ | ✗ | 37.72 | 0.79 | 0.68 | 0.71 | 0.49 |
| | | ARECHO | ✓ | ✓ | **23.26** | **0.86** | **0.72** | **0.74** | **0.57** |
| Scale | Dev. | UniVERSA | ✗ | ✗ | 116.01 | **0.89** | 0.74 | 0.73 | 0.49 |
| | | UniVERSA-T | ✓ | ✗ | **27.98** | 0.86 | 0.75 | 0.74 | **0.52** |
| | | ARECHO | ✓ | ✓ | 29.61 | 0.86 | **0.76** | **0.75** | **0.52** |
| | Enhanced | UniVERSA | ✗ | ✗ | 43.05 | **0.84** | 0.67 | 0.72 | 0.47 |
| | | UniVERSA-T | ✓ | ✗ | 69.94 | 0.80 | 0.71 | 0.74 | 0.50 |
| | | ARECHO | ✓ | ✓ | **32.63** | 0.83 | **0.73** | **0.75** | **0.53** |
| | Corrupted | UniVERSA | ✗ | ✗ | 151.97 | **0.88** | 0.75 | 0.75 | 0.54 |
| | | UniVERSA-T | ✓ | ✗ | 39.80 | 0.77 | 0.74 | 0.76 | 0.54 |
| | | ARECHO | ✓ | ✓ | **34.37** | 0.84 | **0.76** | **0.77** | **0.56** |
| | Synthesized | UniVERSA | ✗ | ✗ | **6.46** | 0.84 | 0.65 | 0.71 | 0.47 |
| | | UniVERSA-T | ✓ | ✗ | 8.23 | 0.84 | 0.68 | 0.73 | 0.49 |
| | | ARECHO | ✓ | ✓ | 8.63 | **0.85** | **0.72** | **0.75** | **0.54** |
| | Avg. Test | UniVERSA | ✗ | ✗ | 67.16 | **0.86** | 0.70 | 0.73 | 0.50 |
| | | UniVERSA-T | ✓ | ✗ | 39.32 | 0.82 | 0.72 | 0.74 | 0.51 |
| | | ARECHO | ✓ | ✓ | **25.21** | 0.85 | **0.74** | **0.76** | **0.54** |

# 5  Experimental Results

**Overall Performance.** Table 1 summarizes the overall performance of the proposed ARECHO model compared to baselines. Across both the Base and Scale training configurations, ARECHO consistently and significantly outperforms the UniVERSA and UniVERSA-T baselines on the majority of evaluation metrics.[11] These results highlight the effectiveness of ARECHO's dynamic classifier chain and confidence-oriented decoding strategy in capturing inter-metric dependencies and improving prediction robustness. Notably, the improvements observed with UniVERSA-T over UniVERSA demonstrate the benefit of tokenizing numerical metrics, validating our unified representation approach. Building upon this, ARECHO achieves further gains by leveraging structured autoregressive modeling, which enables more informed and context-aware metric prediction.

**Effects of Data Scaling.** The Scale training set introduces greater domain imbalance, with a heavier emphasis on corrupted speech compared to other scenarios.[12] Under this condition, ARECHO delivers substantial improvements on corrupted speech evaluation, showcasing its strong modeling capacity in data-rich domains. However, the gains are comparatively smaller on synthesized and enhanced speech, suggesting that domain balance remains an important factor for achieving broad generalization.

**Observations on Baseline Behavior.** While the original UniVERSA model underperforms in most settings, it still shows relative strength in modeling fine-grained numerical metrics, particularly reflected in its high LCC scores under the Scale configuration. This indicates that tokenization, while beneficial overall, may introduce granularity loss for certain regression tasks. Slight performance degradation in UniVERSA-T and ARECHO on specific numerical metrics highlights a trade-off between discrete modeling and numerical precision, an open challenge we aim to address in future work.

---

[11]Statistical significance is evaluated via permutation testing.
[12]See Appendix E for detailed statistics on domain distribution.

**Ablation and Extended Analysis.** We conduct a set of ablations and diagnostic studies to assess robustness, efficiency, and practical behavior; full results are reported in Appendix I and J. These studies examine (i) tokenization resolution, (ii) decoding strategy, (iii) MOS-style perceptual metrics, and (iv) task-oriented metric subsets. Across all settings, ARECHO remains stable: it performs well with compact tokenizations, achieves near-optimal accuracy with simple greedy decoding, and shows strong gains on perceptual quality metrics (e.g., MOS). When trained on either task-specific metric subsets or the full union of metrics, ARECHO shows no evidence of negative transfer from "irrelevant" metrics and often improves on core targets (e.g., SRMR, SDR, human MOS). Together, these results indicate that ARECHO is both effective and adaptive: it can leverage cross-metric structure when helpful while preserving efficiency and specialization. We additionally report efficiency analysis in Appendix M, showing that ARECHO attains these benefits with substantially reduced training and inference cost relative to prior multi-metric systems.

**Further Discussion on Dependency Modeling.**
The proposed ARECHO framework learns to adaptively control the inference order of metrics through its dynamic classifier chain and two-step decoding strategy, as introduced in Sec. 3.3 and Sec. 3.4. This design enables the model to prioritize more informative or stable metrics early in the prediction sequence, providing contextual cues that improve downstream metric predictions.

Table 2 highlights how ARECHO internally discovers and exploits an ordering rationale.[13] Across different test sets, metrics related to structured annotations or acoustic scene characteristics (e.g., Q-SpeakerGender, Q-SpeechImpairment, RIR Room Size) consistently appear early in the prediction sequence. These metrics are arguably easier to estimate and provide strong prior information for subsequent metrics. In contrast, more subjective and unstable metrics, such as MOS scores from Voice-MOS and NISQA, tend to appear later in the sequence. This suggests that ARECHO defers harder or noisier predictions until more context is available from previously decoded metrics. Depending on the context, some metrics are also

Table 2: Top-5 and Bottom-5 metrics ranked by average position (Avg. Pos.) across three test sets. Please refer to Appendix F for more details about the metrics.

| Test Set | Rank | Metric Name | Avg. Pos. |
|---|---|---|---|
| **Enhanced** | Top-1 | Q-SpeakerGender | 16.50 |
| | Top-2 | Q-SpeechImpairment | 20.35 |
| | Top-3 | Q-SpeechStyle | 21.47 |
| | Btm-3 | SNR Simulation | 163.52 |
| | Btm-2 | NISQA Real MOS | 167.91 |
| | Btm-1 | VoiceMOS Real MOS | 171.58 |
| **Corrupted** | Top-1 | RIR Room Size | 1.82 |
| | Top-2 | Q-SpeechImpairment | 12.65 |
| | Top-3 | Q-SpeechDelivery | 13.15 |
| | Btm-3 | CER | 167.26 |
| | Btm-2 | NISQA Real MOS | 170.62 |
| | Btm-1 | VoiceMOS Real MOS | 171.38 |
| **Synthesized** | Top-1 | Q-Background | 12.09 |
| | Top-2 | NISQA Coloration | 27.51 |
| | Top-3 | Q-Purpose | 27.95 |
| | Btm-3 | $C_{bak}$ | 154.75 |
| | Btm-2 | SNR Simulation | 158.64 |
| | Btm-1 | CER | 161.61 |

inherently harder to predict e.g., SNR simulation for enhanced and synthesized test sets. For enhanced speech data, the noise was removed by an SE model, while for the synthesized speech, the residual noise in the text-to-speech generated speech signal is usually very low.

This emergent ordering aligns with human intuition and supports the benefit of structured inter-metric reasoning. It also allows the system to maintain robustness when certain labels are missing or unreliable, by leveraging earlier predictions to guide later stages.

As discussed in Sec. 3.3, our proposed dynamic classifier chain supports both flexible order search and inference under arbitrary query sets. In particular, it also allows for inference with a fixed, static order of metrics when such an order is used during training. To investigate how different static orderings affect the performance of ARECHO, we provide additional analyses in the Appendix L.

**Complexity and Efficiency**. While ARECHO provides several advantages as discussed above, it also maintains modest training cost and strong computational efficiency. By effectively utilizing partially labeled data, ARECHO achieves shorter training time comparable to standard multi-task learning with masking (e.g., UniVERSA).[14] Despite its sequential decoding process, ARECHO remains practical: computing the same set of metrics through their original estimators (e.g., LLM-based evaluators) is over $100\times$ slower than a single forward pass of ARECHO.

---

[13]While Table 2 presents part of the resulting orders, we further elaborate the details in Appendix K.

[14]We present a detailed analysis in Appendix M.

# 6 Conclusion

We introduce **ARECHO** (*Autoregressive Evaluation via Chain-based Hypothesis Optimization*), a versatile and interpretable framework for multi-metric speech assessment. By unifying diverse metric types through tokenization and modeling inter-metric dependencies via a dynamic classifier chain, ARECHO offers a flexible and scalable alternative to traditional parallel prediction models. Backed with our proposed two-step confidence-oriented decoding, ARECHO maintains the flexibility in decoding, which can support diverse challenging evaluation settings. Extensive experiments across corrupted, enhanced, and synthesized speech signals demonstrate that ARECHO consistently outperforms strong baselines while enabling more structured and adaptable evaluation. We believe ARECHO provides a step toward general-purpose, dependency-aware modeling for speech and potentially broader machine learning evaluation tasks.

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

# A    Terminology: Definition and use of "Metric"

In this work, we adopt the term "metric" to broadly refer to any form of meta-information that serves as a measurable characterization of a speech signal. This includes both conventional objective evaluation scores (e.g., PESQ, STOI, SNR) and more abstract attributes (e.g., emotion category, speaker identity, language, or environment type) that contribute to a comprehensive understanding of the signal.

Although some of these attributes may not be considered "metrics" in the traditional mathematical sense, such as binary tags or categorical labels, we use the term uniformly to emphasize their role in systematic evaluation and profiling. This unified terminology supports our goal of building a versatile and extensible framework that can evaluate various aspects of speech within a consistent and interpretable structure.

By adopting this general definition, we align with recent trends in universal speech evaluation (Shi et al., 2024), where a wide range of measurements are treated under a common umbrella to facilitate joint modeling, dependency reasoning, and scalable system benchmarking.

# B    Tokenization with Probability Density Functions

To accommodate heterogeneous evaluation metrics under a unified modeling interface, we transform all metric values into discrete tokens using either categorical mapping or quantization-based tokenization. This appendix elaborates on the numerical-metric tokenization strategies, explains the difference between *uniform-over-value* and *percentile-based* binning, and reports a reconstruction study comparing the two.

## B.1    Quantization-Based Tokenization for Numerical Metrics

For a numerical metric $b \in \mathcal{B}_{\text{num}}$, we partition its continuous value domain into $T$ bins, each mapped to a token in the vocabulary $\mathcal{V}_b$. Two general binning paradigms are considered:

- **Uniform-over-value binning.** The raw range $[y_{\min}, y_{\max}]$ is divided into $T$ equal-width *value* intervals. This simple strategy disregards the data distribution and can lead to highly imbalanced token frequencies when the metric values are skewed.

- **Percentile-based binning.** Instead of equal value widths, we choose $T$ equally spaced points on the empirical cumulative distribution of the training data. Each bin therefore contains approximately the same number of samples, producing balanced token priors. This strategy (our default) preserves fine detail in dense regions of the data while avoiding sparse bins.

Table 3: Quantize→dequantize reconstruction error (RMSE/MAE) on held-out test sets. Percentile-based binning consistently outperforms uniform-over-value binning across all domains. While 1000 bins improve reconstruction fidelity, they introduce more difficult prediction targets, so $T = 500$ is adopted by default.

| Domain | Tokenization | Bins | RMSE | MAE |
|---|---|---|---|---|
| Synthesized | Percentile | 500 | 0.056 | 0.014 |
| | Uniform | 500 | 0.100 | 0.025 |
| | Percentile | 1000 | 0.034 | 0.007 |
| | Uniform | 1000 | 0.097 | 0.020 |
| Enhanced | Percentile | 500 | 0.139 | 0.023 |
| | Uniform | 500 | 0.223 | 0.045 |
| | Percentile | 1000 | 0.103 | 0.013 |
| | Uniform | 1000 | 0.220 | 0.039 |
| Corrupted | Percentile | 500 | 0.261 | 0.024 |
| | Uniform | 500 | 0.512 | 0.045 |
| | Percentile | 1000 | 0.165 | 0.012 |
| | Uniform | 1000 | 0.507 | 0.040 |

Formally, for percentile-based binning we compute empirical percentiles $\{q_{p_t}\}_{t=1}^{T-1}$ with $p_t = 100t/T$ and assign each sample value

$$z_b^i = \mathcal{T}_b(y_b^i) = \text{bucket}(y_b^i; \{q_{p_t}\}), \tag{7}$$

where $\text{bucket}(\cdot)$ returns the index of the interval $\left[q_{p_{t-1}}, q_{p_t}\right)$ containing $y_b^i$. The inverse mapping $\mathcal{T}_b^{-1}$ decodes tokens to their bin centroids.

To evaluate discretization fidelity, we conducted a quantize→dequantize reconstruction experiment. Percentile boundaries were learned from the Base set described in Sec. 5, ensuring that the bin definitions reflect the empirical distribution used for model training. We then applied both the percentile-based and uniform-over-value tokenizations to three held-out test sets, *Synthesized*, *Enhanced*, and *Corrupted* speech, to examine robustness across domains. Two bin counts ($T = 500$ and $T = 1000$) were compared.

According to Table 3, percentile-based tokenization achieves markedly lower reconstruction error than uniform-over-value binning, confirming that balancing token frequencies improves discretization fidelity and generalizes well across domains. Increasing the number of bins from 500 to 1000 further reduces RMSE and MAE, yet we found that such fine granularity makes the autoregressive prediction problem harder (more classes and sparser supervision). Hence, we adopt $T = 500$ as a practical trade-off between reconstruction quality and modeling stability.

### B.2 Direct Mapping for Categorical Metrics

For categorical metrics $b \in \mathcal{B}_{\text{cat}}$, the tokenization function $\mathcal{T}_b$ maps each ground-truth label $y_b^i$ directly to a unique discrete token in a pre-defined label set $\mathcal{C}_b$:

$$z_b^i = \mathcal{T}_b(y_b^i) \in \mathcal{V}_b = \{\text{cls}_1, \text{cls}_2, \ldots, \text{cls}_{|\mathcal{C}_b|}\}. \tag{8}$$

This tokenization is lossless and preserves the label semantics in the discrete representation.

## C  Illustrative Example of Dynamic Classifier Chain

To concretely demonstrate the operation of the proposed dynamic classifier chain model, we present a running example based on three representative metrics from our evaluation framework:

- Q-Emotion: speech emotion quality
- Q-Background: background environment condition
- Q-Clarity: perceptual clarity of the spoken content

Each of these metrics corresponds to a metadata token in the model's vocabulary, denoted as $m_{\texttt{Q-Emotion}}, m_{\texttt{Q-Background}}$, and $m_{\texttt{Q-Clarity}}$, respectively. These tokens act as prompts, instructing the model on which metric to predict next.

**Training Phase**

During training, the model is exposed to sequences of interleaved metadata and value tokens. Suppose the ground-truth metric values for a sample are:

$$\texttt{Q-Emotion} = \texttt{"neutral"}, \quad \texttt{Q-Background} = \texttt{"indoor"}, \quad \texttt{Q-Clarity} = \texttt{"clear"}.$$

A possible training target sequence (after random permutation of the metric order) could be:

$$\mathbf{T} = [m_{\texttt{Q-Background}}, \texttt{"indoor"}, m_{\texttt{Q-Clarity}}, \texttt{"clear"}, m_{\texttt{Q-Emotion}}, \texttt{"neutral"}].$$

The model learns to generate this sequence autoregressively:

$$P(\mathbf{T} \mid \mathbf{S}) = \prod_{t=1}^{6} P(z_t \mid z_{<t}, \mathbf{S}),$$

where $\mathbf{S}$ is the input speech representation, and $z_t$ denotes each token in the sequence (either metadata or value).

**Inference Phase**

At inference time, the user may wish to evaluate only a subset of metrics, in any order. For instance, if the user queries $\texttt{Q-Clarity}$ first, followed by $\texttt{Q-Emotion}$, the model proceeds as:

1. Input $m_{\texttt{Q-Clarity}} \rightarrow$ predict $z_{\texttt{Q-Clarity}} = \texttt{"clear"}$
2. Input $m_{\texttt{Q-Emotion}} \rightarrow$ predict $z_{\texttt{Q-Emotion}} = \texttt{"neutral"}$

The autoregressive context includes both the input speech and all previously predicted tokens, i.e.,

$$P(z_{\texttt{Q-Emotion}} \mid m_{\texttt{Q-Clarity}}, z_{\texttt{Q-Clarity}}, m_{\texttt{Q-Emotion}}, \mathbf{S}).$$

This allows the model to leverage contextual signals from earlier metric predictions when estimating subsequent ones.

**Partially Labeled Training**

If only some metrics are annotated in the training data (e.g., only $\texttt{Q-Background}$ is known), the model can still be trained with the partial sequence:

$$\mathbf{T}_{\text{partial}} = [m_{\texttt{Q-Background}}, \texttt{"indoor"}].$$

This design obviates the need for label imputation or masking, and allows full exploitation of partially labeled datasets.

**Key Takeaways**

This example illustrates how metadata tokens serve as interpretable and flexible prompts for metric prediction. The dynamic classifier chain enables:

- **Contextual Reasoning**: Predictions are conditioned on both input speech and prior metric-value pairs.
- **Flexible Querying**: Arbitrary subsets of metrics can be queried at test time, in any order.
- **Efficient Supervision**: The model can be trained on partially labeled samples without architectural changes.

In practice, this flexibility and generalization capability makes the dynamic classifier chain particularly well-suited for large-scale, real-world evaluation scenarios where annotations may be sparse, user goals diverse, and metric interdependencies significant.

# D  Detailed Procedure for Two-step Confidence-oriented Decoding

To enhance robustness in autoregressive multi-metric inference under uncertain or ambiguous acoustic conditions, we introduce a *two-step confidence-oriented decoding* strategy. This method explicitly incorporates token-level confidence when predicting each metric's value, focusing on *value tokens* rather than the less reliable *metadata tokens*.

For a decoded prefix sequence $\hat{\mathbf{T}}_{\text{prev}}$, assume that $k$ metrics have already been predicted, forming the set $\mathcal{B}_k \subseteq \mathcal{B}_K$, where $\mathcal{B}_K$ is the complete set of $K$ metrics. For each remaining metric $b \in \mathcal{B}_K \setminus \mathcal{B}_k$, the decoding proceeds as follows.

We define the confidence of a predicted value token $z_t$ at decoding step $t$ as the maximum softmax probability over the token vocabulary:

$$\text{Conf}(z_t) = \max_{v \in \mathcal{V}} P(z_t = v \mid \mathbf{S}, \mathbf{T}_{<t}), \tag{9}$$

where $\mathbf{S}$ is the input speech signal and $\mathbf{T}_{<t}$ is the sequence of previously decoded tokens.

The decoding process is then carried out in two steps, as outlined in Algorithm 1:

- **Step 1: Preliminary Prediction.** Append the metadata token $\hat{m}_b$ corresponding to metric $b$ to the current prefix:
  $$\hat{\mathbf{T}} = \hat{\mathbf{T}}_{\text{prev}} + \hat{m}_b.$$
  Using this updated prefix, the model generates a provisional value token $\hat{z}_b$ and computes its softmax confidence.

- **Step 2: Confidence-driven Candidate Search.** To mitigate uncertainty in the preliminary prediction, we extract the top-$B$ candidate values:
  $$\mathcal{Z}_b^{(B)} = \text{Top-}B \left\{ P(v \mid \mathbf{S}, \tilde{\mathbf{T}}_{\text{pre}}) \right\}, \quad \text{where } \tilde{\mathbf{T}}_{\text{pre}} = \hat{\mathbf{T}}_{\text{prev}} + \hat{m}_b.$$
  Each candidate $\tilde{z}_b \in \mathcal{Z}_b^{(B)}$ is appended to the prefix, forming a full candidate sequence $\tilde{\mathbf{T}} = \tilde{\mathbf{T}}_{\text{pre}} + \tilde{z}_b$. We then evaluate the full-sequence log-likelihood $\log P(\tilde{\mathbf{T}} \mid \mathbf{S})$, selecting the candidate with the highest score as the final prediction for metric $b$.

After decoding all remaining metrics in $\mathcal{B}_K \setminus \mathcal{B}_k$, we retain the top-$B$ partial hypotheses for the next decoding step. This strategy ensures that low-confidence predictions can be revised through a targeted, confidence-aware re-ranking, thereby improving the accuracy and stability of metric prediction.

# E  Dataset Details

## E.1  Basic Speech Data

We use the OWSM data collection for the training, which includes a large variety of speech datasets. Specifically, in this study, we use a subset of up to 1,000 hours of the OWSM V3 data, including datasets from AISHELL-1 (Bu et al., 2017), AMI (Kraaij et al., 2005), BABEL (IARPA, 2011), and CommonVoice (Ardila et al., 2020), ranging from 48 languages. The licenses of the datasets are Apache 2.0, CC-BY 4.0, IARPA Babel License, and CC0-1.0, respectively.

## E.2  Corrupted Speech Data (Simulation)

The simulated corrupted speech data from the URGENT Challenge contains source speech from five public corpora, including DNS5 LibriVox data (Dubey et al., 2024), LibriTTS (Zen et al., 2019), CommonVoice 11.0 English portion (Ardila et al., 2020), VCTK, and WSJ (Paul & Baker, 1992). Noises are taken from the DNS5 challenge corpora (which collected them from Audioset (Gemmeke et al., 2017) and Freesound (Fonseca et al., 2017)) as well as from WHAM! (Wichern et al., 2019). Artificial reverberation uses room impulse responses from the DNS challenge that have been simulated based on the image method (Allen & Berkley, 1979; Dubey et al., 2024). We use the same configuration as described in Zhang et al. (2024): SNR ratio from $\mathcal{U}(-5, 20)$ dB, reverberation and clipping probability of 25% and bandwidth limitations sampled from $\{8, 16, 22.05, 24, 32\}$ kHz.

**Algorithm 1:** Two-step Confidence-oriented Decoding (Single Metric Case)

---

**Input:** Speech input $\mathbf{S}$; model $f$; current prefix $\hat{\mathbf{T}}_{\text{prev}}$; target metric $b$; beam width $B$

**Output:** Updated prefix $\hat{\mathbf{T}}_{\text{new}} = \hat{\mathbf{T}}_{\text{prev}} + [\hat{m}_b, \hat{z}_b]$

**Step 1: Preliminary Prediction**

Let $\hat{m}_b$ be the metadata token for metric $b$

Append $\hat{m}_b$ to prefix: $\hat{\mathbf{T}} \leftarrow \hat{\mathbf{T}}_{\text{prev}} + \hat{m}_b$

**Step 2: Confidence-driven Search**

Let prefix up to current metric: $\tilde{\mathbf{T}}_{\text{pre}} \leftarrow \hat{\mathbf{T}}_{\text{prev}} + \hat{m}_b$

Retrieve top-$B$ alternative values:

$$\mathcal{Z}_b^{(B)} \leftarrow \text{Top-}B\left\{P(v \mid \mathbf{S}, \tilde{\mathbf{T}}_{\text{pre}})\right\}$$

Initialize best score $s^* \leftarrow -\infty$, best sequence $\hat{\mathbf{T}}^* \leftarrow \hat{\mathbf{T}}$

**foreach** $\tilde{z}_b \in \mathcal{Z}_b^{(B)}$ **do**

    Construct candidate prefix: $\tilde{\mathbf{T}} \leftarrow \tilde{\mathbf{T}}_{\text{pre}} + \tilde{z}_b$

    Compute log-probability score $s \leftarrow \log P(\tilde{\mathbf{T}} \mid \mathbf{S})$

    **if** $s > s^*$ **then**

        $\hat{\mathbf{T}}^* \leftarrow \tilde{\mathbf{T}}, \quad s^* \leftarrow s$

**return** $\hat{\mathbf{T}}^*$

---

All data is sampled at 48 kHz. In total, we generated in this way 1300 hours of clean and corrupted speech pairs.

This dataset is split into 808,181 training, 3k validation, and test examples, respectively. Since these data are derived through simulation from existing datasets, the licenses of such datasets are inherited.

For additional training data we use also the Voice Bank+DEMAND Veaux et al. (2013); Thiemann et al. (2013) dataset. The Voice Bank+DEMAND Veaux et al. (2013); Thiemann et al. (2013) dataset is a widely-used benchmark for SE and noise suppression research. This dataset combines clean speech recordings from the Voice Bank corpus with noise samples from the Diverse Environments Multichannel Acoustic Noise Database (DEMAND). The dataset provides paired clean and noisy utterances and is sampled at 48 kHz. The clean speech component consists of recordings from 30 speakers (15 male, 15 female) from the Voice Bank corpus, sampled at 48 kHz. For training purposes, 28 speakers (14 male, 14 female) contribute approximately 11,572 utterances, while the test set contains around 824 utterances from 2 speakers (1 male, 1 female) not included in the training set. The noise samples from the DEMAND database, which includes 16 diverse real-world environments. For creating the noisy mixtures, the training set incorporates 10 noise types (8 from DEMAND plus 2 artificially generated noises) at four different signal-to-noise ratios (SNRs): 0, 5, 10, and 15 dB. The license of the dataset is Creative Commons Attribution 4.0 International (CC-BY 4.0).

### E.3 Enhanced Speech Data

For enhanced speech evaluation, we leverage data released as part of the Universality, Robustness, and Generalizability for EnhancemeNT (URGENT) 2024 challenge, hosted at NeurIPS 2024 (Zhang et al., 2024). The dataset includes a blind test set consisting of 1,000 noisy utterances, 500 from simulated conditions with available clean references, and 500 from real-world recordings without references. These real-world samples reflect a wide range of acoustic challenges, including environmental noise, reverberation, and device artifacts. All recordings are provided at a 16 kHz sampling rate.

The noisy inputs were processed by 114 enhancement systems (the official baseline and 113 participant submissions), yielding a total of 114,000 enhanced utterances (approximately 293 hours of speech). Human evaluation was also conducted: 300 utterances per system were rated using mean opinion scores for each of the 23 final-round systems, with an equal number drawn from simulated and real-world scenarios.

Following Shi et al. (2025b), the noisy utterances are partitioned into training, development, and test subsets in an 85:5:10 ratio. These splits correspond to approximately 249, 14, and 30 hours of speech,

respectively. Half of the data is paired with reference signals, allowing both reference-based and reference-free assessments.

This diverse and high-quality dataset enables us to rigorously assess ARECHO's ability to generalize across enhancement algorithms, acoustic conditions, and evaluation configurations in real-world SE applications.

### E.4 Synthesized Speech Dataset

For synthesized speech evaluation, we utilize two complementary datasets: the VoiceMOS 2022 Challenge dataset (Huang et al., 2022) and the NISQA dataset (Mittag et al., 2021).

The VoiceMOS 2022 Challenge dataset (Huang et al., 2022) was developed for a community-driven initiative to advance speech quality assessment research. It contains speech samples from 187 distinct text-to-speech (TTS) systems sampled at 24 kHz. Each utterance receives quality ratings from at least 8 different human listeners using a standardized 5-point scale, providing reliable mean opinion scores for training and evaluation.

The NISQA dataset (Mittag et al., 2021) offers a different perspective with its multi-dimensional quality annotations. Beyond overall MOS ratings, it provides fine-grained assessments of specific quality attributes: coloration, discontinuity, noisiness, and loudness. Sampled at 48 kHz, this dataset comprises over 14,000 audio samples created under diverse conditions, both simulated (including various codecs, packet-loss scenarios, and background noise levels) and live environments (such as mobile phone calls, Zoom, Skype, and WhatsApp communications). This comprehensive approach enables detailed analysis of quality perception factors across different communication contexts.

The licenses of the VoiceMOS 2022 challenge dataset scripts for downloading, processing, and listening test results are based on BSD 3-Clause License. The audio samples retain the licenses from their original sources. The Voice Conversion Challenge (VCC) datasets have varying licenses: VCC2016 and VCC2018 use Creative Commons Attribution 4.0 International (CC-BY 4.0), while VCC2020 audio samples are licensed under Open Data Commons Open Database License (ODbL) v1.0 with database contents under Database Contents License (DbCL) v1.0. The BLIZZARD dataset is available under a research license agreement requiring individual approval. The ESPNET-TTS dataset uses LibriVox recordings, which are in the public domain in the USA. The NISQA dataset is licensed under Creative Commons Attribution 4.0 International (CC-BY 4.0).

### E.5 Experimental Dataset Details

Here, we provide a detailed breakdown of the datasets used in our experiments. The data spans four major domains: simulated corrupted speech, enhanced speech, synthesized speech, and emotion-labeled speech. All audio is standardized to a 16 kHz sampling rate, and where applicable, samples are paired with reference signals (e.g., for simulated corrupted speech). The datasets were preprocessed and partitioned to support both multi-domain training and domain-specific evaluation.

**Training and Development Sets.**   To investigate the effect of data scale on model performance, we prepare two training configurations with varying coverage across domains. The following table summarizes the number of hours contributed by each domain to the `Base` and `Scale` training sets:

Table 4: Training data composition across domains.

| Domain | `Base` (hrs) | `Scale` (hrs) |
|---|---|---|
| Enhanced Speech | 50.00 | 100.00 |
| Simulated Corrupted Speech | 99.91 | 999.00 |
| Basic Speech Data | 97.91 | 977.80 |
| Synthesized Speech | 60.94 | 60.94 |
| **Total** | 308.77 | 2137.74 |

In both configurations, we use a shared development set of 18.65 hours for validation and early stopping.

**Test Sets.** Each domain is evaluated using a held-out test set to enable domain-specific benchmarking:

- **Simulated Corrupted Speech**: 4.51 hours
- **Enhanced Speech (URGENT2024)**: 30.12 hours
- **Synthesized Speech**: 3.46 hours

For all domains except simulated corrupted speech data, we preserve the original test split provided by the corresponding source dataset to ensure comparability with prior work. For corrupted simulated speech, where no canonical split was available, we randomly sample 3,000 utterances from the training data and reserve them for development and test purposes, respectively.

To facilitate reproducibility, we will release the detailed utterance IDs used in each split, along with the corresponding evaluation metrics used during training and testing.

# F   Metrics in ARECHO

As discussed in Sec. 5, ARECHO incorporates two types of metrics: (1) those computed using VERSA, a standardized evaluation toolkit that integrates a range of open-source expert models for speech assessment; and (2) those derived from pre-annotated information provided within the datasets. In this appendix, we first detail the VERSA configuration used to compute the various metrics, followed by a complete list of all metrics included in ARECHO modeling.

## F.1   VERSA Configuration

The VERSA configuration used for ARECHO's metric computation is shown in Listing 1. In practice, some metrics are CPU-only, while others can benefit from GPU acceleration. To optimize efficiency, we divide the configuration into two subsets, one for CPU-based metrics and another for GPU-supported metrics, and submit them as parallel jobs using SLURM to accelerate the overall evaluation process.

We apply a post-filtering step to remove NaN values from the predicted metrics. For metrics derived from Qwen2-Audio, we additionally discard any predictions that do not fall within the predefined category sets.

```
# This file contains the configuration for various universal metrics
    used in speech quality assessment for ARECHO.

# visqol metric
# -- visqol: visual quality of speech
- name: visqol
  model: default

# Word error rate with ESPnet-OWSM model
# More model_tag can be from the ESPnet huggingface https://
    huggingface.co/espnet .
# The default model is 'espnet/owsm_v3.1_ebf'.
# --lid: the nbest language tag
- name: lid
  model_tag: default
  nbest: 1

# nomad (reference-based) metric
# -- nomad: nomad reference-based model
- name: nomad
  model_cache: versa_cache/nomad_pt-models

# srmr related metrics
# -- srmr: speech-to-reverberation modulation energy ratio
- name: srmr
  n_cochlear_filters: 23
  low_freq: 125
```

```yaml
    min_cf: 4
    max_cf: 128
    fast: True
    norm: False

# Emotion similarity calculated based on emo2vec
# --emo2vec_similarity: the emotion similarity with emo2vec
- name: emo2vec_similarity

# noresqa related metrics
# -- noresqa: non-matching reference based speech quality assessment
- name: noresqa
  metric_type: 1 #0: NORESQA-score, 1: NORESQA-MOS

# pysepm related metrics
# -- pysepm_fwsegsnr: frequency-weighted segmental SNR
# -- pysepm_llr: Log likelihood ratio
# -- pysepm_wss: weighted spectral slope
# -- pysepm_cd: cepstral distance objective speech quality measure
# -- pysepm_Csig, pysepm_Cbak, pysepm_Covl: composite objective speech
#    quality
# -- pysepm_csii_high, pysepm_csii_mid, pysepm_csii_low: coherence and
#    speech intelligibility index
# -- pysepm_ncm: normalized-covariance measure
- name: pysepm

# nisqa score for speech quality assessment
#  -- nisqa_mos_pred: NISQA MOS prediction
#  -- nisqa_noi_pred: NISQA noise prediction
#  -- nisqa_dis_pred: NISQA distortion prediction
#  -- nisqa_col_pred: NISQA color prediction
#  --nisqa_loud_pred: NISQA loudness prediction
# NOTE: pretrain model can be downloaded with './tools/setup_nisqa.sh'
- name: nisqa
  nisqa_model_path: ./tools/NISQA/weights/nisqa.tar

# discrete speech metrics
# -- speech_bert: speech bert score
# -- speech_bleu: speech bleu score
# -- speech_token_distance: speech token distance score
- name: discrete_speech

# mcd f0 related metrics
#  -- mcd: mel cepstral distortion
#  -- f0_corr: f0 correlation
#  -- f0_rmse: f0 root mean square error
- name: mcd_f0
  f0min: 40
  f0max: 800
  mcep_shift: 5
  mcep_fftl: 1024
  mcep_dim: 39
  mcep_alpha: 0.466
  seq_mismatch_tolerance: 0.1
  power_threshold: -20
  dtw: false

# An overall model on MOS-bench from Sheet toolkit
# --sheet_ssqa: the mos prediction from sheet_ssqa
- name: sheet_ssqa

# pesq related metrics
# -- pesq: perceptual evaluation of speech quality
- name: pesq
```

```yaml
# stoi related metrics
# -- stoi: short-time objective intelligibility
- name: stoi

# pseudo subjective metrics
# -- utmos: UT-MOS score
# -- dnsmos: DNS-MOS score
# -- plcmos: PLC-MOS score
# -- aecmos: AEC-MOS score
- name: pseudo_mos
  predictor_types: ["utmos", "dnsmos", "plcmos", "singmos", "utmosv2"]
  predictor_args:
    utmos:
      fs: 16000
    dnsmos:
      fs: 16000
    plcmos:
      fs: 16000
    singmos:
      fs: 16000
    utmosv2:
      fs: 16000

# Word error rate with OpenAI-Whisper model
# -- whisper_wer: word error rate of openai-whisper
- name: whisper_wer
  model_tag: default
  beam_size: 1
  text_cleaner: whisper_basic

# scoreq (reference-based) metric
# -- scoreq_ref: scoreq reference-based model
- name: scoreq_ref
  data_domain: natural
  model_cache: versa_cache/scoreq_pt-models

# scoreq (non-reference-based) metric
# -- scoreq_nr: scoreq non-reference-based model
- name: scoreq_nr
  data_domain: natural
  model_cache: versa_cache/scoreq_pt-models

# Speech Enhancement-based Metrics
# model tag can be any ESPnet-SE huggingface repo
# -- se_si_snr: the SI-SNR from a reference speech enhancement model
- name: se_snr
  model_tag: default

# PAM: Prompting Audio-Language Models for Audio Quality Assessment
# https://github.com/soham97/PAM/tree/main

- name: pam
  repro: true
  cache_dir: versa_cache/pam
  io: soundfile
  # TEXT ENCODER CONFIG
  text_model: 'gpt2'
  text_len: 77
  transformer_embed_dim: 768
  freeze_text_encoder_weights: True
  # AUDIO ENCODER CONFIG
  audioenc_name: 'HTSAT'
  out_emb: 768
  sampling_rate: 44100
  duration: 7
```

```
    fmin: 50
    fmax: 8000 #14000
    n_fft: 1024 # 1028
    hop_size: 320
    mel_bins: 64
    window_size: 1024
    # PROJECTION SPACE CONFIG
    d_proj: 1024
    temperature: 0.003
    # TRAINING AND EVALUATION CONFIG
    num_classes: 527
    batch_size: 1024
    demo: False

# Speaking rate calculating
# --speaking_rate: correct matching words/character counts
- name: speaking_rate
  model_tag: default
  beam_size: 1
  text_cleaner: whisper_basic

# Audiobox Aesthetics (Unified automatic quality assessment for speech
    , music, and sound.)
- name: audiobox_aesthetics
  batch_size: 1
  cache_dir: versa_cache/audiobox

# ASR-match calculating
# --asr_match_error_rate: correct matching words/character counts
- name: asr_match
  model_tag: default
  beam_size: 1
  text_cleaner: whisper_basic

# speaker related metrics
# -- spk_similarity: speaker cosine similarity
- name: speaker
  model_tag: default

# asvspoof related metrics
# -- asvspoof_score: evaluate how the generated speech is likely to be
    classifiied by a deepfake classifier
- name: asvspoof_score

# signal related metrics
# -- sir: signal to interference ratio
# -- sar: signal to artifact ratio
# -- sdr: signal to distortion ratio
# -- ci-sdr: scale-invariant signal to distortion ratio
# -- si-snri: scale-invariant signal to noise ratio improvement
- name: signal_metric

# Metrics with Qwen2Audio
# Inference pipeline follow the qwen2_audio release https://github.com
    /QwenLM/Qwen2-Audio

# exmaple of using a customized prompt (we offer default ones)
# 1. Speaker Characteristics
- name: qwen2_audio_speaker_count

- name: qwen2_audio_speaker_gender

- name: qwen2_audio_speaker_age

# To add customized prompt, use following item:
```

```
# - name: qwen2_audio_speaker_age
#   prompt: What is the age of the speaker? Please answer in 'child',
#   '20s', '30s', '40s', '50s', '60s', '70s', 'senior'.

- name: qwen2_audio_speech_impairment

# 2. Voice Properties
- name: qwen2_audio_voice_pitch

- name: qwen2_audio_pitch_range

- name: qwen2_audio_voice_type

- name: qwen2_audio_speech_volume_level

# 3. Speech Content
- name: qwen2_audio_language

- name: qwen2_audio_speech_register

- name: qwen2_audio_vocabulary_complexity

- name: qwen2_audio_speech_purpose

# 4. Speech Delivery
- name: qwen2_audio_speech_emotion

- name: qwen2_audio_speech_clarity

- name: qwen2_audio_speech_rate

- name: qwen2_audio_speaking_style

- name: qwen2_audio_laughter_crying

# 5. Interaction Patterns
- name: qwen2_audio_overlapping_speech

# 6. Recording Environment
- name: qwen2_audio_speech_background_environment

- name: qwen2_audio_recording_quality

- name: qwen2_audio_channel_type
```

Listing 1: VERSA configuration for ARECHO.

## F.2 Metrics Information

The metrics information is detailed in Table 5, 6, 7, and 8.

Table 5 lists the 47 supported independent metrics in VERSA. We provide a summary of their key characteristics below:

- **Metric Types.** Among the 47 metrics, 25 are numerical and 22 are categorical.

- **Model Dependency.** All 47 metrics require pre-trained models for inference (*model-based*).

- **Value Ranges.** The supported numerical metrics cover a diverse range of value scales:
    - 11 metrics are bounded in $[1, 5]$ (e.g., DNSMOS, NISQA, UTMOS).
    - 4 metrics are bounded in $[1, 10]$ (Audiobox Aesthetics metrics).
    - 2 metrics are bounded in $[0, 1]$ (e.g., PAM, SpoofS).
    - 4 metrics have unbounded ranges $(-\infty, \infty)$ (e.g., SE-SI-SNR, SE-SDR).

Table 5: List of supported **independent** metrics in *VERSA*. The "Model Based" column represents metrics that need pre-trained models. The "Target Direction" column indicates which direction is desirable for each metric without being overly technical.

| No. | Name | Type | Range | Model Based | Target Direction | Reference |
|---|---|---|---|---|---|---|
| 1 | Deep Noise Suppression MOS Score of P.835 (DNSMOS P.835) | numerical | [1, 5] | ✓ | ↑ | Reddy et al. (2022) |
| 2 | Deep Noise Suppression MOS Score of P.808 (DNSMOS P.808) | numerical | [1, 5] | ✓ | ↑ | Reddy et al. (2021) |
| 3 | Speech Quality and Naturalness Assessment Coloration (NISQA-COL) | numerical | [1, 5] | ✓ | ↑ | Mittag et al. (2021) |
| 4 | Speech Quality and Naturalness Assessment Discontinuity (NISQA-DIS) | numerical | [1, 5] | ✓ | ↑ | Mittag et al. (2021) |
| 5 | Speech Quality and Naturalness Assessment Loudness (NISQA-LOUD) | numerical | [1, 5] | ✓ | ↑ | Mittag et al. (2021) |
| 6 | Speech Quality and Naturalness Assessment MOS (NISQA-MOS) | numerical | [1, 5] | ✓ | ↑ | Mittag et al. (2021) |
| 7 | Speech Quality and Naturalness Assessment Noisiness (NISQA-NOI) | numerical | [1, 5] | ✓ | ↑ | Mittag et al. (2021) |
| 8 | UTokyo-SaruLab System for VoiceMOS 2022 (UTMOS) | numerical | [1, 5] | ✓ | ↑ | Saeki et al. (2022) |
| 9 | Packet Loss Concealment-focus MOS (PLCMOS) | numerical | [1, 5] | ✓ | ↑ | Diener et al. (2023) |
| 10 | Singing voice MOS (SingMOS) | numerical | [1, 5] | ✓ | ↑ | Tang et al. (2024) |
| 11 | Subjective Speech Quality Assessment (SSQA) in SHEET Toolkit | numerical | [1, 5] | ✓ | ↑ | Huang et al. (2024b) |
| 12 | UTokyo-SaruLab System for VoiceMOS 2024 (UTMOSv2) | numerical | [1, 5] | ✓ | ↑ | Baba et al. (2024) |
| 13 | Speech Quality with Contrastive Regression (SCOREQ) wo. Ref. | numerical | [1, 5] | ✓ | ↑ | Ragano et al. (2024b) |
| 14 | Speech Enhancement-based SI-SNR (SE-SI-SNR) | numerical | (-inf, inf) | ✓ | ↑ | Zhang et al. (2024) |
| 15 | Speech Enhancement-based CI-SDR (SE-CI-SDR) | numerical | (-inf, inf) | ✓ | ↑ | Zhang et al. (2024) |
| 16 | Speech Enhancement-based SAR (SE-SAR) | numerical | (-inf, inf) | ✓ | ↑ | Zhang et al. (2024) |
| 17 | Speech Enhancement-based SDR (SE-SDR) | numerical | (-inf, inf) | ✓ | ↑ | Zhang et al. (2024) |
| 18 | Prompting Audio-Language Models (PAM) metric | numerical | [0, 1] | ✓ | ↑ | Deshmukh et al. (2024) |
| 19 | Speech-to-reverberation Modulation Energy Ratio (SRMR) | numerical | [0, inf) | ✓ | ↑ | Falk et al. (2010) |
| 20 | Speaking Word/Character Rate (SWR/SCR) | numerical | [0, inf) | ✓ | - | Radford et al. (2023) |
| 21 | Anti-spoofing Score (SpoofS) | numerical | [0, 1] | ✓ | ↑ | Jung et al. (2022) |
| 22 | Language Identification (LID) | categorical | - | ✓ | - | Peng et al. (2023) |
| 23 | Audiobox Aesthetics Content Enjoyment (AA-CE) | numerical | [1, 10] | ✓ | ↑ | Tjandra et al. (2025) |
| 24 | Audiobox Aesthetics Content Usefulness (AA-CU) | numerical | [1, 10] | ✓ | ↑ | Tjandra et al. (2025) |
| 25 | Audiobox Aesthetics Production Complexity (AA-PC) | numerical | [1, 10] | ✓ | ↑ | Tjandra et al. (2025) |
| 26 | Audiobox Aesthetics Production Quality (AA-PQ) | numerical | [1, 10] | ✓ | ↑ | Tjandra et al. (2025) |
| 27 | Qwen2 Recording Environment - Channel Type (Q-ChannelType) | categorical | - | ✓ | - | Chu et al. (2024) |
| 28 | Qwen2 Speech Content - Language (Q-Lang) | categorical | - | ✓ | - | Chu et al. (2024) |
| 29 | Qwen2 Speech Delivery - Emotional Vocalizations (Q-EmoVocalization) | categorical | - | ✓ | - | Chu et al. (2024) |
| 30 | Qwen2 Voice Properties - Pitch Range (Q-PitchRange) | categorical | - | ✓ | - | Chu et al. (2024) |
| 31 | Qwen2 Recording Environment - Quality (Q-EnvQuality) | categorical | - | ✓ | - | Chu et al. (2024) |
| 32 | Qwen2 Speaker Characteristics - Age (Q-Age) | categorical | - | ✓ | - | Chu et al. (2024) |
| 33 | Qwen2 Speaker Characteristics - Count (Q-SpeakerCount) | categorical | - | ✓ | - | Chu et al. (2024) |
| 34 | Qwen2 Speaker Characteristics - Gender (Q-Gender) | categorical | - | ✓ | - | Chu et al. (2024) |
| 35 | Qwen2 Speech Delivery - Style (Q-SpeakingStyle) | categorical | - | ✓ | - | Chu et al. (2024) |
| 36 | Qwen2 Recording Environment - Background (Q-Background) | categorical | - | ✓ | - | Chu et al. (2024) |
| 37 | Qwen2 Speech Delivery - Clarity (Q-Clarity) | categorical | - | ✓ | - | Chu et al. (2024) |
| 38 | Qwen2 Speech Delivery - Emotion (Q-Emotion) | categorical | - | ✓ | - | Chu et al. (2024) |
| 39 | Qwen2 Speaker Characteristics - Speech Impairment (Q-SpeechImpairment) | categorical | - | ✓ | - | Chu et al. (2024) |
| 40 | Qwen2 Speech Content - Purpose (Q-Purpose) | categorical | - | ✓ | - | Chu et al. (2024) |
| 41 | Qwen2 Speech Delivery - Rate (Q-SpeechRate) | categorical | - | ✓ | - | Chu et al. (2024) |
| 42 | Qwen2 Speech Content - Register (Q-ContentRegister) | categorical | - | ✓ | - | Chu et al. (2024) |
| 43 | Qwen2 Voice Properties - Volume Level (Q-VolumeLevel) | categorical | - | ✓ | - | Chu et al. (2024) |
| 44 | Qwen2 Speech Content - Vocabulary Complexity (Q-VocComplexity) | categorical | - | ✓ | - | Chu et al. (2024) |
| 45 | Qwen2 Voice Properties - Pitch (Q-Pitch) | categorical | - | ✓ | - | Chu et al. (2024) |
| 46 | Qwen2 Voice Properties - Voice Type (Q-VoiceType) | categorical | - | ✓ | - | Chu et al. (2024) |
| 47 | Predicted Text Length | numerical | [0, inf) | ✓ | - | Radford et al. (2023) |

  – 3 metrics are semi-bounded in $[0, \infty)$ (e.g., SRMR, SWR/SCR, Predicted Text Length).

- **Coverage Domains.** The metrics span a wide set of evaluation domains:

  – *Perceptual speech quality:* DNSMOS, NISQA, UTMOS, SSQA, SCOREQ.
  – *Speech enhancement:* SE-SI-SNR, SE-CI-SDR, SE-SAR, SE-SDR.
  – *Speech generation and profiling:* Audiobox Aesthetics, PAM, Predicted Text Length.
  – *Security and robustness:* SpoofS (anti-spoofing).
  – *Speech metadata analysis:* Qwen2 suite, covering 22 distinct categorical dimensions.

Several metrics are provided for overlapping domains to improve robustness and account for domain-specific modeling biases. For instance, multiple metrics exist for speech quality prediction (e.g., DNSMOS, NISQA, UTMOS, SSQA, SCOREQ). These metrics differ in terms of training data, model architecture, and target annotations, leading to varied sensitivities across distortion types and speaker/content conditions. By including multiple predictors within the same domain, VERSA enables cross-validation of quality assessments and mitigates the risk of relying on a single potentially biased estimator. This redundancy also supports ensemble or consensus-based evaluations, which are critical when deploying models across diverse real-world scenarios.

Table 6 presents 25 supported dependent metrics in VERSA. These metrics require auxiliary references, such as clean speech, transcripts, or pitch tracks, and are primarily used in settings where such ground-truth information is available (e.g., supervised evaluation of synthesis, enhancement, or recognition systems). Below we summarize their characteristics:

Table 6: List of supported **dependent** metrics in *VERSA*. The "Model Based" column represents metrics that need pre-trained models. The "Target Direction" column indicates which direction is desirable for each metric without being overly technical.

| No. | Name | Type | Range | Model Based | Target Direction | Reference |
|---|---|---|---|---|---|---|
| 1 | Mel Cepstral Distortion (MCD) | numerical | [0, inf] | ✗ | ↓ | Kubichek (1993) |
| 2 | F0 Correlation (F0-CORR) | numerical | [-1, 1] | ✗ | ↑ | Hayashi et al. (2021) |
| 3 | F0 Root Mean Square Error (F0-RMSE) | numerical | [0, inf] | ✗ | ↓ | Hayashi et al. (2021) |
| 4 | Signal-to-artifact Ratio (SAR) | numerical | (-inf, inf) | ✗ | ↑ | Févotte et al. (2005) |
| 5 | Signal-to-distortion Ratio (SDR) | numerical | (-inf, inf) | ✗ | ↑ | Févotte et al. (2005) |
| 6 | Perceptual Evaluation of Speech Quality (PESQ) | numerical | [1, 5] | ✓ | ↑ | Rix et al. (2001) |
| 7 | Short-Time Objective Intelligibility (STOI) | numerical | [0, 1] | ✗ | ↑ | Taal et al. (2011) |
| 8 | Speech BERT Score (D-BERT) | numerical | [-1, 1] | ✓ | ↑ | Saeki et al. (2024) |
| 9 | Discrete Speech BLEU Score (D-BLEU) | numerical | [0, 1] | ✓ | ↑ | Saeki et al. (2024) |
| 10 | Discrete Speech Token Edit Distance (D-Distance) | numerical | [0, 1] | ✓ | ↑ | Saeki et al. (2024) |
| 11 | Speech Quality with Contrastive Regression (SCOREQ) w. Ref. | numerical | [1, 5] | ✓ | ↑ | Ragano et al. (2024b) |
| 12 | ASR-oriented Mismatch Error Rate (ASR-Mismatch) | numerical | [0, inf] | ✓ | ↓ | Radford et al. (2023) |
| 13 | Virtual Speech Quality Objective Listener (VISQOL) | numerical | [1,5] | ✓ | ↑ | Chinen et al. (2020) |
| 14 | Frequency-Weighted SEGmental SNR (FWSEGSNR) | numerical | (-inf, inf) | ✗ | ↑ | Tribolet et al. (1978) |
| 15 | Weighted Spectral Slope (WSS) | numerical | [0, inf) | ✗ | ↓ | Klatt (1982) |
| 16 | Cepstrum Distance (CD) | numerical | [0, inf) | ✗ | ↓ | Barnwell III (1979) |
| 17 | Composite Objective Speech Quality - Signal (Csig) | numerical | [1, 5] | ✓ | ↑ | Hu & Loizou (2007a) |
| 18 | Composite Objective Speech Quality - Background (Cbak) | numerical | [1, 5] | ✓ | ↑ | Hu & Loizou (2007a) |
| 19 | Composite Objective Speech Quality - Overall (Covl) | numerical | [1, 5] | ✓ | ↑ | Hu & Loizou (2007a) |
| 20 | Coherence and Speech Intelligibility Index - High (CSII-HIGH) | numerical | [0, 1] | ✗ | ↑ | Kates & Arehart (2005) |
| 21 | Coherence and Speech Intelligibility Index - Low (CSII-LOW) | numerical | [0, 1] | ✗ | ↑ | Kates & Arehart (2005) |
| 22 | Coherence and Speech Intelligibility Index - Mid (CSII-MID) | numerical | [0, 1] | ✗ | ↑ | Kates & Arehart (2005) |
| 23 | Normalized-Covariance Measure (NCM) | numerical | [-1, 1] | ✗ | ↑ | Chen & Loizou (2010) |
| 24 | Convolutive-invariant Speech-to-distortion Ratio (CI-SDR) | numerical | (-inf, inf) | ✗ | ↑ | Boeddeker et al. (2021) |
| 25 | Scale-invariant Speech-to-noise Ratio (SI-SNR) | numerical | (-inf, inf) | ✗ | ↑ | Boeddeker et al. (2021) |

- **Model Dependency.** Among the 25 dependent metrics, 10 are model-based (e.g., PESQ, VISQOL, D-BERT, D-BLEU), while the remaining 15 are traditional signal-based or statistical metrics that operate without pretrained models.

- **Target Direction.**
  - 21 metrics have ↑ as the preferred direction, indicating better performance with higher values.
  - 4 metrics are better when minimized (↓), including Mel Cepstral Distortion (MCD), F0-RMSE, Weighted Spectral Slope (WSS), and Cepstrum Distance (CD).

- **Value Ranges.** The dependent metrics span various value scales:
  - 8 metrics are bounded within fixed ranges, such as $[0, 1]$ (e.g., STOI, CSII variants, D-BLEU).
  - 6 metrics use perceptual MOS-like scales (e.g., PESQ, Csig, Covl) within $[1, 5]$.
  - 3 metrics span $[-1, 1]$ (e.g., F0-CORR, D-BERT, NCM).
  - Several metrics are unbounded or semi-bounded in $[0, \infty)$ or $(-\infty, \infty)$, particularly SNR- and SDR-based metrics.

- **Metric Coverage and Redundancy.** The dependent metrics are intentionally diverse to capture distinct aspects of speech fidelity and intelligibility. For instance:
  - *Pitch-aware metrics:* MCD, F0-CORR, and F0-RMSE quantify pitch and spectral similarity.
  - *SNR/SDR variants:* Multiple versions (SAR, SDR, SI-SNR, CI-SDR, FWSEGSNR) are included to assess distortions under different assumptions (e.g., scale- or convolution-invariance).
  - *Perceptual quality:* PESQ, VISQOL, Csig/Cbak/Covl provide subjective approximations of human ratings.
  - *Token-based semantic fidelity:* D-BERT, D-BLEU, and D-Distance evaluate similarity in discrete or latent representation spaces.

Including multiple metrics within the same subdomain (e.g., both PESQ and VISQOL for perceptual quality, or both SDR and SI-SNR for distortion) enhances robustness and allows for comprehensive evaluation across system types and data conditions. This redundancy is particularly important when no single metric reliably aligns with human perception in all scenarios.

Table 7: List of supported **non-matching** metrics in *VERSA*. The "Model Based" column represents metrics that need pre-trained models. The "Target Direction" column indicates which direction is desirable for each metric without being overly technical.

| No. | Name | Type | Range | Model Based | Target Direction | Reference |
|---|---|---|---|---|---|---|
| 1 | Non-matching Reference Speech Quality Assessment (Noresqa) | numerical | [1, 5] | ✓ | ↑ | Manocha et al. (2021) |
| 2 | OpenAI Whisper Model Word Error Rate (WER) | numerical | [0, inf] | ✓ | ↓ | Radford et al. (2023) |
| 3 | OpenAI Whisper Model Character Error Rate (CER) | numerical | [0, inf] | ✓ | ↓ | Radford et al. (2023) |
| 4 | Emotion Similarity (EMO-SIM) | numerical | [-1, 1] | ✓ | ↑ | Ma et al. (2024) |
| 5 | Speaker Similarity (SPK-SIM) | numerical | [-1, 1] | ✓ | ↑ | Jung et al. (2024) |
| 6 | Non-Matching Reference Audio Quality Assessment (NOMAD) | numerical | [1, 5] | ✓ | ↑ | Ragano et al. (2024a) |
| 7 | Log Likelihood Ratio (LLR) | numerical | [0, inf] | ✗ | ↑ | Hu & Loizou (2007a) |

Table 7 summarizes the 7 supported non-matching metrics in VERSA. These metrics are designed to operate in scenarios where the reference is not a direct ground-truth pair (e.g., a sample from the same speaker, emotion class, or general quality distribution), enabling broader evaluation capabilities such as speaker consistency or semantic fidelity under more relaxed constraints.

- **Model Dependency.** All but one metric (LLR) rely on pre-trained models for their prediction. These include advanced encoders for speaker, emotion, or language content, reflecting recent trends toward model-based alignment and comparison.

- **Target Direction.**
  - 5 metrics prefer higher values (↑), including Noresqa, NOMAD, and similarity-based measures like EMO-SIM and SPK-SIM.
  - 2 metrics, Whisper-based WER and CER, are evaluated with lower-is-better semantics (↓), indicating reduced transcription error.

- **Value Ranges.** These metrics span various scales:
  - 3 metrics are bounded in $[1, 5]$ (Noresqa, NOMAD, LLR).
  - 2 metrics range from $[-1, 1]$ (EMO-SIM, SPK-SIM), reflecting cosine similarity scales.
  - 2 metrics are semi-unbounded in $[0, \infty)$ (WER, CER).

- **Use Case Rationale.** Non-matching metrics are particularly valuable in settings where exact pairwise references are unavailable or inappropriate. For example:
  - *Speaker and emotion similarity* (SPK-SIM, EMO-SIM) assess style preservation or consistency across generated outputs, even if reference content is not identical.
  - *Quality predictors* such as Noresqa and NOMAD estimate subjective quality with a flexible reference that may differ in content or length.
  - *Whisper-based WER/CER* offer a standardized ASR-oriented evaluation interface without needing paired transcriptions from a target reference set.

Together, these metrics expand the evaluation scope beyond classical paired setups, allowing for more generalizable and accessible assessments across real-world conditions.

Table 8 summarizes the 8 supported ground-truth metrics in VERSA. These metrics represent either directly observed attributes (e.g., simulation configurations, annotation-based ratings) or oracle-level information that serves as the ultimate reference for evaluating predictive models.

- **Metric Types.** The set includes both numerical and categorical metrics:
  - 5 metrics are numerical (e.g., RT60, SNR, MOS scores).
  - 3 metrics are categorical (e.g., language identity, room size).

- **Target Direction.** Most ground-truth metrics do not define a direction of improvement, as they serve as reference labels rather than performance indicators. However, 3 numerical metrics (SNR Simulation, URGENT MOS, VoiceMOS Real MOS, and NISQA Real MOS) are directionally desirable with higher values (↑), indicating better signal quality or subjective perception.

- **Value Ranges.**
  - 3 metrics use a MOS-like scale bounded in $[1, 5]$ (VoiceMOS, URGENT MOS, NISQA).

Table 8: List of supported **ground-truth** metrics in *VERSA*. The "Model Based" column represents metrics that need pre-trained models. The "Target Direction" column indicates which direction is desirable for each metric without being overly technical.

| No. | Name | Type | Range | Target Direction | Reference |
|-----|------|------|-------|------------------|-----------|
| 1 | Real Language | categorical | - | - | - |
| 2 | Reference Text Length | numerical | [0, inf) | - | - |
| 3 | RIR Room Size | categorical | - | - | - |
| 4 | RT60 | numerical | [0, inf) | - | - |
| 5 | SNR Simulation | numerical | (-inf, inf) | ↑ | - |
| 6 | URGENT MOS | numerical | [1, 5] | ↑ | Zhang et al. (2024) |
| 7 | VoiceMOS Real MOS | numerical | [1, 5] | ↑ | Huang et al. (2024c) |
| 8 | NISQA Real MOS | numerical | [1, 5] | ↑ | Mittag et al. (2021) |

- RT60 and Reference Text Length are semi-bounded in $[0, \infty)$.
- SNR Simulation is unbounded in $(-\infty, \infty)$, representing real-world variability in noise conditions.

- **Use Case and Role.** Ground-truth metrics serve three primary purposes:
  - *Evaluation targets:* MOS ratings (e.g., VoiceMOS, NISQA, URGENT) are used to supervise or validate quality prediction models.
  - *Auxiliary context:* Variables such as RT60 or SNR are useful for conditioning or interpreting model behavior in specific acoustic environments.
  - *Oracle supervision:* Some categorical features like language or room size are ground-truth labels used in training or stratified evaluation.

These metrics are typically unavailable in fully automatic pipelines but are crucial during dataset construction, model validation, and controlled benchmarking.

## F.3 Metric Coverage

Due to failure metrics' calculation, missing reference information, or missing annotation, it is common to have a incomplete metric set for each sample. To this end, we provide the coverage of each metrics for the `Base` and `Scale` set in Table 9, which helps to understand the metric imbalance issue existing in the multi-metric estimation.

# G   Detailed Experimental Setup

## G.1   Model Architecture

**Baseline**. We adopt the UniVERSA architecture with support for both numerical and categorical metrics. The model leverages a pretrained WavLM-Large encoder (Chen et al., 2022) as the audio frontend, extracted via the S3PRL interface with multilayer features enabled (wen Yang et al., 2021). To retain pretrained knowledge, we freeze all parameters in the upstream encoder during training. The frontend outputs are passed to a Transformer-based audio encoder composed of 4 layers with 4 attention heads per layer, a hidden dimension of 1024, and dropout regularization applied at various levels (general: 0.1, attention: 0.1, positional: 0.1). The encoder uses a convolutional input layer and applies layer normalization before self-attention blocks, adopting linear position-wise layers and a lightweight kernel size of 1.

Mean pooling is used to aggregate encoded sequences, followed by a metric-specific projection head implemented as an X-vector-based prediction head. The model set the prediction head for each metric individually to accommodate multiple simultaneous prediction heads (e.g., for different metrics). The total parameter size for the baseline UniVERSA is 604.38M with 288.93M learnable parameters.

**Tokenization**. We use a token size of 500 for the default numerical tokenization. **Linear** tokenization is applied to all numerical metrics.

**ARECHO**. For the ARECHO model, we use the same audio frontend and encoder as the UniVERSA model. The final metric decoder is a Transformer-based module designed to handle the diverse

Table 9: Metrics with Percentage of Occurrences in the `Base` and `Scale` training sets.

Base Training Set

| Occ. (%) | Metrics |
|---|---|
| 98.49 | Q-PitchRange, Q-Background, Q-VoiceType, Q-EmoVocalization, Q-Gender, Q-SpeakerCount, Q-SpeakingStyle, Q-Emotion, Q-Pitch, Q-Purpose, Q-VolumeLevel, Q-EnvQuality, Q-SpeechImpariment, Q-Age, Q-VocComplexity, Q-Clarity, Q-ContentRegister, Q-SpeechRate, Q-ChannelType |
| 98.41 | Q-Lang |
| 86.55 | SE-CI-SDR, LID, PAM, NISQA-MOS, UTMOS, SingMOS, SCOREQ, AA-PC, Real Language, NISQA-NOI, NISQA-LOUD, SE-SAR, NISQA-DIS, SWR/SCR, PLCMOS, NISQA-COL, DNSMOS P.835, SpoofS, UTMOSv2, SE-SDR, AA-CU, AA-CE, DNS-MOS P.808, AA-PQ, SSQA |
| 86.54 | SE-SI-SNR |
| 80.44 | SRMR |
| 38.44 | RIR Room Size, SNR Simulation |
| 35.65 | SPK-SIM, D-BLEU, D-Distance, D-BERT |
| 29.78 | Reference Text Length, ASR-Mismatch, NOMAD, EMO-SIM, Noresqa, SCOREQ w. Ref., Predicted Text Length |
| 26.79 | STOI, F0RMSE, MCD, SAR, PESQ, SDR, CI-SDR |
| 26.78 | F0Corr, SI-SNR |
| 20.92 | Cbak, FWSEGSNR, LLR, Covl, WSS, VISQOL, Csig, CD, NCM |
| 20.91 | CSII-HIGH, CSII-MID |
| 20.24 | CSII-LOW |
| 19.18 | RT60 |
| 13.22 | NISQA Real MOS |
| 11.46 | WER, CER |
| 2.93 | VoiceMOS Real MOS |
| 0.65 | URGENT MOS |

Scale Training Set

| Occ. (%) | Metrics |
|---|---|
| 98.24 | Q-SpeakingStyle, Q-VoicePitch, Q-SpeechRate, Q-ChannelType, Q-Gender, Q-EmoVocalization, Q-Emotion, Q-Age, Q-SpeakerCount, Q-Clarity, Q-VocComplexity, Q-ContentRegister, Q-Purpose, Q-SpeechImpariment, Q-PitchRange, Q-VoiceType, Q-VolumeLevel, Q-Background, Q-EnvQuality |
| 98.16 | Q-Lang |
| 83.28 | UTMOSv2, DNSMOS P.835, Real Language, NISQA-DIS, SpoofS, NISQA-LOUD, AA-PQ, AA-CE, AA-CU, SingMOS, NISQA-COL, NISQA-MOS, PAM, SSQA, SpeakingRate, SE-SAR, AA-PC, SE-SDR, SE-CI-SDR, SCOREQ, NISQA-NOI, UTMOS, DNSMOS P.808, PLCMOS |
| 83.27 | SE-SI-SNR |
| 75.17 | SRMR |
| 50.45 | SNR Simulation, RIR Room Size |
| 40.63 | D-BLEU, D-BERT, SPK-SIM, D-Distance |
| 39.11 | NOMAD, Reference Text Length, EMO-SIM, Noresqa, ASR-Mismatch, SCOREQ w. Ref., Predicted Text Length |
| 29.04 | F0RMSE, PESQ, STOI, SAR, CI-SDR, MCD, SDR, F0Corr, SI-SNR |
| 27.51 | Cbak, Covl, LLR, WSS, CD, VISQOL, FWSEGSNR, Csig, NCM |
| 27.50 | CSII-MID, CSII-HIGH |
| 26.61 | CSII-LOW |
| 25.19 | RT60 |
| 3.01 | WER, CER |
| 1.74 | NISQA Real MOS |
| 1.52 | SIR |
| 0.38 | VoiceMOS Real MOS |
| 0.16 | URGENT MOS |

space of evaluation targets. It comprises 4 self-attention blocks, each with 4 attention heads and 1024-dimensional feedforward layers. Regularization is applied with dropout rates of 0.1 for general, positional, source-attention, and self-attention components. The decoder adopts an embedding-based input layer and applies layer normalization before each attention block. Similar to the encoder, it uses no concatenation after self-attention and supports stochastic layer dropping with a rate of 0.1.

To enhance the modeling of metric sequences, rotary position embeddings (RoPE) are enabled (Heo et al., 2024), allowing better generalization to longer sequences. The decoder is trained with label smoothing to improve generalization and reduce overconfidence. Standard start-of-sequence and end-of-sequence tokens (`<sos>`, `<eos>`) are used for autoregressive decoding when applicable. The total parameter size of the proposed ARECHO model is 581.21M with 265.76M learnable parameters.

## G.2 Training and Decoding Setup

All models in our experiments use Xavier uniform initialization and optimize using the AdamW optimizer with a learning rate of 0.001, scheduled via a warm-up mechanism over 25k steps. Training is conducted with a gradient accumulation of 2 and a batch size of 16, sorted by descending sequence length for efficiency. All the experiments are trained with GH200 for up to 5 days, with 100 epochs at maximum. It is worth noting that ARECHO uses significantly less GPU memory (50GB vs. 85GB) than UniVERSA in a similar parameter size.

During decoding, we use a beam size of 1 to conduct a greedy search if not specified.

Table 10: Main experimental results with complete evaluation metrics for comparison between baseline and ARECHO. The "Domain" indicates the evaluation set used for the model assessment.

| Data | Domain | Model | Token | Chain | Regression Metrics | | | | | | | Classification Metrics | | | |
|---|---|---|---|---|---|---|---|---|---|---|---|---|---|---|---|
| | | | | | MSE (↓) | RMSE (↓) | MAE (↓) | BMAE (↓) | LCC (↑) | SRCC (↑) | KTAU (↑) | Acc (↑) | Precision (↑) | Recall (↑) | F1 (↑) |
| Base | Dev. | UniVERSA | ✗ | ✗ | 160.06 | 5.17 | 4.13 | 5.08 | 0.69 | 0.68 | 0.53 | 0.68 | 0.43 | 0.47 | 0.42 |
| | | UniVERSA-T | ✓ | ✗ | 40.95 | 2.70 | 1.62 | 1.96 | 0.78 | 0.82 | 0.68 | 0.70 | 0.49 | 0.50 | 0.46 |
| | | ARECHO | ✓ | ✓ | **25.73** | **2.16** | **1.27** | **1.51** | **0.86** | **0.86** | **0.72** | **0.71** | **0.52** | **0.53** | **0.51** |
| | Enhanced | UniVERSA | ✗ | ✗ | 61.54 | 4.22 | 3.48 | 3.61 | 0.71 | 0.71 | 0.54 | 0.69 | 0.43 | 0.48 | 0.43 |
| | | UniVERSA-T | ✓ | ✗ | 27.34 | 2.65 | 1.60 | 1.75 | 0.81 | 0.84 | 0.68 | 0.70 | 0.49 | 0.51 | 0.47 |
| | | ARECHO | ✓ | ✓ | **20.58** | **2.09** | **1.32** | **1.43** | **0.84** | **0.85** | **0.69** | **0.72** | **0.52** | **0.54** | **0.51** |
| | Corrupted | UniVERSA | ✗ | ✗ | 170.65 | 4.84 | 3.74 | 4.79 | 0.61 | 0.63 | 0.48 | 0.70 | 0.47 | 0.50 | 0.46 |
| | | UniVERSA-T | ✓ | ✗ | 77.72 | 2.91 | 1.70 | 2.02 | 0.74 | 0.81 | 0.67 | 0.71 | 0.52 | 0.53 | 0.50 |
| | | ARECHO | ✓ | ✓ | **44.22** | **2.37** | **1.29** | **1.52** | **0.82** | **0.84** | **0.70** | **0.72** | **0.56** | **0.56** | **0.55** |
| | Synthesized | UniVERSA | ✗ | ✗ | 58.79 | 3.82 | 3.29 | 3.36 | 0.76 | 0.73 | 0.54 | 0.69 | 0.45 | 0.49 | 0.45 |
| | | UniVERSA-T | ✓ | ✗ | 8.10 | 1.52 | 0.91 | 0.97 | 0.84 | 0.83 | 0.68 | 0.72 | 0.52 | 0.53 | 0.50 |
| | | ARECHO | ✓ | ✓ | **4.99** | **1.13** | **0.58** | **0.61** | **0.91** | **0.91** | **0.78** | **0.79** | **0.67** | **0.66** | **0.65** |
| | Avg. Test | UniVERSA | ✗ | ✗ | 96.99 | 4.29 | 3.50 | 4.21 | 0.69 | 0.69 | 0.52 | 0.69 | 0.45 | 0.49 | 0.45 |
| | | UniVERSA-T | ✓ | ✗ | 37.72 | 2.36 | 1.40 | 1.68 | 0.79 | 0.83 | 0.68 | 0.71 | 0.51 | 0.52 | 0.49 |
| | | ARECHO | ✓ | ✓ | **23.26** | **1.86** | **1.06** | **1.27** | **0.86** | **0.87** | **0.72** | **0.74** | **0.58** | **0.59** | **0.57** |
| Scale | Dev. | UniVERSA | ✗ | ✗ | 116.01 | 3.54 | 2.28 | 4.02 | **0.89** | **0.89** | 0.74 | 0.73 | 0.52 | 0.52 | 0.49 |
| | | UniVERSA-T | ✓ | ✗ | **27.98** | **2.39** | **1.22** | **1.43** | 0.86 | 0.86 | 0.75 | 0.74 | **0.54** | **0.54** | **0.52** |
| | | ARECHO | ✓ | ✓ | 29.61 | 2.49 | 1.32 | 1.55 | 0.86 | 0.87 | **0.76** | **0.75** | **0.54** | **0.54** | **0.52** |
| | Enhanced | UniVERSA | ✗ | ✗ | 43.05 | 2.53 | 1.81 | 1.93 | **0.84** | 0.84 | 0.67 | 0.72 | 0.49 | 0.51 | 0.47 |
| | | UniVERSA-T | ✓ | ✗ | 69.94 | 3.99 | 2.17 | 2.33 | 0.80 | 0.86 | 0.71 | 0.74 | 0.53 | 0.53 | 0.50 |
| | | ARECHO | ✓ | ✓ | **32.63** | **2.86** | **1.53** | **1.67** | 0.83 | **0.87** | **0.73** | **0.75** | **0.56** | **0.55** | **0.53** |
| | Corrupted | UniVERSA | ✗ | ✗ | 151.97 | 3.69 | 2.26 | 4.11 | **0.88** | **0.89** | 0.75 | 0.75 | 0.57 | 0.57 | 0.54 |
| | | UniVERSA-T | ✓ | ✗ | 39.80 | 2.42 | 1.18 | 1.37 | 0.77 | 0.86 | 0.74 | 0.76 | 0.58 | 0.57 | 0.54 |
| | | ARECHO | ✓ | ✓ | **34.37** | **2.32** | **1.10** | **1.35** | 0.84 | 0.87 | **0.76** | **0.77** | **0.59** | **0.58** | **0.56** |
| | Synthesized | UniVERSA | ✗ | ✗ | **6.46** | **1.49** | 1.00 | 1.05 | 0.84 | 0.82 | 0.65 | 0.71 | 0.48 | 0.51 | 0.47 |
| | | UniVERSA-T | ✓ | ✗ | 8.23 | **1.49** | 0.94 | 0.99 | 0.84 | 0.83 | 0.68 | 0.73 | 0.50 | 0.52 | 0.49 |
| | | ARECHO | ✓ | ✓ | 8.63 | **1.49** | **0.90** | **0.94** | **0.85** | **0.85** | **0.72** | **0.75** | **0.56** | **0.55** | **0.54** |
| | Avg. Test | UniVERSA | ✗ | ✗ | 67.16 | 2.57 | 1.69 | 2.78 | **0.86** | 0.86 | 0.70 | 0.73 | 0.51 | 0.53 | 0.50 |
| | | UniVERSA-T | ✓ | ✗ | 39.32 | 2.63 | 1.43 | 1.53 | 0.82 | 0.86 | 0.72 | 0.74 | 0.53 | 0.54 | 0.51 |
| | | ARECHO | ✓ | ✓ | **25.21** | **2.22** | **1.18** | **1.38** | 0.85 | **0.87** | **0.74** | **0.76** | **0.57** | **0.56** | **0.54** |

# H  Complete Main Experimental Table

To provide a more comprehensive assessment of model performance, we expand our evaluation with additional regression metrics (i.e., *Root Mean Squared Error* (RMSE), *Mean Absolute Error* (MAE), *Balanced Mean Absolute Error* (BMAE) in Baccianella et al. (2009) and *Spearman's Rank Correlation Coefficient* (SRCC)) as well as classification metrics (i.e., *Precision* and *Recall*) complementing the original metrics presented in Table 10.

Across all domains and data conditions, ARECHO consistently outperforms both baseline models (UniVERSA and UniVERSA-T) in the majority of regression and classification metrics. Specifically, the lower RMSE and MAE values indicate that ARECHO yields more stable and accurate point-wise predictions, with a marked reduction in prediction variance and absolute deviation. For instance, on the Base-AvgTest condition, ARECHO achieves an RMSE of 1.86 and MAE of 1.06, compared to 2.36/1.40 for UniVERSA-T and 4.29/3.50 for UniVERSA, respectively.

Furthermore, the improvement in SRCC suggests stronger alignment with ranking orders. In both Base and Scale conditions, ARECHO attains the highest SRCC values, especially in acoustically challenging domains such as Corrupted and Synthesized, where robustness to distortions is crucial. These results reflect the model's enhanced capability to preserve ordinal consistency across perceptual quality metrics.

From a classification perspective, ARECHO demonstrates consistent superiority in F1 scores, primarily due to a balanced improvement in both precision and recall. For example, in the Scale-Corrupted domain, ARECHO achieves an F1 score of 0.56, with a precision of 0.59 and recall of 0.58, which is substantially higher than the corresponding metrics from baseline models. This indicates better discriminative performance in multi-metric classification, which is critical for downstream applications such as quality control and speech diagnostics.

Taken together, the augmented evaluation substantiates that the proposed chain-based approach not only improves overall prediction accuracy but also enhances ranking reliability and classification robustness across diverse testing scenarios.

Table 11: Ablation study: the effect of token size for ARECHO.

| Domain | Model | Token Bins | Regression Metrics | | | | | | Classification Metrics | | | |
|---|---|---|---|---|---|---|---|---|---|---|---|---|
| | | | MSE (↓) | RMSE (↓) | MAE (↓) | LCC (↑) | SRCC (↑) | KTAU (↑) | Acc (↑) | Precision (↑) | Recall (↑) | F1 (↑) |
| Dev. | UniVERSA-T | 500 | 27.98 | **2.39** | **1.22** | **0.86** | **0.86** | 0.75 | 0.74 | 0.54 | **0.54** | **0.52** |
| | | 1,000 | 50.46 | 3.21 | 1.63 | 0.82 | 0.84 | 0.72 | 0.73 | 0.53 | 0.52 | 0.50 |
| | ARECHO | 500 | **29.61** | 2.49 | 1.32 | 0.85 | **0.86** | **0.76** | **0.75** | 0.54 | **0.54** | **0.52** |
| | | 1,000 | 30.86 | 2.53 | 1.33 | 0.84 | 0.85 | 0.74 | 0.74 | **0.55** | **0.54** | 0.51 |
| Enhanced | UniVERSA-T | 500 | 69.94 | 3.99 | 2.17 | 0.80 | 0.86 | 0.71 | 0.74 | 0.53 | 0.53 | 0.50 |
| | | 1,000 | 59.33 | 3.80 | 2.07 | 0.81 | 0.85 | 0.70 | 0.73 | 0.53 | 0.53 | 0.50 |
| | ARECHO | 500 | **32.63** | **2.86** | **1.53** | **0.83** | **0.87** | **0.73** | **0.75** | **0.56** | **0.55** | **0.53** |
| | | 1,000 | 43.32 | 3.17 | 1.70 | 0.81 | 0.85 | 0.72 | **0.75** | 0.54 | 0.54 | 0.52 |
| Corrupted | UniVERSA-T | 500 | 39.80 | 2.42 | 1.18 | 0.77 | 0.86 | 0.74 | 0.76 | 0.58 | 0.57 | 0.54 |
| | | 1,000 | 60.73 | 2.66 | 1.29 | 0.82 | 0.85 | 0.73 | 0.75 | 0.57 | 0.56 | 0.54 |
| | ARECHO | 500 | 34.37 | 2.32 | 1.10 | **0.84** | **0.87** | **0.76** | **0.77** | **0.59** | **0.58** | **0.56** |
| | | 1,000 | **31.68** | **2.02** | **1.07** | **0.84** | 0.85 | 0.74 | 0.76 | **0.59** | **0.58** | **0.56** |
| Synthesized | UniVERSA-T | 500 | 8.23 | 1.49 | **0.94** | 0.84 | 0.83 | 0.68 | 0.73 | 0.50 | 0.52 | 0.49 |
| | | 1,000 | 8.50 | 1.53 | 0.92 | 0.83 | 0.83 | 0.65 | 0.73 | 0.51 | 0.52 | 0.49 |
| | ARECHO | 500 | 8.63 | 1.49 | 0.90 | **0.85** | **0.85** | **0.72** | **0.75** | 0.56 | **0.55** | **0.54** |
| | | 1,000 | **7.31** | **1.41** | 0.83 | **0.85** | **0.85** | 0.71 | **0.75** | **0.57** | **0.55** | 0.53 |

# I  Ablation Experiments and Model Analysis

## I.1  The Effect of Token Size

Previous experiments utilized a default token size of 500, which represents a balance between computational efficiency and model performance. However, to gain a more comprehensive understanding of the model's capacity to handle longer contexts, we extend our experiments to include a token size of 1,000.

Table 11 presents the results of this ablation study across different domains, comparing both models at token sizes of 500 and 1,000. Several interesting patterns emerge from this analysis. First, we observe that the effect of increasing token size is not uniform across domains and models, suggesting a complex interaction between input characteristics and model architecture.

The enhanced speech test set reveals a contrasting trend for UniVERSA-T, where increasing token size actually improves performance (MSE decreases from 69.94 to 59.33). However, ARECHO still outperforms the baseline at both token sizes, achieving an MSE of 32.63 at 500 tokens compared to 43.32 at 1,000 tokens. This suggests that while UniVERSA-T benefits from increased context length in enhanced inputs, ARECHO achieves optimal performance with more concise representations.

Perhaps the most interesting results come from the corrupted speech test, where ARECHO shows a unique pattern: it is the only configuration where increasing the token size to 1,000 yields substantial improvements across all regression metrics (MSE decreases from 34.37 to 31.68, RMSE from 2.32 to 2.02, and MAE from 1.10 to 1.07). This suggests that when dealing with corrupted inputs, the additional context provided by longer token sequences allows ARECHO to better identify and mitigate noise through its confidence-oriented factorization mechanism. In contrast, UniVERSA-T shows significant performance degradation with increased token size in this domain, with MSE increasing from 39.80 to 60.73.

For the synthesized speech domain, both models show relatively stable performance across token sizes, with ARECHO achieving modest improvements at the larger token size (MSE decreases from 8.63 to 7.31). This stability in the most controlled experimental setting suggests that both models have sufficient capacity to capture the underlying patterns in synthetic data, though ARECHO consistently outperforms the baseline in classification metrics.

From a practical standpoint, these results have important implications for deploying ARECHO in real-world applications. First, the default token size of 500 appears to provide a good balance between performance and computational efficiency for most domains, particularly for Dev. and Enhanced inputs. Second, when dealing with corrupted inputs, increasing the token size to 1,000 can yield meaningful improvements, suggesting that adaptive token sizing based on input characteristics could be a valuable strategy. Finally, the consistent outperformance of ARECHO over UniVERSA-T across most metrics and domains, regardless of token size, underscores the robust advantages of our confidence-oriented factorization approach.

Table 12: Ablation study: the effect of beam size for ARECHO.

| Domain | Beam Size | Regression Metrics | | | | | | Classification Metrics | | | |
|---|---|---|---|---|---|---|---|---|---|---|---|
| | | MSE (↓) | RMSE (↓) | MAE (↓) | LCC (↑) | SRCC (↑) | KTAU (↑) | Acc (↑) | Precision (↑) | Recall (↑) | F1 (↑) |
| Dev. | 1 | 25.73 | 2.16 | 1.27 | 0.86 | 0.86 | 0.72 | 0.71 | 0.52 | 0.53 | 0.51 |
| | 2 | 25.12 | 2.14 | 1.27 | 0.86 | 0.86 | 0.72 | 0.71 | 0.52 | 0.53 | 0.50 |
| | 3 | 25.52 | 2.13 | 1.26 | 0.86 | 0.86 | 0.72 | 0.71 | 0.52 | 0.53 | 0.50 |
| | 4 | 26.07 | 2.22 | 1.28 | 0.85 | 0.86 | 0.72 | 0.71 | 0.52 | 0.53 | 0.50 |
| Enhanced | 1 | 20.58 | 2.09 | 1.32 | 0.84 | 0.85 | 0.69 | 0.72 | 0.52 | 0.54 | 0.51 |
| | 2 | 22.67 | 2.19 | 1.36 | 0.84 | 0.85 | 0.69 | 0.72 | 0.52 | 0.54 | 0.52 |
| | 3 | 22.98 | 2.20 | 1.37 | 0.84 | 0.85 | 0.69 | 0.72 | 0.52 | 0.54 | 0.51 |
| | 4 | 22.63 | 2.18 | 1.36 | 0.84 | 0.85 | 0.69 | 0.72 | 0.52 | 0.54 | 0.52 |
| Corrupted | 1 | 44.22 | 2.37 | 1.29 | 0.82 | 0.84 | 0.70 | 0.72 | 0.56 | 0.56 | 0.55 |
| | 2 | 41.02 | 2.26 | 1.23 | 0.83 | 0.84 | 0.70 | 0.73 | 0.56 | 0.56 | 0.55 |
| | 3 | 40.91 | 2.26 | 1.23 | 0.83 | 0.84 | 0.70 | 0.73 | 0.56 | 0.56 | 0.55 |
| | 4 | 41.10 | 2.26 | 1.23 | 0.83 | 0.84 | 0.70 | 0.73 | 0.56 | 0.56 | 0.55 |
| Synthesized | 1 | 4.99 | 1.13 | 0.58 | 0.91 | 0.91 | 0.78 | 0.79 | 0.67 | 0.66 | 0.65 |
| | 2 | 4.68 | 1.11 | 0.58 | 0.91 | 0.91 | 0.78 | 0.79 | 0.67 | 0.67 | 0.65 |
| | 3 | 4.85 | 1.12 | 0.57 | 0.91 | 0.91 | 0.78 | 0.79 | 0.67 | 0.66 | 0.64 |
| | 4 | 4.95 | 1.13 | 0.58 | 0.91 | 0.91 | 0.78 | 0.79 | 0.67 | 0.66 | 0.64 |

It is worth noting that the classification metrics remain relatively stable across token sizes for both models, with ARECHO consistently achieving higher accuracy, precision, recall, and F1 scores. This indicates that while regression performance may be more sensitive to token size variations, the models' discriminative capabilities are more robust to such changes.

## I.2 The Effect of Beam Search

In our main experiments, we mainly use greedy search to conduct the two-step confidence-oriented decoding. However, given the search space in the factorization space, we can also conduct the beam search on the problem. We use the ARECHO model trained on Base set as the candidate to test different beam sizes across various domains.

Our experimental results, as shown in Table 12, illustrate several important findings regarding the impact of beam search on model performance. First, we observe that increasing the beam size does not consistently lead to performance improvements across all domains and metrics. For instance, in the development set, beam size 2 yields the lowest MSE (25.12) and RMSE (2.14), while beam size 3 achieves the lowest MAE (1.26). However, these improvements over greedy search (beam size 1) are relatively marginal, suggesting that the confidence-oriented decoding approach is already effective at identifying high-quality factorizations even with a simple greedy search strategy.

Interestingly, the Enhanced domain shows a contrasting pattern, where greedy search (beam size 1) outperforms larger beam sizes on regression metrics such as MSE (20.58 vs. 22.67+ for larger beams). This counterintuitive result may stem from the nature of our confidence estimation mechanism, which might be optimized for the scoring function used in greedy search rather than the more complex search patterns in beam search. The classification metrics, however, remain consistent across beam sizes, indicating that the model's discriminative capabilities are robust to variations in the search strategy.

For the Corrupted domain, we observe that beam sizes 2-4 yield notable improvements over greedy search in regression metrics, with approximately 7% reduction in MSE (from 44.22 to around 41). This suggests that when dealing with noisy or corrupted inputs, the expanded search space provided by beam search allows the model to identify more reliable factorizations, potentially avoiding local optima that might trap the greedy approach.

The Synthesized domain, which represents our most controlled experimental setting, shows the best overall performance across all metrics. Here, beam size 2 achieves the optimal results with the lowest MSE (4.68) and RMSE (1.11), while beam size 3 yields the lowest MAE (0.57). The high correlation coefficients (LCC=0.91) across all beam sizes indicate that the model effectively captures the underlying relationships in this domain, regardless of the search strategy employed.

These findings have several implications for the deployment of ARECHO in practical applications. First, the choice of beam size should be context-dependent, with smaller beam sizes (1-2) preferable for standard and enhanced domains to balance computational efficiency and performance. For corrupted inputs, larger beam sizes may provide more robust results, albeit with diminishing returns beyond beam size 3. Second, the relatively consistent performance across beam sizes suggests that our

Table 13: Teacher-forced decoding analysis. We compare the original baseline, autoregressive ARECHO, and teacher-forced decoding (which injects ground-truth values at each decoding step to suppress error propagation). We use (T.F.) to note model use teacher-forcing decoding.

| Domain | Model | Regression Metrics | | | | | | Classification Metrics | |
|---|---|---|---|---|---|---|---|---|---|
| | | MSE (↓) | RMSE (↓) | MAE (↓) | LCC (↑) | SRCC (↑) | KTAU (↑) | Acc (↑) | F1 (↑) |
| Dev. | ARECHO | 40.95 | 2.16 | 1.27 | 0.86 | 0.86 | 0.72 | **0.71** | **0.51** |
| | ARECHO (T.F.) | **25.08** | **1.80** | **0.94** | **0.88** | **0.89** | **0.77** | 0.71 | 0.50 |
| Enhanced | ARECHO | 20.58 | 2.09 | 1.32 | 0.84 | 0.85 | 0.69 | **0.72** | 0.51 |
| | ARECHO (T.F.) | **3.80** | **1.01** | **0.43** | **0.88** | **0.91** | **0.76** | 0.72 | 0.51 |
| Corrupted | ARECHO | 44.22 | 2.37 | **1.29** | 0.82 | 0.84 | **0.70** | 0.72 | 0.55 |
| | ARECHO (T.F.) | **8.58** | **1.22** | 0.56 | **0.83** | **0.85** | **0.70** | 0.72 | 0.55 |
| Synthesized | ARECHO | 4.99 | 1.13 | 0.58 | 0.91 | 0.91 | 0.78 | 0.72 | 0.55 |
| | ARECHO (T.F.) | **2.00** | **0.80** | **0.30** | **0.92** | **0.93** | **0.79** | **0.79** | **0.64** |

Table 14: Comparison on MOS-related metrics across four public datasets. ARECHO shows consistent improvements in both regression and rank correlation measures over UniVERSA-T.

| Metric | Model | Regression Metrics | | | Correlation Metrics | | |
|---|---|---|---|---|---|---|---|
| | | MSE (↓) | RMSE (↓) | MAE (↓) | LCC (↑) | SRCC (↑) | KTAU (↑) |
| NISQA-MOS | UniVERSA-T | 0.41 | 0.64 | 0.43 | 0.86 | 0.83 | 0.69 |
| | ARECHO | **0.30** | **0.55** | **0.37** | **0.89** | **0.88** | **0.72** |
| UTMOS | UniVERSA-T | 0.06 | 0.25 | 0.19 | 0.97 | 0.97 | 0.85 |
| | ARECHO | **0.05** | **0.22** | **0.16** | **0.97** | **0.97** | **0.87** |
| PLCMOS | UniVERSA-T | 0.37 | 0.61 | 0.41 | 0.86 | 0.86 | 0.69 |
| | ARECHO | **0.32** | **0.56** | **0.38** | **0.89** | **0.89** | **0.72** |
| URGENT-MOS | UniVERSA-T | **0.20** | **0.45** | **0.35** | **0.78** | **0.78** | **0.62** |
| | ARECHO | 0.26 | 0.51 | 0.40 | 0.74 | 0.75 | 0.61 |

confidence-oriented decoding approach effectively identifies high-quality factorizations regardless of the exact search strategy, speaking to the robustness of our proposed methodology.

It is worth noting that while beam search expands the exploration of the factorization space, it comes with increased computational costs. The marginal improvements observed in most domains may not justify the additional computational burden in resource-constrained environments. Nevertheless, in critical applications where even small improvements in accuracy are valuable, selective application of beam search may be warranted, particularly when dealing with corrupted or noisy inputs.

## I.3 Error Propagation Analyais / Hybrid Decoding

To better understand the effect of error propagation in autoregressive decoding, we investigate the decoding with **teacher-forcing**, where ground-truth metric values are injected at each step during inference. This removes the accumulation of decoding errors and provides an upper bound on model performance under ideal conditioning.

As shown in Table 13, teacher-forced decoding consistently improves over autoregressive ARECHO, yielding a 25–50% reduction in prediction error. This quantifies the impact of error propagation in sequential metric estimation. The effect is most pronounced in the *corrupted* and *synthesized* conditions, where domain shifts amplify early mistakes.

These findings highlight a practical trade-off: while teacher-forcing suppresses propagation and approximates an upper bound, it relies on ground-truth access and thus forfeits the fully reference-free nature of ARECHO.

## I.4 Analysis on MOS Prediction

To further analyze the per-metric performance, we conduct a detailed comparison to investigate how inter-metric dependencies affect the modeling of MOS scores. As shown below, ARECHO consistently outperforms UniVERSA-T across multiple datasets, highlighting the benefit of inter-metric prediction and autoregressive dependency modeling.

Table 15: DNSMOS ablation on `ARECHO`. "Full" denotes multi-metric training, while "Only" refers to single-metric prediction.

| Metric | Model | Regression Metrics | | | Correlation Metrics | | |
|---|---|---|---|---|---|---|---|
| | | MSE (↓) | RMSE (↓) | MAE (↓) | LCC (↑) | SRCC (↑) | KTAU (↑) |
| DNSMOS | `ARECHO-full` | 0.04 | 0.21 | 0.16 | 0.85 | 0.81 | 0.65 |
| | `ARECHO-only` | **0.04** | **0.20** | **0.15** | **0.86** | **0.83** | **0.66** |

Overall, `ARECHO` demonstrates notable improvements in MSE and MAE, particularly for *NISQA-MOS*, *UTMOS*, and *PLCMOS*. The correlation gains (LCC/SRCC/KTAU) further suggest that `ARECHO` effectively captures cross-metric dependencies that are beneficial for reliable MOS prediction.

Interestingly, as shown in Table 15, the difference between `ARECHO-full` (multi-metric prediction) and `ARECHO-only` (single-metric prediction) is marginal for DNSMOS. This suggests that the benefit of inter-metric conditioning varies by dataset complexity and metric interdependence: tasks with stronger perceptual correlation (e.g., MOS-related metrics) benefit more from autoregressive chaining, whereas narrowly defined metrics like DNSMOS can be learned independently.

# J   Task-Oriented Metric Subset Analysis

To further examine whether training with all metrics is always optimal and whether unrelated metrics might negatively affect task-relevant ones, we conduct targeted experiments comparing *subset* and *fullset* training configurations across two representative domains: **speech synthesis** and **speech enhancement**. The results show that `ARECHO` remains robust under both settings, with diverse metric supervision generally improving or maintaining performance.

The dynamic classifier chain design enables each metric to be predicted at a random position within the autoregressive chain, conditioned on a variable subset of preceding metrics. This stochastic ordering allows `ARECHO` to capture informative inter-metric dependencies while factorizing away irrelevant information, preventing negative transfer even under mixed supervision.

All experiments (`UniVERSA` baseline, Subset, and Fullset) are trained on the same `Base` dataset described in Sec. 4.1 and detailed in Appendix E. The only difference lies in the set of target metrics:

- **Speech Synthesis Subset:** UTMOS, F0 correlation, WER, Language, etc.

- **Speech Enhancement Subset:** DNSMOS, PESQ, RIR Room Size, etc.

- **Fullset:** Union of all metrics from both domains.

No additional data is introduced for Fullset training; any performance differences therefore reflect the effect of training objectives rather than data volume.

Not all utterances contain all metrics in the base dataset due to data-source variability. For example:

- Only **13.22%** of samples contain *NISQA Real MOS* (human-annotated).

- Nearly all have *UTMOS* and *Language* (non-intrusive model-based).

- For enhancement data, ∼26.79% include intrusive metrics (*PESQ/STOI/SDR*), 80.44% have *SRMR*, and all have *DNSMOS* and *Qwen-Recording Quality*.

`ARECHO` naturally handles partial labels via its dynamic chain formulation, learning from whatever subset of metrics is available per sample. In contrast, `UniVERSA` relies on masking-based multi-task training, which reduces both data efficiency and inter-metric contextualization.

The results shown in Table 16-19 confirm three main trends:

- **Fullset training outperforms Subset training** for most metrics (*NISQA MOS*, *SRMR*, *SDR*) and performs on par for metrics such as *DNSMOS* and *STOI*.

- **No negative transfer is observed:** unrelated metrics do not degrade the performance of task-relevant ones.

Table 16: Subset vs. fullset training comparison for speech synthesis metrics (Synthesized test set is used for evaluation).

| Metric | Setup | MSE (↓) | RMSE (↓) | MAE (↓) |
|---|---|---|---|---|
| NISQA Real MOS | UniVERSA | 1.22 | 1.10 | 0.85 |
| | Subset | 0.43 | 0.65 | 0.41 |
| | Fullset | **0.05** | **0.23** | **0.12** |
| UTMOS | UniVERSA | 0.25 | 0.50 | 0.39 |
| | Subset | 0.09 | 0.30 | 0.22 |
| | Fullset | **0.04** | **0.20** | **0.13** |

Table 17: Subset vs. fullset training comparison for speech synthesis classification metrics (Synthesized test set is used for evaluation).

| Metric | Setup | Acc. (↑) | F1 (↑) |
|---|---|---|---|
| Language | UniVERSA | 0.90 | 0.88 |
| | Subset | 0.96 | 0.96 |
| | Fullset | **0.98** | **0.98** |

Table 18: Subset vs. fullset training comparison for speech enhancement metrics (Enhanced test set is used for evaluation).

| Metric | Setup | MSE (↓) | RMSE (↓) | MAE (↓) |
|---|---|---|---|---|
| SRMR | UniVERSA | 73.73 | 8.59 | 7.86 |
| | Subset | 4.02 | 2.01 | 1.26 |
| | Fullset | **1.83** | **1.35** | **0.96** |
| DNSMOS | UniVERSA | 5.50 | 2.34 | 2.29 |
| | Subset | 0.05 | 0.22 | 0.16 |
| | Fullset | **0.05** | **0.22** | **0.16** |
| STOI | UniVERSA | 0.16 | 0.41 | 0.39 |
| | Subset | 0.04 | 0.06 | 0.03 |
| | Fullset | **0.00** | **0.05** | **0.03** |
| SDR | UniVERSA | 151.75 | 12.32 | 10.70 |
| | Subset | 69.43 | 8.33 | 4.02 |
| | Fullset | **19.83** | **4.45** | **2.93** |
| PESQ | UniVERSA | 0.46 | 0.68 | 0.53 |
| | Subset | 0.19 | 0.44 | 0.33 |
| | Fullset | **0.17** | **0.41** | **0.31** |

Table 19: Subset vs. fullset training comparison for speech enhancement classification metrics (Enhanced test set is used for evaluation).

| Metric | Setup | Acc. (↑) | F1 (↑) |
|---|---|---|---|
| Qwen-Recording Quality | UniVERSA | 0.97 | 0.95 |
| | Subset | 0.97 | 0.95 |
| | Fullset | **0.97** | **0.95** |

- **Dynamic classifier chain improves stability:** randomizing dependency order during training allows decoupling of irrelevant signals while leveraging cross-metric correlations when beneficial.

Performance gains differ based on metric availability and difficulty:

**Label Availability and Dependency Opportunity.** Metrics with limited supervision but strong inter-metric correlation (e.g., *NISQA MOS*) benefit the most from joint training. Densely available metrics (*UTMOS*, *DNSMOS*) show smaller improvements.

**Metric Type and Modeling Difficulty.** Subjective and high-variance metrics gain more from multi-metric context, while objective or low-variance ones are largely self-sufficient.

**Examples.** *NISQA MOS* exhibits large improvements due to its sparse labels and correlation with perceptual quality scores. *SDR* benefits from complementary cues from *PESQ*, *DNSMOS*, and *STOI*, while *UTMOS* and *DNSMOS* remain stable due to high label coverage.

For further analysis of task-specific metric sensitivity, metrics can be grouped into:

- **Easy-dependent metrics:** high correlation and availability (e.g., *UTMOS*, *DNSMOS*) that perform well without additional context.
- **Hard-dependent metrics:** sparse or subjective metrics (e.g., *NISQA MOS*, *STOI*) that benefit strongly from auxiliary conditioning.

Overall, ARECHO demonstrates consistent performance across both homogeneous and heterogeneous metric sets. Joint training with diverse metric objectives does not harm, and often enhances, task-specific predictions. This stability stems from the dynamic classifier chain's ability to balance inter-metric dependency and independence, enabling scalable and interpretable multi-domain evaluation.

## K  Dependency Analysis - Dynamic Dependency Analysis

In this appendix, we show the complete order sequence of different metrics in Figure 1, 2, and 3. Based on the detailed orders, our analysis of the autoregressive prediction sequences reveals several important patterns:

1. **Foundational-to-Derived Metric Flow:** Across all speech types, we observe a consistent pattern where foundational characteristics (speaker, environment) predict derived measures (quality, intelligibility). This suggests the model has learned that basic physical properties constrain possible values of perceptual qualities, reflecting an implicit understanding of the causal structure in speech quality assessment.

2. **Context-Specific Dependency Anchors:** Each speech type exhibits distinct "anchor metrics" that appear early in the sequence:
   - Enhanced speech anchors on speaker identity (Q-Gender)
   - Corrupted speech anchors on environmental acoustics (RIR Room Size)
   - Synthesized speech anchors on background conditions (Q-Background)

   This reveals that the primary determinants of quality differ fundamentally based on the speech processing context.

3. **Categorical-Before-Numerical Pattern:** The consistent positioning of categorical metrics before numerical metrics suggests that discrete classifications provide efficient information compression that enables more accurate prediction of continuous measures. This aligns with efficient coding principles where high-level abstractions enable more precise low-level predictions.

4. **Signal Processing Metrics as Terminal Nodes:** Technical metrics (MCD, SI-SNR, PESQ) consistently appear in later positions, indicating they represent terminal nodes in the dependency graph that are influenced by multiple upstream factors rather than serving as predictive foundations.

5. **Non-Uniform Metric Inter-dependencies:** The varied positioning of some metrics across speech types (e.g., Q-Gender at position 0 in enhanced speech but position 12 in synthesized speech) suggests that inter-dependencies are not fixed but are highly context-dependent, challenging universal models of speech quality assessment.

These findings suggest that the model has learned a contextually adaptive compression of the speech quality space, where prediction sequences are optimized to minimize uncertainty in a hierarchical fashion. The emergent structures appear to reflect not just statistical correlations but meaningful organization of speech quality dimensions that aligns with human perceptual hierarchies.

## L  Dependency Analysis - Static Dependency Analysis

To investigate the effectiveness of dependency, we conduct static dependency analysis in the inference and fine-tuning. Here, we mainly focus on two static order discussions, including (1) an order of matching-required metrics comes first (order-mr) and (2) a coarse-to-fine conceptual order (order-c2f). Firstly, for the matching-required order, we follow the definition in VERSA Shi et al. (2025b), and create a static ordinary in both matching-required and non-matching-required. Secondly, to further

| Q-Gender | Q-SpeechImpariment | Q-SpeakingStyle | Q-EnvQuality | Q-PitchRange | Q-VocComplexity | Q-VolumeLevel | RealLanguage | Q-ContentRegister | SRMR |
|---|---|---|---|---|---|---|---|---|---|
| SpoofS | NISQA-NOI | Q-Emotion | AA-PC | Q-Background | AA-PQ | Q-ChannelType | LID | Q-Clarity | SE-CI-SDR |
| DNSMOSP.835 | SWR/SCR | Q-Purpose | WER | Q-VoiceType | SingMOS | SE-SI-SNR | Q-Lang | Q-SpeechRate | SCOREQ |
| NISQA-COL | Q-EmoVocalization | NISQA-LOUD | Q-SpeakerCount | Q-Age | PAM | UTMOS | AA-CE | NISQA-MOS | DNSMOSP.808 |
| SSQA | PLCMOS | Q-Pitch | AA-CU | CER | SE-SDR | UTMOSv2 | NISQA-DIS | CI-SDR | D-Distance |
| STOI | SE-SAR | SDR | SI-SNR | MCD | D-BERT | F0Corr | F0RMSE | SPK-SIM | D-BLEU |
| PESQ | URGENT MOS | SAR | CD | WSS | LLR | EMO-SIM | NCM | Covl | Reference Text Length |
| VISQOL | Csig | CSII-MID | Cbak | CSII-HIGH | SCOREQ w. Ref. | ASR-Mismatch | CSII-LOW | NOMAD | RIR Room Size |
| Noresqa | FWSEGSNR | RT60 | Predicted Text Length | SNR Simulation | NISQA Real MOS | VoiceMOS Real MOS | | | |

Figure 1: Visualization of metric order for enhanced speech test set via color blocks (Red = Early, Blue = Late).

| RIR Room Size | Q-SpeechImpariment | Q-Clarity | Q-EmoVocalization | Q-Purpose | Q-VocComplexity | Q-ContentRegister | Q-Gender | Q-Background | Q-SpeakerCount |
|---|---|---|---|---|---|---|---|---|---|
| Q-Lang | Q-ChannelType | SNR Simulation | Q-PitchRange | Q-SpeakingStyle | Q-VolumeLevel | Q-Pitch | Q-Emotion | Q-SpeechRate | Q-EnvQuality |
| Q-Age | Q-VoiceType | RT60 | Noresqa | UTMOS | DNSMOSP.835 | AA-CU | UTMOSv2 | NISQA-COL | ASR-Mismatch |
| AA-PC | D-Distance | Real Language | SCOREQ | NISQA-NOI | Predicted Text Length | PLCMOS | Reference Text Length | NOMAD | EMO-SIM |
| SPK-SIM | SWR/SCR | AA-PQ | AA-CE | LID | SingMOS | PAM | SpoofS | NISQA-LOUD | SE-SDR |
| D-BERT | DNSMOSP.808 | SCOREQ w. Ref. | SSQA | SE-SAR | SE-SI-SNR | SE-CI-SDR | NISQA-DIS | NISQA-MOS | D-BLEU |
| WSS | FWSEGSNR | CD | F0Corr | SDR | SAR | Cbak | LLR | Covl | CSII-HIGH |
| STOI | CSII-LOW | NCM | Csig | F0RMSE | MCD | VISQOL | CI-SDR | SRMR | PESQ |
| CSII-MID | SI-SNR | URGENT MOS | WER | CER | NISQA Real MOS | VoiceMOS Real MOS | | | |

Figure 2: Visualization of metric order for corrupted speech test set via color blocks (Red = Early, Blue = Late).

| Q-Background | NISQA-COL | Q-Purpose | Q-SpeechImpariment | AA-PQ | Q-ContentRegister | Q-EmoVocalization | Q-VoiceType | Q-PitchRange | Q-Emotion |
|---|---|---|---|---|---|---|---|---|---|
| Q-VolumeLevel | SSQA | Q-Gender | SWR/SCR | Q-Age | Q-ChannelType | NISQA-MOS | Q-SpeakerCount | UTMOS | Q-Lang |
| AA-CU | Q-Clarity | NISQA-DIS | Q-Pitch | Q-SpeakingStyle | LID | Q-EnvQuality | SCOREQ | DNSMOSP.835 | SE-SI-SNR |
| SingMOS | Q-VocComplexity | NISQA-NOI | SE-SAR | AA-PC | UTMOSv2 | Q-SpeechRate | RealLanguage | SE-SDR | SRMR |
| DNSMOSP.808 | SE-CI-SDR | PAM | PLCMOS | AA-CE | NISQA-LOUD | SpoofS | NISQAReal MOS | D-Distance | SCOREQ w. Ref. |
| D-BERT | RIR Room Size | SPK-SIM | D-BLEU | ASR-Mismatch | VoiceMOSReal MOS | SAR | Reference Text Length | F0Corr | EMO-SIM |
| Noresqa | MCD | F0RMSE | WSS | NOMAD | STOI | SDR | Predicted Text Length | CI-SDR | PESQ |
| WER | CD | SI-SNR | RT60 | Covl | Csig | NCM | CSII-HIGH | FWSEGSNR | URGENT MOS |
| CSII-LOW | LLR | VISQOL | CSII-MID | Cbak | SNR Simulation | CER | | | |

Figure 3: Visualization of metric order for synthesized speech test set via color blocks (Red = Early, Blue = Late).

explore the effectiveness of metric granularity, we then design a coarse-to-fine order, from the general audio quality and perceptual metrics, distortion and noise quality metrics, speech enhancement quality metrics, acoustic and prosodic characteristics and then down to speaker information, more complicated speech content, and final parts in emotion, and environmental contextualization. Both order-mr and order-c2f are detailed below:

```
order-mr Metrics

SE-SDR, SE-SAR, SE-SI-SNR, SE-CI-SDR, SDR, SAR, SI-SNR, CI-SDR, VISQOL,
PESQ, STOI, FWSEGSNR, LLR, WSS, CD, Csig, Cbak, Covl, CSII-HIGH, CSII-MID,
CSII-LOW, NCM, MCD, FORMSE, F0Corr, SPK-SIM, D-BERT, D-BLEU, D-Distance,
SCOREQ w.  Ref., ASR-Mismatch, WER, CER, Reference Text Length, Predicted
Text Length, SRMR, LID, RealLanguage, Q-Lang, NISQA-MOS, NISQA-NOI, NISQA-DIS,
NISQA-COL, NISQA-LOUD, SSQA, DNSMOSP.835, DNSMOSP.808, SCOREQ, PAM, SWR/SCR,
AA-CE, AA-CU, AA-PC, AA-PQ, SpoofS, Noresqa, NOMAD, Q-SpeakerCount, Q-Gender,
Q-Age, Q-SpeechImpariment, Q-Pitch, Q-PitchRange, Q-VoiceType, Q-VolumeLevel,
Q-ContentRegister, Q-VocComplexity, Q-Purpose, Q-Emotion, Q-Clarity,
Q-SpeechRate, Q-SpeakingStyle, Q-EmoVocalization, Q-Background, Q-EnvQuality,
Q-ChannelType, SNR Simulation, RIR Room Size, RT60, EMO-SIM, NISQA Real MOS,
UTMOS, UTMOSv2, PLCMOS, SingMOS, URGENT MOS, VoiceMOS Real MOS
```

```
order-c2f Metrics

NISQA-MOS, NISQAReal MOS, VoiceMOSReal MOS, UTMOS, UTMOSv2, PLCMOS, SingMOS,
URGENT MOS, SSQA, SCOREQ, SCOREQ w.  Ref., NISQA-NOI, NISQA-DIS, NISQA-COL,
NISQA-LOUD, DNSMOSP.835, DNSMOSP.808, SNR Simulation, RIR Room Size, Noresqa,
NOMAD, SpoofS, SE-SDR, SE-SAR, PESQ, STOI, SE-SI-SNR, SE-CI-SDR, SDR, SAR,
SI-SNR, CI-SDR, FWSEGSNR, LLR, WSS, CD, Csig, Cbak, Covl, CSII-HIGH, CSII-MID,
CSII-LOW, NCM, SWR/SCR, D-BERT, D-BLEU, D-Distance, MCD, FORMSE, F0Corr, PAM,
RT60, Q-Clarity, Q-SpeechRate, VISQOL, SPK-SIM, Q-SpeakerCount, Q-Gender,
Q-Age, Q-Pitch, Q-PitchRange, Q-VoiceType, Q-VolumeLevel, LID, RealLanguage,
Q-Lang, Reference Text Length, Predicted Text Length, ASR-Mismatch, WER,
CER, Q-VocComplexity, Q-ContentRegister, Q-Purpose, EMO-SIM, Q-Emotion,
Q-SpeakingStyle, Q-SpeechImpariment, Q-EmotionalVocalization, AA-CE, AA-CU,
AA-PC, AA-PQ, Q-Background, Q-EnvQuality, Q-ChannelType, SRMR
```

The results, shown in Table 20, reveal that static ordering can outperform dynamic beam search when the beam size is limited to 1. In the inference stage, order-c2f achieves notably better regression performance in the Enhanced and Synthesized domains. This suggests that gradually increasing the complexity of predicted metrics helps the model build robust internal representations, especially for perceptually grounded or synthetic speech signals. In contrast, order-mr yields improved regression metrics in the Corrupted domain and stronger classification metrics across most domains. This indicates that prioritizing matching-required metrics can help the model more effectively attend to severe distortions, particularly in noisy or degraded speech, which benefits tasks involving subjective perceptual scores (e.g., MOS).

In the fine-tuning stage, we initialize from the best-performing random-order sampled model and continue training using the static orders. However, we observe that performance tends to degrade compared to the pre-trained baseline. We attribute this to the reduced exploration capability of the model under a fixed decoding order, which may limit its flexibility to adapt across domains. Additionally, the divergent behavior across domains suggests that the optimal decoding order may be domain-specific, further highlighting the potential limitations of a static ordering scheme. In such cases, dynamic decoding strategies like beam search may offer better adaptability and robustness by allowing the model to flexibly determine the optimal metric prediction sequence based on context.

## M    Efficiency Analysis

To complement the performance evaluation of different models, we provide an analysis of their training efficiency in terms of average time per training epoch. Figure 4 compares the epoch-wise training time of UniVERSA, UniVERSA-T, and the proposed ARECHO model.

As shown in the figure, ARECHO achieves a notable reduction in training time, averaging **6632.09 seconds per epoch**, compared to UniVERSA (**7668.49 s**) and UniVERSA-T (**7923.34 s**). This improvement reflects the design efficiency of the ARECHO architecture, which maintains strong performance while accelerating the training process.

Table 20: Static dependency results on ARECHO.

| Stage | Domain | Static Order | Regression Metrics | | | | | | Classification Metrics | | | |
|---|---|---|---|---|---|---|---|---|---|---|---|---|
| | | | MSE (↓) | RMSE (↓) | MAE (↓) | LCC (↑) | SRCC (↑) | KTAU (↑) | Acc (↑) | Precision (↑) | Recall (↑) | F1 (↑) |
| Inference | Enhanced | Beam-1 | 20.58 | 2.09 | 1.32 | 0.84 | 0.84 | 0.70 | 0.72 | 0.52 | 0.54 | 0.51 |
| | | order-mr | 21.66 | 2.19 | 1.34 | 0.83 | 0.84 | 0.70 | 0.72 | **0.54** | **0.55** | **0.53** |
| | | order-c2f | **20.40** | 2.10 | **1.31** | **0.84** | 0.84 | 0.70 | 0.72 | 0.52 | 0.54 | 0.52 |
| | Corrupted | Beam-1 | 44.22 | 2.37 | 1.29 | 0.82 | 0.84 | 0.70 | 0.72 | 0.56 | 0.56 | 0.55 |
| | | order-mr | **39.19** | **2.25** | **1.23** | **0.83** | 0.84 | 0.70 | 0.72 | 0.55 | **0.57** | 0.55 |
| | | order-c2f | 42.83 | 2.32 | 1.26 | 0.82 | 0.84 | 0.70 | 0.72 | 0.56 | 0.56 | 0.55 |
| | Synthesized | Beam-1 | 4.99 | 1.13 | 0.58 | 0.91 | 0.91 | 0.78 | 0.79 | 0.67 | 0.66 | 0.65 |
| | | order-mr | 4.80 | 1.12 | 0.57 | 0.91 | 0.91 | 0.78 | 0.79 | 0.67 | **0.67** | 0.65 |
| | | order-c2f | **4.65** | **1.11** | **0.56** | 0.91 | 0.91 | 0.78 | 0.79 | 0.67 | 0.66 | 0.65 |
| Fine-tune | Enhanced | order-mr | 22.95 | 2.25 | 1.41 | 0.83 | 0.84 | 0.67 | 0.72 | 0.54 | 0.56 | 0.53 |
| | | order-c2f | 25.40 | 2.41 | 1.44 | 0.82 | 0.84 | 0.67 | 0.72 | 0.53 | 0.55 | 0.53 |
| | Corrupted | order-mr | 39.79 | 2.30 | 1.29 | 0.82 | 0.84 | 0.70 | 0.73 | 0.56 | 0.57 | 0.55 |
| | | order-c2f | 52.72 | 2.49 | 1.36 | 0.82 | 0.84 | 0.70 | 0.72 | 0.55 | 0.56 | 0.54 |
| | Synthesized | order-mr | 4.82 | 1.13 | 0.60 | 0.90 | 0.90 | 0.76 | 0.78 | 0.65 | 0.65 | 0.63 |
| | | order-c2f | 5.36 | 1.19 | 0.65 | 0.90 | 0.90 | 0.76 | 0.77 | 0.63 | 0.62 | 0.60 |

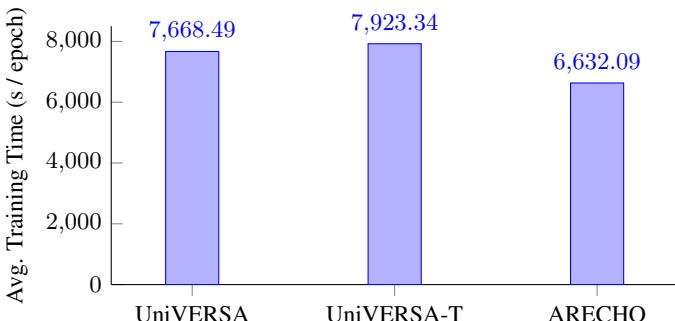

Figure 4: Average training time per epoch for `UniVERSA`, `UniVERSA-T`, and `ARECHO` on the `Base` training set.. The proposed `ARECHO` model demonstrates improved training efficiency.

Such efficiency is particularly advantageous for large-scale training or scenarios requiring frequent model updates. The reduction in computational cost also supports the scalability of `ARECHO` in practical deployment.

# N   Limitations

While ARECHO demonstrates strong performance and versatility across multiple speech evaluation tasks, several limitations remain:

- **Tokenization Granularity.** The conversion of continuous metrics into discrete tokens introduces a trade-off between resolution and model complexity. Although configurable, coarse quantization may lose subtle perceptual differences, while fine-grained tokenization increases sequence length and decoding burden.

- **Autoregressive Inference Overhead.** The sequential nature of the dynamic classifier chain, while beneficial for modeling inter-metric dependencies, results in higher inference latency compared to parallel prediction frameworks, potentially limiting real-time applicability.

- **Metric Order Sensitivity.** Despite randomized ordering during training, the model may still exhibit sensitivity to decoding order during inference, especially under domain shift or when limited context is available in early steps.

- **Partial Label Generalization.** Although ARECHO supports partially labeled supervision, its effectiveness under extreme label sparsity or domain-mismatched metric availability has not been extensively evaluated.

- **Dependence on Predefined Metadata.** The current system relies on manually defined metadata tokens for each metric. Scaling to hundreds of fine-grained evaluation metrics may require automated schema learning or ontology-aware modeling.

While ARECHO reduces reliance on expensive estimators, it does not replace human listening in high-stakes settings. Automated scores can be misapplied if used out of context (e.g., clinical decisions). We therefore recommend human-in-the-loop verification, transparent reporting of uncertainty, and restricted deployment. Finally, several targets are ordinal; integrating ordinal-aware objectives (e.g., Cao et al. (2020); Gutiérrez et al. (2015); Baccianella et al. (2009)) remains promising future work.

We leave addressing these limitations to future work, particularly exploring hybrid decoding strategies, more efficient dependency modeling, and extension to open-vocabulary or structured metric spaces.

## O   Broder Impact

**Positive Impacts.** ARECHO has the potential to improve the transparency, accessibility, and scalability of speech evaluation, particularly in applications involving speech synthesis, enhancement, and emotional communication. By providing a unified and interpretable multi-metric evaluation framework, this work can facilitate more equitable benchmarking of speech technologies across languages, devices, and acoustic conditions. Moreover, ARECHO's support for partial supervision and reference-free evaluation makes it especially valuable in low-resource or real-world deployment settings, where traditional evaluation pipelines may be infeasible. This could help advance assistive technologies, conversational AI, and accessibility tools for underrepresented communities and individuals with speech impairments.

**Negative Impacts.** At the same time, automated speech evaluation systems carry certain risks. First, if deployed naively or trained on biased datasets, they may reflect or amplify social, demographic, or linguistic biases, leading to unfair assessments of speech quality, intelligibility, or expressiveness across speaker groups. Second, while ARECHO supports partial supervision and reference-free evaluation, improper use in sensitive domains (e.g., hiring, education, or healthcare) could result in over-reliance on automated metrics without adequate human oversight. Lastly, as the system models multi-metric dependencies, interpretability claims must be contextualized carefully to avoid misleading conclusions about causality or human perception.

We encourage responsible use of this framework, particularly in human-facing applications, and emphasize the importance of representative training data, transparency in metric selection, and human-in-the-loop validation to ensure fairness and reliability.

## P   ARECHO's Logo Design

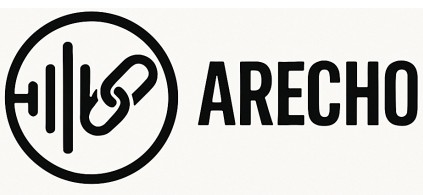

Figure 5: The ARECHO logo: a visual representation of chain-based, autoregressive speech evaluation.

The ARECHO logo was designed to reflect the system's core principles, autoregessive dependency modeling, multi-metric speech evaluation, and structured reasoning. It consists of two key elements:

- **Waveform Bars:** A stylized set of vertical lines representing an audio waveform, symbolizing the raw speech signal input.
- **Interlocked Chain Link:** Depicts ARECHO's classifier chain architecture. It highlights the autoregressive inference procedure and the inter-metric dependencies leveraged during prediction.

These elements are enclosed in a minimal circular boundary to suggest a cohesive and holistic system. Alongside, the bold, uppercase logotype "ARECHO" uses a geometric sans-serif font to convey precision, clarity, and technical strength.

Two visual variants of the logo were created:

- A **light-on-dark** version for slides and visual presentations.
- A **symbol-only** version for compact branding use (e.g., repository icons or badges).

The design encapsulates ARECHO's mission: to unify diverse speech evaluation metrics under a structured, dependency-aware modeling framework.

## Q  Acknowledgment

This work is supported by the Defence Science and Technology Agency (DSTA) in Singapore. We would like to thank Daniel Leong and Megan Choo for their valuable comments. Experiments of this work used the Bridges2 at PSC and Delta/DeltaAI NCSA computing systems through allocation CIS210014 from the Advanced Cyberinfrastructure Coordination Ecosystem: Services & Support (ACCESS) program, supported by National Science Foundation grants 2138259, 2138286, 2138307, 2137603, and 2138296.

