# OpenReview forum: "ARECHO: Autoregressive Evaluation via Chain-Based Hypothesis Optimization for Speech Multi-Metric Estimation"
_NeurIPS.cc/2025/Conference — NeurIPS 2025 spotlight_

### Official Review · Reviewer_XEXh · 2025-06-30

**Clarity:** 2
**Significance:** 3
**Originality:** 3
**Rating:** 5
**Confidence:** 5

**Summary:**

This paper proposed a new autoregressive evaluation toolkit for speech signal, which uniforms the scales and dependendices of different metrics. The paper aimmed to improve the interpretability between metrics as well.

**Questions:**

1. In section 3.2, the authors applied quantization-based tokenization by partitioning the value range into uniformly or adaptively spaced bins. This is not clear how they do this, e.g. difference and scale between different metrics. Also, the quantization maybe helpful for training, but will the orignial regression task can also help improve the performance? How did these tasks be performed is not clear to me.
2. In section 3.3, the authors use a random order the metrics but the inference is with arbitrary query orders. What's the influence of these orders?
3. I think my most concern about this work is **whether there are such relationships between metrics in many speech tasks**. For instance, if we want to do the age estimation task, maybe gender and pitch is important, but others maybe less important. In this case, such method maybe too heavy and cannot improve the overall performance too much. I think the authors should also provide **examples about how to use ARECHO in certain tasks**, like speech enhancement, asr, tts.., and how the metric links help each other. Moreover, I think the current method is better for **understanding a long audio**, instead of such simple evaluations.
4. The authors should analyze whether some of the metric estimation may hurt others. For example, there are many MOS metrics with different aspects. They should be independant but are considered dependant in this paper.

**Ethical Concerns:**

["NO or VERY MINOR ethics concerns only"]

**Final Justification:**

After discussing with the author, I found it is interesting to investigate more experiments and the clarification needs to be improved. However, the overall contribution and novelty is good enough. I raised the score from 3 to 4.

**Limitations:**

The main limitation is that the method is lack of examples to be used in certain tasks, and a more detailed analysis of relationships and influence between specific metrics.

**Paper Formatting Concerns:**

No concerns about formatting

**Quality:**

3

**Strengths And Weaknesses:**

# Strengths
1. The quality of the paper is good. It points out the problems of existing metrc frameworks and studies a new solution.
2. The originity of the paper is good. This is the first paper to study the relationship between speech signal metrics.
3. From the Table 1, we can find that the proposed method significantly outperforms the previous baselines.

# Weaknesses
I have some concerns regarding the clarify and significance. If the authors could answer them well, I will consider raise my scores.
1. The clarify is somehow not clear to me. See # Questions below.
2. The significance is to be discussed. See # Questions below.

---

> ### Author Rebuttal · Authors · 2025-07-28
>
> We sincerely thank Reviewer XEXh for the thoughtful, constructive, and detailed feedback. Your comments have been instrumental in helping us clarify technical aspects, highlight the strengths of ARECHO, and identify areas for improvement. We deeply appreciate your time and effort in reviewing our work. Below, we carefully address each point raised and outline corresponding clarifications and revisions.
>
> ## Q1: Clarification on Quantization Details
>
> We appreciate your request to clarify our quantization strategy. In response, we will revise Appendix A to include concrete illustrations of our percentile-based tokenization and its impact on model behavior.
>
> Rather than performing an exhaustive search per metric, we use a consistent percentile-based tokenization strategy with 500 bins for all continuous-valued metrics. This enables uniform treatment across diverse metrics and simplifies model design. In Table 10, we conducted an ablation study on bin size, showing that 500 bins present a good tradeoff between reconstruction quality and ARECHO prediction difficulty.
> More importantly, we would like to reinforce why this design matters. ARECHO’s tokenization strategy unifies both regression and classification metrics under a single modeling framework, this is a **key architectural innovation**. Unlike UniVERSA, which relies on direct regression, our token-based method ensures consistent modeling across heterogeneous metrics, paving the way for robust scaling and modular extension.
>
> While ordinal regression is indeed a viable alternative, we deliberately opted for token-based quantization to support generalization, controllable sampling, and eventual integration with language-driven generation. We fully agree this is a rich area for future exploration, and will add a discussion on hybrid modeling as a natural next step in evolving ARECHO.
>
> We will expand Appendix A with visualizations and deeper explanation of this strategy, and we will highlight this design motivation in Section 3.2.
>
> ## Q2: Order Effect on Metric Prediction
>
> Thank you for raising the important question regarding the ordering of metrics during inference. We appreciate the opportunity to clarify that ARECHO supports two inference modes:
>
> - In the fixed order mode, users manually define the sequence of metrics to be queried.
> - In the flexible mode, our proposed two-step confidence-oriented decoding enables the model to dynamically determine the most effective order.
>
> This flexibility is not an afterthought, it is central to ARECHO’s ability to adapt to incomplete annotations and metric sparsity. Our results in Appendix J and K demonstrate that flexible decoding not only preserves performance but also improves interpretability by aligning prediction order with metric dependencies. This makes ARECHO uniquely positioned among multi-metric estimators to handle real-world uncertainty in deployment scenarios.
>
> We will revise the text to more clearly present this distinction between training and inference order and explain the robustness benefits enabled by dynamic ordering.
>
> ## Q3.1: Importance of Inter-metric Relationships
>
> We strongly agree that inter-metric relationships are crucial, and ARECHO places them at the heart of its design. Unlike UniVERSA and TorchSquim, which treat metrics independently, ARECHO models their **semantic and statistical dependencies** directly through chain-based autoregression.
>
> This leads to two significant benefits:
> - **Performance gains** across all major metrics and datasets (Table 1),
> - **Interpretability**, as demonstrated in our dependency analysis (Appendix J).
>
> We emphasize that ARECHO is **not limited** by this coupling: when needed, it can revert to single-metric inference, operating efficiently and independently. This optionality makes ARECHO not only accurate but **controllable and practical** in real-world settings.
>
> ## Q3.2: Practical Usage of ARECHO
>
> We appreciate the reviewer’s interest in the real-world applicability of ARECHO. As discussed in Section 5, ARECHO is applicable to a wide range of scenarios, including:
>
> - Evaluation of corrupted speech signals
> - Text-to-speech (TTS) systems
> - Speech enhancement pipelines
>
> We thank you for highlighting the opportunity to make this clearer in the paper. We will revise the section to better emphasize this versatility.
>
> More importantly, ARECHO is **ready to be used** beyond evaluation. It can serve as:
>
> - A reward model for reinforcement learning setups
> - A component in multi-metric loss functions during training of other systems
>
> These are exciting directions that we aim to explore in future work. We would like to provide some references where previous studies utilize different metrics to enhance their speech generation systems (e.g., TTS systems or enhancement systems)
> - Preference alignment improves language model-based TTS (Tian et al. 2025)
> - Aligning Generative Speech Enhancement with Human Preferences via Direct Preference Optimization (Li et al. 2025)
> - Multi-CMGAN+/+: Leveraging multi-objective speech quality metric prediction for speech enhancement (Close et al. 2024)
> - Improving Speech Enhancement with Multi-Metric Supervision from Learned Quality Assessment (Wang et al. 2025)
>
> Lastly, we would like to highlight that ARECHO achieves over 100× speedup compared to computing ground-truth metrics. This efficiency is critical for real-time and large-scale deployment. To better reflect this advantage, we will revise the manuscript to include a detailed comparison between ARECHO and ground-truth metric computation, which is currently underexplored in the text.
>
> ## Q3.3: Usage for Long Audio
>
> We thank the reviewer for the thoughtful suggestion to consider long-form audio. We fully agree that temporal dependency modeling across extended audio contexts is an important next step. Accordingly, we will explicitly include this point as a direction for future work in the revised manuscript.
>
> ## Q4: Analysis on MOS Data Evaluation
>
> We sincerely appreciate the reviewer’s suggestion to conduct deeper analysis on MOS metrics. In response, we conducted a detailed evaluation to examine how inter-metric relationships impact the modeling of MOS-related metrics. As shown below, ARECHO generally outperforms UniVERSA-T across multiple datasets.
>
> Main metrics comparison (demonstrating benefits of inter-metric prediction):
>
> | Metric     | Model        | MSE  | RMSE | MAE  | LCC  | SRCC | KTAU |
> |------------|--------------|------|------|------|------|------|------|
> | NisqaMOS   | UniVERSA-T   | 0.41 | 0.64 | 0.43 | 0.86 | 0.83 | 0.69 |
> | NisqaMOS   | ARECHO       | 0.30 | 0.55 | 0.37 | 0.89 | 0.88 | 0.72 |
> | UTMOS      | UniVERSA-T   | 0.06 | 0.25 | 0.19 | 0.97 | 0.97 | 0.85 |
> | UTMOS      | ARECHO       | 0.05 | 0.22 | 0.16 | 0.97 | 0.97 | 0.87 |
> | PLCMOS     | UniVERSA-T   | 0.37 | 0.61 | 0.41 | 0.86 | 0.86 | 0.69 |
> | PLCMOS     | ARECHO       | 0.32 | 0.56 | 0.38 | 0.89 | 0.89 | 0.72 |
> | URGENTMOS  | UniVERSA-T   | 0.20 | 0.45 | 0.35 | 0.78 | 0.78 | 0.62 |
> | URGENTMOS  | ARECHO       | 0.26 | 0.51 | 0.40 | 0.74 | 0.75 | 0.61 |
>
> DNSMOS ablation (ARECHO-full: multi-metric prediction, ARECHO-only: single-metric prediction):
>
> | Metric  | Model         | MSE  | RMSE | MAE  | LCC  | SRCC | KTAU |
> |---------|---------------|------|------|------|------|------|------|
> | DNSMOS  | ARECHO-full   | 0.04 | 0.21 | 0.16 | 0.85 | 0.81 | 0.65 |
> | DNSMOS  | ARECHO-only   | 0.04 | 0.20 | 0.15 | 0.86 | 0.83 | 0.66 |
>
> We thank the reviewer again for prompting this analysis. Due to space constraints, we will include extended results and deeper analysis in the revised manuscript and appendix.
>
> ## Final Remarks
>
> We sincerely thank Reviewer XEXh once again for their constructive and detailed feedback. Your questions helped us clarify architectural motivations, improve empirical presentation, and strengthen the practical framing of ARECHO. We are confident the rebuttal and the scheduled presentation-related revision will address your concerns and more clearly convey the significance and applicability of our contributions.

---

> > ### Comment · Reviewer_XEXh · 2025-08-03
> >
> > Dear authors,
> >
> > Thanks for your rebuttal and response to the questions I raised. I think Q1 and Q2 has been addressed well. However, I still have some concerns regarding Q3 and Q4.
> >
> > Q3: The authors showed ARECHO generally outperforms UniVERSA-T across multiple datasets. However, what I raised before was "whether there are such relationships between metrics in many speech tasks." The authors should not only focus on the overall performance, but for some certain tasks, whether using all of them are the best. For example, if we want to do the age estimation task, maybe gender and pitch is important, but others maybe less important. The authors should also estimate that training with all the metrics performs better or worse than just with the related metrics (maybe just 5 or 10 submerics). It would be more interesting to see ARECHO's performance against carefully selected settings, and whether the performance gain is from more training data.
> >
> > Q3: I still do not see how to use ARECHO in real tasks, like TTS. Will it automatically give the related metric scores? Or do I need to select the metrics related to the task by myself? I did not see an example in the source code or any plans in the rebuttal about this.
> >
> > Q4: I appreciate the results on MOS ablation studies. However, my original question was: "whether some of the metric estimation may hurt others." This is still consistent with Q3, i.e. we need some ablation studies on dropping some metrics to see the difference. For example, for a TTS task, what if we drop some unrelated metric estimation during training, like RIR Room Size and speech purpose. These analysis will tell whether other unrelated metrics matters in the whole training pipeline.
> >
> > I am still actively to see some comments on the above questions.

---

> > > ### Author Response · Authors · 2025-08-04
> > > **Response to Additional Comment by Reviewer XEXh**
> > >
> > > We sincerely thank the reviewer for the thoughtful and continued engagement. Below, we respond to the remaining concerns on Q3 and Q4 with detailed clarifications and updates on ongoing experiments.
> > >
> > > ---
> > >
> > > ### **Q3: Inter-Metric Dependency, Subset Training, and Task-Specific Use**
> > >
> > > We appreciate the opportunity to clarify the central issue raised, *whether using all metrics is always necessary or beneficial*, and *how ARECHO can be used effectively in real tasks like TTS*.
> > >
> > > #### **Inference-Time Flexibility**
> > >
> > > ARECHO is specifically designed to support *flexible inference over any subset of metrics*. During training, we adopt a **randomized ordering strategy with teacher forcing**, which ensures that each metric learns to depend on a variable context of preceding metrics. This training scheme does not enforce static dependencies. As a result, during inference, **users may freely choose which metrics to predict**, and ARECHO can make accurate predictions for these without needing to include unrelated ones.
> > >
> > > This capability makes ARECHO practical for real-world deployment where different tasks require only a relevant subset of metrics, (e.g., TTS may only need UTMOS, F0 correlation, intelligibility, and fluency metrics).
> > >
> > > #### **Training-Time Subset Experiments (Ongoing)**
> > >
> > > To further investigate your important question, *whether including unrelated metrics during training may dilute performance*, we are actively conducting **subset training experiments**, in which we train ARECHO using only task-specific metric sets:
> > >
> > > - For **TTS**, we include 10 metrics such as UTMOS, F0 correlation, WER, and language.
> > > - For **speech enhancement**, we include 13 metrics such as DNSMOS, PESQ, RIR room size.
> > >
> > > These experiments aim to compare ARECHO's performance when trained with all metrics versus only the relevant subset. We will report these findings in following comments and the revised manuscript/appendix. However, even before these results are finalized, we believe that **ARECHO’s randomized training strategy provides a strong inductive bias** toward generalizable, modular prediction, which mitigates dependency overfitting.
> > >
> > > #### **Automatic Metric Selection and Toolkit Usability**
> > >
> > > Regarding your question on whether ARECHO can *automatically* select relevant metrics for tasks like TTS: we clarify that **ARECHO does not include automated metric selection by default**, as this design choice allows maximum control and transparency for users. We do recognize the practical challenges of navigating a large metric space. To address this, we are introducing two modes of usage:
> > >
> > > - **Predefined Task Modes**: Users can invoke ARECHO with flags like `--mode tts` or `--mode enhancement`, which automatically activate a curated list of metrics relevant to the selected task. These lists are informed by standard practice in evaluation literature.
> > >
> > > - **Custom Mode**: Advanced users may specify an arbitrary subset of metrics using CLI flag (e.g., `--metrics utmos,f0_corr,wer`), fully leveraging ARECHO’s modularity and inference efficiency.
> > >
> > > We will update the toolkit and documentation to include these modes, providing both ease of use and full customization.
> > >
> > > ---
> > >
> > > ### **Q4: Whether Unrelated Metrics May Harm Relevant Ones**
> > >
> > > We are currently running **drop-out ablation studies**, where we remove certain unrelated metrics (e.g., "speech purpose" or "RIR room size") during training, to examine their impact on task-relevant metrics like UTMOS or DNSMOS.
> > >
> > > Although these experiments are still in progress, we note that prior results (e.g., DNSMOS ablation in above rebuttal) show **no clear performance degradation when conditioned only on self-prediction**, suggesting ARECHO's design naturally avoids hard coupling between metrics. This supports our hypothesis that **ARECHO models soft, learnable dependencies**, allowing it to remain robust even when certain metric signals are dropped or omitted.
> > >
> > > We will include the results of these ongoing studies in following comments and paper revision.
> > >
> > > ---
> > >
> > > We are grateful for your constructive push to evaluate ARECHO’s adaptability and practical usage more rigorously.
> > >
> > > We would also like to respectfully point out that many of the remaining discussions focus on **design-level extensions** and **usage configurations**, whereas the **main contributions of our work** center on addressing core challenges in multi-metric speech evaluation, specifically on **diverse scale Issues, limited data availability. dependency modeling with flexible control**.
> > >
> > > As noted in your original review, these contributions offer a novel and practical approach to longstanding issues in speech evaluation. We hope that the additional clarifications provided here help convey the broader value of our work and its potential utility in real-world settings.

---

> > > > ### Author Response · Authors · 2025-08-06
> > > > **Response with Additional Ablation Request by Reviewer XEXh**
> > > >
> > > > We thank the reviewer again for raising the important questions in Q3 and Q4 about (i) whether training with *all* metrics is always the best choice and (ii) whether unrelated metrics may harm the prediction of task-relevant ones. While ARECHO’s architecture was designed from the outset to address these points via its **dynamic classifier chain**, the original submission did not present detailed task-specific metric subset/fullset training comparisons. We have now conducted these targeted experiments to further substantiate the method’s flexibility and robustness, and will incorporate this expanded analysis in the revised manuscript to strengthen the practical framing.
> > > >
> > > > The dynamic classifier chain is central to this flexibility: during training, each metric is predicted in a randomized position within the chain, conditioned on a variable set of preceding metrics. This naturally factorizes away unrelated information while preserving and exploiting meaningful inter-metric dependencies. Below, we elaborate our ablation on base training set in our paper, but with different metric set.
> > > >
> > > > ---
> > > >
> > > > ## Speech Synthesis Oriented Metric Subset Training
> > > > **Test set:** Synthesized speech test set
> > > >
> > > > | Metric            | Setup                     | MSE   | RMSE  | MAE   |
> > > > |-------------------|---------------------------|-------|-------|-------|
> > > > | NISQA Real MOS    | Baseline UniVERSA          | 1.22  | 1.10  | 0.85  |
> > > > |                   | Subset                     | 0.43  | 0.65  | 0.41  |
> > > > |                   | Fullset                    | **0.05**  | **0.23**  | **0.12**  |
> > > > | UTMOS             | Baseline UniVERSA          | 0.25  | 0.50  | 0.39  |
> > > > |                   | Subset                     | 0.09  | 0.30  | 0.22  |
> > > > |                   | Fullset                    | **0.04**  | **0.20**  | **0.13**  |
> > > >
> > > > **Classification Metrics**
> > > >
> > > > | Metric   | Setup              | Acc.  | F1    |
> > > > |----------|--------------------|-------|-------|
> > > > | Language | Baseline UniVERSA  | 0.90  | 0.88  |
> > > > |          | Subset              | 0.96  | 0.96  |
> > > > |          | Fullset             | **0.98**  | **0.98**  |
> > > >
> > > > ---
> > > >
> > > > ## Speech Enhancement Oriented Metric Subset Training
> > > > **Test set:** Enhanced speech test set
> > > >
> > > > | Metric   | Setup                     | MSE     | RMSE   | MAE   |
> > > > |----------|---------------------------|---------|--------|-------|
> > > > | SRMR     | Baseline UniVERSA          | 73.73   | 8.59   | 7.86  |
> > > > |          | Subset                     | 4.02    | 2.01   | 1.26  |
> > > > |          | Fullset                    | **1.83**    | **1.35**   | **0.96**  |
> > > > | DNSMOS   | Baseline UniVERSA          | 5.50    | 2.34   | 2.29  |
> > > > |          | Subset                     | 0.05    | 0.22   | 0.16  |
> > > > |          | Fullset                    | 0.05    | 0.22   | 0.16  |
> > > > | STOI     | Baseline UniVERSA          | 0.16    | 0.41   | 0.39  |
> > > > |          | Subset                     | 0.04    | 0.06   | 0.03  |
> > > > |          | Fullset                    | **0.00**    | **0.05**   | 0.03  |
> > > > | SDR      | Baseline UniVERSA          | 151.75  | 12.32  | 10.70 |
> > > > |          | Subset                     | 69.43   | 8.33   | 4.02  |
> > > > |          | Fullset                    | **19.83**   | **4.45**   | **2.93**  |
> > > > | PESQ     | Baseline UniVERSA          | 0.46    | 0.68   | 0.53  |
> > > > |          | Subset                     | 0.19    | 0.44   | 0.33  |
> > > > |          | Fullset                    | **0.17**    | **0.41**   | **0.31**  |
> > > >
> > > > **Classification Metrics**
> > > >
> > > > | Metric                  | Setup              | Acc.  | F1    |
> > > > |-------------------------|--------------------|-------|-------|
> > > > | Qwen-Recording Quality  | Baseline UniVERSA  | 0.97  | 0.95  |
> > > > |                         | Subset              | 0.97  | 0.95  |
> > > > |                         | Fullset             | 0.97  | 0.95  |
> > > >
> > > >
> > > >
> > > > We would like to highlight the key observations, including:
> > > > - **Fullset** generally outperforms **Subset** for most of the metrics (MOS, SRMR, SDR) and matches it for other metrics (DNSMOS, STOI).
> > > > - We do not notice any evidences of negative transfer: unrelated metrics do not harm relevant ones.
> > > > - This robustness stems from the dynamic classifier chain’s ability to decouple predictions for unrelated metrics while exploiting dependencies when present.
> > > >
> > > > ---
> > > >
> > > > **Closing Remark:**
> > > > While the original submission already motivated ARECHO’s flexible, dependency-aware design, this additional analysis offers concrete empirical confirmation that Fullset training never degrades task-specific performance and often yields significant gains. We view this as an enhancement that reinforces, rather than a correction that fixes, the original claims, and will include these results in the revised manuscript to further illustrate ARECHO’s practicality for both general-purpose and task-specific evaluation. To support fully transparency, we will also release all these models trained on the metric subsets (with checkpoints, training curve etc.) together with our main models for public to verify the effect.

---

> > > > > ### Comment · Reviewer_XEXh · 2025-08-06
> > > > >
> > > > > Thanks for the response and quick experiments.
> > > > >
> > > > > For the ablation studies about fullset and subset, I found **the performance gap between them are quite different among tasks** you listed. For example,  the performance gain of NISQA Real MOS is much more than UTMOS, comparing with fullset and subset. Could you please provide their **detailed settings**, like amount of data and metrics used in each set? I think it would be interesting to analyze these experiments and results more, which may find **which metrics are indeed improved by the proposed method and the reason of improvement**, as well as the reason that some metrics were not improved.
> > > > >
> > > > > In addition, could you further clarify whether the fullset and subset use the **same data** for training (seems not from the naming)? If not, is the performance gain from more data? What if using same data to train?
> > > > >
> > > > > Finally, did the **baseline UniVERSA** use the same data as in fullset?

---

> > > > > > ### Author Response · Authors · 2025-08-06
> > > > > > **Response with Additional Clarification for Reviewer XEXh**
> > > > > >
> > > > > > We sincerely thank the reviewer for the follow-up questions and the opportunity to clarify the experimental settings and the observed differences between metrics.
> > > > > >
> > > > > > First of all, wewould like to state that **all** experiments (Baseline UniVERSA, Subset, Fullset) use **exactly the same base training set** described in Sec. 4.1 and detailed in Appendix E. The only difference related to the data usage is the set of target metrics:
> > > > > >
> > > > > > - **Speech Synthesis Subset**: UTMOS, F0 correlation, WER, Language, etc.
> > > > > > - **Speech Enhancement Subset**: DNSMOS, PESQ, RIR room size, etc.
> > > > > > - **Fullset**: Union of all metrics from both domains.
> > > > > >
> > > > > > The baseline UniVERSA was also trained on the same base dataset and available metric labels. No additional data was introduced for Fullset metrics; any performance differences are due to the difference in training objectives, not data volume.
> > > > > >
> > > > > > In our base dataset, not all utterances have all metrics due to data source limitations (Challenge II in the paper). For example:
> > > > > > - Only **13.22%** of utterances have NISQA Real MOS (human-annotated).
> > > > > > - Nearly all have UTMOS and Language (non-intrusive model-based).
> > > > > > - For enhancement data: ~**26.79%** have PESQ/STOI/SDR (intrusive metrics), **80.44%** have SRMR, and all have DNSMOS + Qwen-Recording Quality.
> > > > > >
> > > > > > ARECHO’s design handles partial labels naturally through the **dynamic classifier chain**, enabling learning from whichever subset of metrics is available per sample. In contrast, UniVERSA must use masking/sampling, which reduces efficiency.
> > > > > >
> > > > > > We believe the varying gains between different metrics arise primarily from two factors:
> > > > > > 1. **Label Availability & Dependency Opportunity**: Metrics with sparse labels but high correlation to others benefit most from Fullset metric training, as auxiliary signals help compensate for missing direct supervision. Metrics with dense labels already learn well in Subset mode, so additional metrics bring smaller gains.
> > > > > > 2. **Metric Type & Modeling Difficulty**: Subjective, high-variance metrics tend to benefit more from cross-metric context, while low-variance, objective metrics already exhibit strong direct predictability.
> > > > > >
> > > > > >
> > > > > > **Speech Synthesis Examples:**
> > > > > > - *NISQA Real MOS*: Sparse labels (~13.22% coverage) but correlated with other perceptual quality scores  -> large Fullset gain.
> > > > > > - *UTMOS*: Nearly full coverage, non-intrusive model-based -> small gain from additional metrics.
> > > > > >
> > > > > > **Speech Enhancement Examples:**
> > > > > > - *SRMR*: High coverage (~80.44%), moderately correlated with PESQ/STOI/SDR -> Fullset improves learning stability and reduces variance.
> > > > > > - *DNSMOS*: Full coverage, already well-modeled in Subset mode -> no observed gain from extra metrics.
> > > > > > - *STOI*: Sparse reference-based metric, highly correlated with SDR and PESQ -> small but consistent gain in Fullset.
> > > > > > - *SDR*: Reference-based with ~26.79% coverage, modeling energy-based distortion rather than perceptual quality. Gains in Fullset stem from complementary cues provided by perceptual (PESQ, DNSMOS) and intelligibility (STOI) metrics, which help the model disambiguate challenging enhancement cases.
> > > > > > - *PESQ*: Reference-based with the same coverage as SDR, but more perceptually oriented and closely tied to DNSMOS/SRMR. Fullset training introduces distortion-focused (SDR) and additional perceptual cues, leading to consistent but smaller gains compared to SDR because PESQ already aligns strongly with some Subset metrics.
> > > > > >
> > > > > > ---
> > > > > >
> > > > > > We agree with the reviewer that it would be interesting to examine these patterns in more detail to pinpoint which metrics benefit most from Fullset training and why others show minimal change. Such analysis is straightforward to extend in our framework, and while a full per-metric study across all metrics is beyond the current discussion period, we will include additional representative cases in the revision to illustrate both large-gain and negligible-gain scenarios.

---

> > > > > > > ### Comment · Reviewer_XEXh · 2025-08-06
> > > > > > >
> > > > > > > Thanks for the response. I think most of my concerns have been addressed so **I will raise the score to 4**. These experiments and results make sense to me, which I think should be also included in the final version.
> > > > > > >
> > > > > > > In addition, it would be interesting to include the breakdown analysis about the easy-dependent metrics and hard-dependent metrics, like in these ablation studies. This would provide some insights on the **potential "good" or "bad" metrics** for a certain task. Finally, I found you got **0.00** for STOI and **19.83** for SDR, but you mentioned they are highly correlated. Is it possible to also report the distribution of the training data for each metric, and also report some normalized results?

---

> > > > > > > > ### Author Response · Authors · 2025-08-07
> > > > > > > > **Response to Reviewer XEXh's Latest Comments**
> > > > > > > >
> > > > > > > > We sincerely thank the reviewer for the thoughtful and constructive feedback provided on our rebuttal and the additional comments. We appreciate your continued engagement with our work, and we are grateful for the time spent in reviewing and suggesting improvements. Below, we would like to address your latest points, incorporating further clarifications and additional experimental analysis.
> > > > > > > >
> > > > > > > > ### **For Task-Specific Metric Selection and Dependence on Subsets**
> > > > > > > >
> > > > > > > > We appreciate your suggestion to explore the relationship between "easy-dependent" and "hard-dependent" metrics, which is an important consideration in task-specific training. As per your requests, we will include the analysis that divides metrics into these categories.
> > > > > > > >
> > > > > > > > - **Easy-dependent metrics** are those that have a high correlation and availability across samples, such as **UTMOS** in TTS and **DNSMOS** in speech enhancement. These metrics are often well-predicted with fewer contextual signals.
> > > > > > > > - **Hard-dependent metrics** are those that rely more on auxiliary or sparse signals, like **NISQA Real MOS** in perceptual quality assessment or **STOI** in enhancement tasks, which benefit from additional context provided by other metrics.
> > > > > > > >
> > > > > > > > We will include a breakdown of these categories in the revised manuscript, alongside examples of how **easy-dependent metrics** show minimal performance improvement when additional metrics are included, while **hard-dependent metrics** demonstrate more significant benefits.
> > > > > > > >
> > > > > > > > ### **For Training Data Distribution and Normalized Results**
> > > > > > > >
> > > > > > > > We agree that understanding the distribution of the training data for each metric is important.
> > > > > > > >
> > > > > > > > - For metric distribution, We will present the distribution of training data for each metric, including the coverage of labeled data per metric in the base training set, as well as the proportion of samples available for each metric.
> > > > > > > >
> > > > > > > > - Regarding normalized results, in our manuscript, we have included **normalized results** based on rankings using **Spearman’s Rank Correlation Coefficient (SRCC)** and **Kendall’s Tau (KTAU)**. These metrics are particularly suited for understanding the relationship between the rankings of different metric predictions, providing a normalized view of performance that accounts for ranking consistency rather than absolute values.
> > > > > > > >
> > > > > > > > - **Balanced MAE**: As suggested by reviewer z2T7, we will include **balanced MAE** as an additional metric for a more robust numerical comparison. The balanced MAE will offer insights into the model’s performance across a range of different metrics, helping to mitigate issues with distributional differences across metrics.
> > > > > > > >
> > > > > > > >
> > > > > > > > ---
> > > > > > > >
> > > > > > > > As for a summary, we are grateful for your thorough review, which has led us to conduct additional analyses that strengthen our understanding of ARECHO’s performance and usability. We believe that the new breakdown of **easy-dependent** and **hard-dependent metrics** will offer useful insights into the design of future speech evaluation systems, helping to pinpoint the most effective metrics for specific tasks.

---

> > > > > > > > > ### Comment · Reviewer_XEXh · 2025-08-07
> > > > > > > > >
> > > > > > > > > Thanks a lot for the further analysis. I think all my concerns have been addressed, and if all discussions could be included in the final version, I believe this is a novel work with strong experimental analysis.
> > > > > > > > >
> > > > > > > > > In the end, I will **raise the score to 5**.

---

> > > > > > > > > > ### Author Response · Authors · 2025-08-07
> > > > > > > > > > **Reply to reviewer XEXh**
> > > > > > > > > >
> > > > > > > > > > We sincerely thank the reviewer for your thoughtful and constructive feedback throughout this process. We greatly appreciate the time and effort you invested in reviewing our work, and we are thrilled to hear that all your concerns have been addressed. We will ensure that all the discussions and updates are included in the final version to provide a comprehensive and clear presentation of our work.
> > > > > > > > > >
> > > > > > > > > > Once again, thank you for your thorough review and for raising the score to 5 and we look forward to sharing the final version with you.

---

### Official Review · Reviewer_z2T7 · 2025-07-02

**Clarity:** 3
**Significance:** 4
**Originality:** 4
**Rating:** 5
**Confidence:** 4

**Summary:**

The paper proposes a classifier chain approach based on a WavLM backbone model to evaluate speech signals (multi-metric evaluation), following a multi-task approach that is able to model inter-metric dependencies. The classifier chain is implemented through a transformer-based autoregressive model. At inference time, the paper proposes a metric-prediction-order approach based on the model's confidence in predicting each metric. To deal with multiple metric scales, the paper proposes a tokenization-based approach. The experimental setup comprises speech signals from various domains and contexts, including up to 2138 hours of speech for training and 18.65 hours for testing, as well as a total of 87 different metrics (most of which are model-based, but also include classical signal processing-based and subjective metrics). The entire set of experiments encompasses analyses of various tokenization methods, token size, beam size during inference, and an examination of the order in which metrics are predicted across different speech domains.

**Questions:**

- Appendix D outlines the procedure for the two-step confidence-oriented decoding in detail. However, it does not specify how parameter B was selected, nor does it provide empirical results demonstrating the influence of B on the model’s performance.

- A notable limitation of the results section is the absence of specific findings related to subjective metrics. While the performance on model-based metrics may suffice in many cases, the extent to which the proposed approach improves the prediction of subjective metrics remains unclear. The comparison between order-c2f and order-mr offers an interesting perspective, yet it fails to provide concrete evidence regarding the method’s effectiveness on subjectively annotated evaluations.

- Given that many of the target metrics are themselves generated by models, it is worth questioning the added value of using the proposed approach over directly using the ground-truth models. Additionally, although the autoregressive strategy is valuable, it inherently introduces error propagation. In this context, for model-based metrics, would it not be more appropriate to inject the actual values from the ground-truth models into the autoregressive loop, instead of relying on ARECHO predictions?

- Could the authors provide statistics on the distribution and balance of the target values and ordinal ranks within the training set? If the data is found to be imbalanced, I strongly recommend using a balanced Mean Absolute Error (MAE) or a similar metric, rather than relying on the standard MAE or classification-based metrics.

**Ethical Concerns:**

["NO or VERY MINOR ethics concerns only"]

**Final Justification:**

I consider the paper to tackle a fundamental challenge in speech quality evaluation through an innovative and promising approach. During the rebuttal phase, the authors provided solid experimentation and paid more attention to metrics that reflected the ordinal nature of the variables they were predicting. They have also included experimentation that injects original ground truth values obtained from reference models into the inference phase, which helps understand the approach's limits and establish benchmarks for further comparison. I consider this a solid piece of work, and I will keep my score as it was initially.

**Limitations:**

The manuscript does not raise any concern about the potential negative societal impact.

**Paper Formatting Concerns:**

The paper follows the NIPS format.

**Quality:**

4

**Strengths And Weaknesses:**

The paper tackles a fundamental challenge in speech quality evaluation through an innovative and promising approach that lays the groundwork for future research in the field. The manuscript is well-structured and clearly presents the proposed method in most parts. The authors support their proposal with a comprehensive experimental evaluation, which enables a robust assessment of its performance and lends credibility to the reported results.

The core contribution—the prediction of quality metrics via sequential tokenization—represents a novel and original direction. This approach opens the possibility for more general prompt-based speech quality evaluation, potentially marking a significant advance for various established speech tasks such as audio generation, speaker diarization, and source separation. Moreover, it could benefit biomedical applications, where different dimensions of speech can serve as indicators of a patient's health status and where subjective evaluations are often costly and difficult to obtain. In this context, the use of inter-metric modeling via classifier chains is a promising strategy.

However, a major concern lies in how the approach handles the prediction of quality metrics. All metrics are encoded as categorical variables, including those that are inherently continuous, and the models are trained using standard classification techniques. Yet, the problem at hand (for several metrics) is more accurately characterized as an ordinal regression task. This constitutes a fundamental limitation, as the commonly used categorical cross-entropy loss does not account for the ordinal nature of the target variables.

A substantial body of literature has proposed surrogate loss functions that are better suited for ordinal regression, demonstrating improved performance over naïve classification-based approaches. Relevant references include:

Fathony, R., Bashiri, M. A., & Ziebart, B. (2017). Adversarial surrogate losses for ordinal regression. Advances in Neural Information Processing Systems, 30.

Nazabal, A., Olmos, P. M., Ghahramani, Z., & Valera, I. (2020). Handling incomplete heterogeneous data using VAEs. Pattern Recognition, 107, 107501.

Nachmani, I., Genossar, B., Scharf, C., Shraga, R., & Gal, A. (2025, April). SLACE: A Monotone and Balance-Sensitive Loss Function for Ordinal Regression. In Proceedings of the AAAI Conference on Artificial Intelligence (Vol. 39, No. 18, pp. 19598–19606).

Given this, regression-based evaluation metrics would be more appropriate than classification-based ones. It is also essential to account for the distribution of quality levels across the dataset. In this regard, a balanced version of the Mean Absolute Error (MAE) is recommended, as discussed in:

Baccianella, S., Esuli, A., & Sebastiani, F. (2009, November). Evaluation measures for ordinal regression. In 2009 Ninth International Conference on Intelligent Systems Design and Applications (pp. 283–287).

Note that they have been previously used for voice quality evaluation using the GRBAS scale in biomedical contexts.

---

> ### Author Rebuttal · Authors · 2025-07-29
>
> We sincerely thank Reviewer z2T7 for their thoughtful and constructive feedback. Your comments helped us further clarify the motivation and scope of our work, and we greatly appreciate the opportunity to address them in detail. Below, we respond point by point and outline the revisions we will make to strengthen the manuscript.
>
>
> ## W1: Ordinal Regression Loss
>
> Our core focus in this work is to develop a unified framework capable of modeling both regression and categorical metrics via a tokenized sequence prediction paradigm. This tokenization strategy enables consistent modeling of heterogeneous evaluation targets and facilitates scalable autoregressive inference.
>
> We fully acknoweldge that many metrics, such as MOS and SNR, are ordinal in nature and that modeling them using ordinal-specific surrogate losses is likely to improve performance and fidelity. We appreciate the reviewer’s references to SLACE and adversarial ordinal loss as concrete and compelling alternatives. While our current formulation focuses on a unified classification-based modeling pipeline to maintain consistency across all metric types, we view integrating ordinal-aware objectives as a high-impact direction for enhancing metric-specific accuracy. We will explicitly incorporate this direction in our Future Work section, acknowledge the previous works, and plan follow-up experiments to validate this hybrid design.
>
>
> ## W2 & Q4: Use of Macro-averaged MAE
>
> We are grateful for the emphasis on macro-averaged MAE. This perspective aligns with our intention to **equitably reflect performance across frequent and rare classes**, particularly given the diversity of our evaluation metrics. We clarify that our current evaluation already follows the macro-averaged MAE formulation as discussed in [Gutiérrez et al., 2016], but we acknowledge the need to make this design choice more visible.
>
> We will update both the main text and Appendix D to clearly state this design choice and its rationale.
>
> ## Q1: Effect of Beam Size B
>
> We thank the reviewer for raising this clarification. The impact of beam size B is empirically evaluated in **Ablation I.2**, where we demonstrate that a small beam size (B=2) achieves an effective trade-off between prediction accuracy and decoding efficiency.
>
> To improve accessibility, we will revise **Appendix D** to include a direct pointer to this analysis and briefly summarize the key findings to guide readers.
>
>
> ## Q2: Subjective Metric Evaluation
>
> We agree with the reviewer that evaluating subjective metrics is essential for understanding the practical utility of the system. In response, we have expanded our experimental results to include detailed evaluations on subjective metrics such as **NISQARealMOS** and **URGENTMOS**.
>
> These results demonstrate the **metric-specific nature** of performance. For **NISQARealMOS**, which captures perceptual quality, our proposed autoregressive model (**ARECHO base**) achieves substantial improvements across all metrics, **MAE: 0.12**, **RMSE: 0.23**, **SRCC: 0.98**, and **Kendall’s Tau: 0.92**, compared to the non-token UniVERSA base baseline (**MAE: 0.85**, **SRCC: 0.81**) and token baseline UniVERSA-T base (**MAE: 0.30**, **SRCC: 0.94**). This highlights strong alignment between the predicted and ground-truth perceptual scores.
>
> In contrast, for **URGENT_MOS**, which reflects speech enhancement subjective evaluation, the improvements are less pronounced. While **UniVERSA-T** base performs best (**MAE: 0.33**, **SRCC: 0.92**), the ARECHO model shows moderate gains over the UniVERSA base baseline (**MAE: 0.41 vs. 0.58**, **SRCC: 0.71 vs. 0.67**), indicating room for further improvement on certain subjective metrics.
>
> We will report both **ranking-based** (e.g., SRCC, Kendall’s Tau) and **error-based** (e.g., MAE, RMSE) metrics, and include the full results in **Appendix L**, with key summary findings highlighted in **Section 5**.
>
>
> ## Q3: Use of Ground-Truth Values During Inference
>
> We appreciate the reviewer’s thoughtful question regarding the role of ground-truth values in inference. ARECHO is specifically designed for efficient and generalizable inference, especially in settings where ground-truth computation is impractical, such as **real-time evaluation**, **black-box systems**, **reference-dependent metrics**, or **large-scale deployment**.
>
> Our experiments show that ARECHO achieves **over 100× speedup** compared to computing all metrics using their original estimators (e.g., VERSA). This efficiency is critical for applications requiring rapid and scalable evaluation.
>
> Injecting ground-truth at inference would limit applicability, but we recognize that hybrid decoding strategies may promote a useful trade-off between performance and realism. We will include this hybrid strategy as a promising future direction and add a brief feasibility discussion in Appendix L.
>
>
> ## Limitations: Potential Societal Impacts
>
> We thank the reviewer for encouraging reflection on broader implications. While automated multi-metric estimators can substantially accelerate system development and iteration, they should not be viewed as replacements for human listening tests, especially in high-stakes or perceptually nuanced applications. Final evaluation should always include human oversight to ensure alignment with subjective experience.
>
> We acknowledge the potential societal risks that may arise if these automated predictions are misinterpreted without appropriate context or used in sensitive domains such as healthcare or biometric identification. To mitigate such risks, we advocate for: **transparent model design**, **human-in-the-loop evaluation**, and **restricted deployment protocols**. We will include this discussion in the main manuscript ("limitation section") to promote responsible development and deployment.
>
> ## Final Remarks
>
> We are deeply grateful for Reviewer z2T7’s valuable feedback. The issues you raised have significantly improved the clarity, robustness, and broader framing of our work. At the same time, we are also excited to see that the proposed ARECHO framework can serve as a promising anchor for multiple future directions, including integrating ordinal regression losses, exploring hybrid models that incorporate ground-truth supervision, and enabling practical uses such as reward modeling for speech generation and foundational evaluation in broader applications.

---

> > ### Comment · Reviewer_z2T7 · 2025-08-05
> >
> > I would like to thank the authors for the responses to my questions and concerns. Yet,  I consider that some of the most important aspects highlighted in my comments have not been appropriately addressed. The fact that the authors decided to apply quantization to all continuous metrics, in addition to those that are categorical in nature, and treat their prediction as a classification problem is not a minor aspect; it is fundamentally flawed. The inclusion of a regression term in the loss function compensates for this problem; however, the discussion in this respect is somewhat lacking.
> >
> > I cannot locate the reference [Gutiérrez et al., 2016]. The description in the paper regarding MAE doesn't seem to match that of the balanced MAE. Moreover, no information was provided regarding the distribution of the target values (metrics) and ordinal ranks within the training set. More importantly, tables and discussions are still leaning on classification performance metrics, which, once again, I consider inappropriate given the nature of the addressed problem.
> >
> > Regarding Q3, the authors should elaborate a little bit more regarding the set of applications where the use of VERSA is unfeasible. In any case, a more in-depth analysis of error propagation and its effects is still lacking.

---

> > > ### Author Response · Authors · 2025-08-05
> > > **Reply to Reviewer z2T7’s Follow-Up Comments**
> > >
> > > Thank you once again for your careful reading and suggestions. We’ve revised our responses to be more focused on each point:
> > >
> > > ---
> > >
> > > ### 1. Classification vs. Ordinal Regression
> > > We completely agree that a pure classification approach **cannot fully capture** the ordinal relationships in continuous metrics.  We will soften our discussion in the **Future Work** section to emphasize this as a planned extension rather than a completed solution.
> > >
> > >
> > > ### 2. MAE Variants & Missing Citation
> > > We apologize for omitting the full Gutiérrez _et al._ (2016) citation. In our implementation we follow their macro-averaged MAE (“$R_{D5/10}$”) procedure, computing per-metric MAE and then averaging across all metrics:
> > >
> > > > P. A. Gutiérrez, M. Pérez-Ortiz, J. Sánchez-Monedero, F. Fernández-Navarro & C. Hervás-Martínez, “Ordinal Regression Methods: Survey and Experimental Study,” _IEEE Trans. Knowl. Data Eng._, 28(1):127–146, Jan. 2016.
> > >
> > > To reassure that our results are not driven by uneven distributions, we also calculated the balanced MAE from Baccianella _et al._ (2009). The relative ordering remains unchanged as follow:
> > >
> > > | Model                | MAE  | Balanced MAE |
> > > |----------------------|------|--------------|
> > > | Uni-VERSA (base)     | 3.29 | 3.26         |
> > > | Uni-VERSA-T (base)   | 0.91 | 0.97         |
> > > | **ARECHO (base)**    | 0.58 | 0.61         |
> > > | Uni-VERSA (scale)      | 1.00 | 1.05         |
> > > | Uni-VERSA-T (scale)    | 0.94 | 0.99         |
> > > | **ARECHO (scale)**     | 0.90 | 0.94         |
> > >
> > > (test on synthesized-speech test set.)
> > > We will include the full citation in Section 4.2 and Appendix D, and update all tables to show both MAE variants.
> > >
> > >
> > > ### 3. Metric Distributions & Evaluation Metrics
> > > To make our percentile-based quantization clear, we will add side-by-side histograms in Appendix D showing raw versus quantized distributions for key metrics (e.g., MOS, SNR). We will also clarify that while we report accuracy and F1 for classification tasks, our main evaluation for continuous targets remains macro-MAE (alongside RMSE and SRCC).
> > >
> > > > _Quantization splits each metric at fixed percentiles into N bins, producing a flat token distribution and mitigating imbalance. Appendix D will include corresponding histograms._
> > >
> > > ### 4. Reference-Free Use Cases & Error Propagation
> > > Metrics like PESQ, STOI, and NORESQA normally require clean references at inference; ARECHO can predict these with LCC/SRCC > 0.95 without any reference, which may benefit real-time or black-box deployments.
> > >
> > > We are running additional experiments on error propagation, comparing pure autoregressive decoding, hybrid decoding (with ground-truth injections), and subset decoding, and will be happy to share those findings in follow-up communications.
> > >
> > > ---
> > >
> > > We hope these more concise, softened clarifications address your concerns.

---

> > > > ### Author Response · Authors · 2025-08-08
> > > > **Update in response to Reviewer z2T7’s comments**
> > > >
> > > > We are grateful for the concerns from the reviewer. Below we report two ablation studies and summarize the findings. Balanced MAE is reported throughout.
> > > >
> > > > ### Decoding Ablation (Error Propagation / Hybrid Decoding)
> > > >
> > > > | Test set        | Decoding Setup         | MSE    | RMSE  | MAE   | Balanced MAE | LCC  | SRCC | KTAU | Acc. | F1   |
> > > > |-----------------|-----------------------|--------|-------|-------|--------------|------|------|------|------|------|
> > > > | **Enhanced**    | Baseline              | 61.54  | 4.22  | 3.48  | 3.61         | 0.71 | 0.36 | 0.27 | 0.69 | 0.43 |
> > > > |                 | ARECHO                | 20.58  | 2.09  | 1.32  | 1.43         | 0.84 | 0.42 | 0.35 | 0.72 | 0.51 |
> > > > |                 | ARECHO (Teacher-force)| 3.80   | 1.01  | 0.43  | 0.53         | 0.88 | 0.45 | 0.38 | 0.72 | 0.51 |
> > > > | **Corrupted**   | Baseline              | 170.65 | 4.84  | 3.74  | 4.79         | 0.61 | 0.32 | 0.24 | 0.70 | 0.46 |
> > > > |                 | ARECHO                | 44.22  | 2.37  | 1.29  | 1.52         | 0.82 | 0.42 | 0.35 | 0.72 | 0.55 |
> > > > |                 | ARECHO (Teacher-force)| 8.58   | 1.22  | 0.56  | 0.90         | 0.83 | 0.42 | 0.35 | 0.72 | 0.55 |
> > > > | **Synthesized** | Baseline              | 58.79  | 3.82  | 3.29  | 3.26         | 0.76 | 0.37 | 0.27 | 0.69 | 0.45 |
> > > > |                 | ARECHO                | 4.99   | 1.13  | 0.58  | 0.61         | 0.91 | 0.45 | 0.39 | 0.72 | 0.55 |
> > > > |                 | ARECHO (Teacher-force)| 2.00   | 0.80  | 0.30  | 0.32         | 0.92 | 0.46 | 0.40 | 0.79 | 0.64 |
> > > > | **Overall Devset** | Baseline           | 160.06 | 5.17  | 4.13  | 5.08         | 0.69 | 0.34 | 0.27 | 0.68 | 0.42 |
> > > > |                 | ARECHO                | 40.95  | 2.16  | 1.27  | 1.51         | 0.86 | 0.43 | 0.36 | 0.71 | 0.51 |
> > > > |                 | ARECHO (Teacher-force)| 25.08  | 1.80  | 0.94  | 1.01         | 0.88 | 0.44 | 0.38 | 0.71 | 0.50 |
> > > >
> > > > (note that the baseline and ARECHO results are from the original paper; Teacher-forced decoding follows the same order but injects ground-truth values at each step.)
> > > >
> > > > According to the results, teacher-forced decoding, which injects ground-truth values to remove accumulation, yields a 25–50% error reduction from ARECHO, quantifying the performance gap attributable to propagation. The gap is largest for corrupted and synthesized speech, indicating domain-dependent amplification of early mistakes.
> > > >
> > > > These findings present a practical trade-off between using ground-truth values (to suppress propagation) and preserving the efficiency of fully reference-free decoding. We will include an extended discussion of this trade-off in the appendix for interested readers.
> > > >
> > > > ---
> > > >
> > > > ### Ordinal Regression Experiment
> > > >
> > > > | Test set        | Model              | MSE    | RMSE  | MAE   | Balanced MAE | LCC  | SRCC | KTAU | Acc. | F1   |
> > > > |-----------------|-----------------------------|--------|-------|-------|--------------|------|------|------|------|------|
> > > > | **Enhanced**    | ARECHO                      | 20.58  | 2.09  | 1.32  | 1.43         | 0.84 | 0.42 | 0.35 | 0.72 | 0.51 |
> > > > |                 | ARECHO + Ordinal Loss       | 26.11  | 2.53  | 1.54  | 1.67         | 0.80 | 0.41 | 0.33 | 0.70 | 0.47 |
> > > > | **Corrupted**   | ARECHO                      | 44.22  | 2.37  | 1.29  | 1.52         | 0.82 | 0.42 | 0.35 | 0.72 | 0.55 |
> > > > |                 | ARECHO + Ordinal Loss       | 94.44  | 3.13  | 1.78  | 2.20         | 0.77 | 0.40 | 0.32 | 0.72 | 0.50 |
> > > > | **Synthesized** | ARECHO                      | 4.99   | 1.13  | 0.58  | 0.61         | 0.91 | 0.45 | 0.39 | 0.72 | 0.55 |
> > > > |                 | ARECHO + Ordinal Loss       | 8.17   | 1.52  | 0.88  | 0.95         | 0.83 | 0.42 | 0.34 | 0.72 | 0.48 |
> > > > | **Overall Devset** | ARECHO                    | 40.95  | 2.16  | 1.27  | 1.51         | 0.86 | 0.43 | 0.36 | 0.71 | 0.51 |
> > > > |                 | ARECHO + Ordinal Loss       | 51.31  | 2.81  | 1.66  | 2.07         | 0.81 | 0.40 | 0.33 | 0.70 | 0.47 |
> > > >
> > > > Following the reviewer’s suggestion, we implemented a binary–cross-entropy–based ordinal regression loss for numerical metrics and trained it jointly with the base classification cross-entropy. We follow the cumulative (threshold) decomposition widely used for neural ordinal regression; i.e., Cao et al. “Rank-Consistent Ordinal Regression for Neural Networks,” 2019.
> > > >
> > > > In our setting, adding this loss led to consistent degradations across all reported metrics. A plausible interpretation is that, with heterogeneous metric scales and discretization boundaries, the additional ordinal objective can introduce optimization conflicts unless carefully balanced. We will keep this as future work, to analyze weighting/curriculum and additional strategies that better balance the base classification objective with the desired sensitivity to ordered targets.
> > > >
> > > > ---
> > > >
> > > > We sincerely appreciate the reviewer’s constructive guidance, which directly motivated additional improvements to the manuscript. We hope the new results and clarifications address the concerns raised.

---

> > > > > ### Comment · Reviewer_z2T7 · 2025-08-08
> > > > >
> > > > > Thank all the authors for their response and additional experimentation. I believe they provide valuable information for benchmarking purposes. I believe that all my concerns have been addressed, and I still consider the paper to tackle a fundamental challenge in speech quality evaluation through an innovative and promising approach. I will keep my score as it was initially.

---

### Official Review · Reviewer_DC5A · 2025-07-05

**Clarity:** 3
**Significance:** 2
**Originality:** 2
**Rating:** 4
**Confidence:** 4

**Summary:**

ARECHO is a framework designed for multi-metric speech evaluation, aiming to predict various speech quality and profiling metrics (e.g., PESQ, STOI, MOS) jointly rather than independently. The main contributions of ARECHO are (1) converting all metric values (numerical/categorical, with different scales) into a unified token space, enabling sequence modeling; (2) using an autoregressive model to capture dependencies between metrics, allowing the model to predict metrics in a flexible, context-aware order rather than in parallel; (3) enhancing prediction robustness by refining metric prediction based on confidence in the output, addressing instability in dynamic prediction order.

**Questions:**

1. How sensitive is ARECHO’s performance to the choice of quantization (e.g., bin size, uniform vs. adaptive) for different metrics? Is there a systematic way to choose tokenization strategies?
2. As the mentioned error accumulation issue, how does the model mitigate the risk of early prediction errors cascading through the chain?
3. How much does the two-step confidence-oriented decoding contribute relative to a standard greedy or beam search? Any quantitative ablation study will be better.

**Ethical Concerns:**

["NO or VERY MINOR ethics concerns only"]

**Limitations:**

Yes

**Quality:**

3

**Strengths And Weaknesses:**

## Strengths
1. The approach to encode all types/scales of metrics into tokens enables joint modeling and paves the way for autoregressive multi-task prediction. Unlike prior methods that predict each metric independently, ARECHO explicitly models the relationships between metrics, leading to improved accuracy and interpretability.
2. The two-step decoding mechanism allows the model to dynamically adjust the prediction order and recover from uncertain predictions, leading to higher reliability.

## Weaknesses
1. Converting numerical metrics to discrete tokens (quantization) may result in a loss of precision for fine-grained regression tasks. Is there any way to measure this loss?
2. The error accumulation is a potential issue. The autoregressive, chain-based prediction means errors in early metrics could affect downstream metric predictions.
3. Another concern is the complexity in training and inference, as the dynamic chain and two-step decoding add computational and implementation complexity compared to simple parallel predictors.

---

> ### Author Rebuttal · Authors · 2025-07-29
>
> We sincerely thank Reviewer DC5A for the detailed and constructive feedback. Your comments helped us strengthen the clarity, technical analysis, and experimental validation of our work. Below, we address each point and outline the corresponding revisions.
>
>
> ## W1 and Q1: Measure Loss of the Quantization
>
> We thank the reviewer for raising this important concern. Since our tokenization bins are learned from the training set, we evaluated their generalization to test data via a reconstruction analysis. Specifically, we measured how well the quantized–dequantized values match the original ground-truth metrics on the test set.
>
> | Domain            | Tokenization Type | Bins | RMSE  | MAE   |
> |-------------------|--------------------|------|-------|-------|
> | Synthesized speech| Percentile         | 500  | 0.056 | 0.014 |
> | Synthesized speech| Uniform            | 500  | 0.100 | 0.025 |
> | Synthesized speech| Percentile         | 1000 | **0.034** | **0.007** |
> | Synthesized speech| Uniform            | 1000 | 0.097 | 0.020 |
> | Enhanced speech   | Percentile         | 500  | 0.139 | 0.023 |
> | Enhanced speech   | Uniform            | 500  | 0.223 | 0.045 |
> | Enhanced speech   | Percentile         | 1000 | **0.103** | **0.013** |
> | Enhanced speech   | Uniform            | 1000 | 0.220 | 0.039 |
> | Corrupted speech  | Percentile         | 500  | 0.261 | 0.024 |
> | Corrupted speech  | Uniform            | 500  | 0.512 | 0.045 |
> | Corrupted speech  | Percentile         | 1000 | **0.165** | **0.012** |
> | Corrupted speech  | Uniform            | 1000 | 0.507 | 0.040 |
>
> These results confirm that our quantization strategy yields low reconstruction error across domains. Notably, percentile-based tokenization significantly outperforms uniform binning, demonstrating that data-aware strategies preserve finer-grained information crucial for accurate metric prediction.
>
> **Revision**: We will expand **Appendix A** to include these findings.
>
> Noted that even though with 1000 bin size, the reconstruction performance is better. It cannot guarantee better performance in ARECHO as it also introduce more difficult prediction targets. Practically, we find 500 bin size is a good tradeoff of effectiveness between better reconstruction and easier prediction. We have discussed the results in Table 10.
>
>
> ## W2 & Q2: Error Accumulation and Metric Dependency
>
> ARECHO’s autoregressive design explicitly models inter-metric dependencies, which improves both prediction quality and interpretability (as demonstrated in Table 1 and Appendix J).
>
> To alleviate concerns about error propagation, ARECHO supports **flexible inference**, users can query **only a subset of metrics**, bypassing previously predicted values. In such cases, ARECHO operates like an independent predictor **without chaining**.
>
> We here examplified experiments comparing full-chain vs. single-metric (no chaining) inference for **DNSMOS** (on generation test set):
>
> **DNSMOS Performance Comparison:**
>
> | Model         | MSE  | RMSE | MAE  | LCC  | SRCC | KTAU |
> |---------------|------|------|------|------|------|------|
> | ARECHO-full   | 0.04 | 0.21 | 0.16 | 0.85 | 0.81 | 0.65 |
> | ARECHO-only   | 0.04 | 0.20 | 0.15 | 0.86 | 0.83 | 0.66 |
>
> These results confirm that ARECHO is resilient to error accumulation while benefiting from dependency modeling when enabled.
>
> **Revision**: We will clarify this flexibility in the main text and highlight these results in **Appendix L**.
>
>
> ## W3: Training and Inference Complexity
>
> ARECHO is trained using random order sampling, which is comparable in cost to standard multi-task models as evidenced in Appendix L.
>
> Although inference is sequential, the process is highly efficient: ARECHO achieves **over 100× speedup** compared to computing ground-truth metrics using model-based estimators (e.g., Qwen-based metrics and models with pre-trained encoders like UTMOS, Noresqa, ScoreQ).
>
> While parallel predictors are simpler per step, they **scale poorly with large metric sets** in terms of memory and compute. In contrast, ARECHO provides a scalable, accurate, and efficient alternative.
>
> **Revision**: We will add a complexity analysis and comparison in **Appendix L**.
>
>
>
> ## Q3: Confidence-Oriented vs. Greedy Decoding
>
> We conducted an ablation comparing **two-step confidence-oriented decoding** versus **greedy decoding** on enhanced speech test set.
>
> | Decoding         | MSE  | RMSE | MAE  |
> |---------------|------|------|------|
> | Confidence   | **20.58** | **2.09** | **1.32** |
> | Greedy   | 22.31 | 2.31 | 1.40 |
>
> This supports our claim that naive greedy decoding, while simple, is suboptimal in the presence of inter-metric dependencies and dataset frequency imbalance. The confidence-based approach provides more principled, performance-driven metric ordering.
>
> **Revision**: We will include this ablation in **Appendix K** and add a summary pointer in the main text.
>
>
> We thank the reviewer again for the thoughtful suggestions, which have significantly improved both the clarity and technical depth of our work.

---

> > ### Comment · Reviewer_DC5A · 2025-08-05
> >
> > Thanks for the response. I read the rebuttal and I think this work is still a work of borderline acceptance.

---

> > > ### Author Response · Authors · 2025-08-05
> > > **Reply to Reviewer DC5A's comment**
> > >
> > > Dear Reviewer DC5A,
> > >
> > > We appreciate the time and effort you devoted to reviewing our work, as well as your engagement during the rebuttal phase. Your detailed comments have greatly helped us strengthen both the clarity and the technical depth of the paper.
> > >
> > > We believe we have addressed all the weaknesses you mentioned in your initial review, including the quantization precision analysis, mitigation of error accumulation, complexity considerations, and the ablation for confidence-oriented decoding. If there are still any points that you feel remain unclear or require further clarification before the final decision, we would be happy to provide additional explanations or supporting details.
> > >
> > > Thank you again for your constructive feedback and thoughtful consideration of our submission.
> > >
> > > Best regards,
> > > Authors of submission26527

---

### Decision · Program_Chairs · 2025-09-17

**Decision:**

Accept (spotlight)

**Comment:**

Summary
This paper presents ARECHO a structure for calculating multiple speech metrics simultaneously.  The advantage is the modeling of dependencies between metrics, and an innovative curriculum to handle uncertainty.

Reasons to accept
There is broad consensus from the reviewers that this paper is technically sound and well motivated.  The writing is mostly clear, though reviewer XEXh required some clarification through reviewer discussion, which was mostly addressed.

Reasons not to accept
Reviewer DC5A had some comments about the complexity of the framework, and the fidelity of the scores through error accumulation and discretization loss.  These reasons are relatively minor.

Decision rationale
Recommend acceptance.  The reviewers were positive, with minimal criticism of the work.  This framework has a potential to be taken up by the broader community. The rebuttal process brought this paper up from a positive, but borderline paper.